# Additions to the genus *Anthinus* occurring in Minas Gerais and Goiás regions, Brazil, with description of five new species, one of them in the new related genus *Catracca* (Gastropoda, Eupulmonata, Strophocheilidae)

**Luiz Ricardo L. Simone** *

Museu de Zoologia da Universidade de São Paulo, São Paulo, SP, Brazil

* lrsimone@usp.br, lrlsimone@gmail.com

**Data Availability Statement:** All relevant data are within the paper.

## Abstract

Four new species of the strophocheilid genus *Anthinus* are described from Brazil, being *A. synchondrus* from region of Unaí, Minas Gerais; *A. vailanti* from the region of Brasilândia de Minas, Minas Gerais; *A. morenus* from Paracatu region, Minas Gerais; and *A. savanicus* from Formosa region, Goiás. Another similar snail from Itacarambi, Minas Gerais, is described as a new genus and species: *Catracca uhlei*. All species are described conchologically and anatomically, with distinctions explored in almost all structures. For comparative purposes, a similar anatomical investigation on the *Anthinus* type species, *A. multicolor*, from Rio de Janeiro, is also performed. A detailed comparative analysis and discussion is provided, including taxonomic and initial phylogenetic inferences. The preliminary phylogenetic analysis is based on anatomically known species in the literature and in the presently study. A bulimulid, a simpulopsid and two dorcasiids are outgroups. The preliminary clado-gram already shows a monophyletic Strophocheilidae (14 synapomorphies), divided into two also monophyletic subfamilies: Megalobuliminae (14 synapomorphies) and Strophocheilinae (5 synapomorphies). The new *Catracca* resulted as the most basal strophocheilid branch. *Mirinaba* and *Anthinus* were also supported as monophyletic (3 and 6 synapomorphies respectively. Register ZooBank: urn:lsid:zoobank.org:pub:FC4DD323-EF6A-404B-9755-F124F9DBB6D4.

## Introduction

The Strophocheilidae genus *Anthinus* Albers, 1850 ([1]:148) has as the type species *Helix multicolor* Rang, 1831 [2] (Subsequent Designation by Martens [3]: 189–190). In an early monographic treatment of Strophocheilidae, Bequaert ([4]: 196–202), provided detailed taxonomic history of the genus. *Anthinus* characteristically includes the smallest strophocheilids, with adult specimens with shells around 20–30 mm, usually having a tall, bluntly pointed spire, relatively small aperture in a "bulimoid-shape", and a mosaic of brown-beige spots [4,5]. All four

**Funding:** There is no special fund supporting totally or partially the study that generated the present paper. I performed it in my job in the University of São Paulo. The institution contributed with my salary and the usual infrastructure. Is it necessary to inform that? I think that is obvious for any researcher. If so, please, tell me where I inform that I have not received any salary of funders, only from my job.

**Competing interests:** NO authors have competing interests.

valid *Anthinus* species were previously reviewed and illustrated by Simone ([6]: 201–202). Nonetheless, the published anatomical information on *Anthinus* has remained poor, restricted to sparse information and few illustrations [5,7]. The geographic occurrence of *Anthinus* in the southeastern Brazilian Atlantic Rainforest environment, endorsed by Simone ([6]) was from the south region of Minas Gerais-Mato Grosso do Sul in Brazil (latitude ~15˚S) to northern Argentina and Uruguay (latitude ~32˚S). A doubtful occurrence in Tobago has been reported ([6] and references therein; [8]).

In recent years, some samples identified as *Anthinus* have been collected in expeditions by the naturalist José Coltro Jr. and colleagues, in regions from north of Minas Gerais, to Goiás (Brazilian midwest) and in several points in northeast Brazil (latitudes ~15˚S to ~17˚S). This material is used in the present paper. These regions have a dryer environment, including a dryer version of the Atlantic Rainforest, Cerrado and Caatinga semi-arid biomes.

Examination of this material revealed five undescribed species, more than doubling the number of presently known *Anthinus* species. Anatomical information on the type species, *A. multicolor*, collected in Rio de Janeiro, is also included for improving the comparative scenario. The shell of one of the new species, initially appeared to belong to a unicolored *Anthinus*. It has, however, a greatly different set of anatomical features, making necessary the description of a new genus. The description of the type species is presented first, the remaining descriptions and illustrations, to minimize redundancy, are mostly presented compared to it.

The family Strophocheilidae has had an unstable taxonomic history [4,5], but usually it has been considered endemic from South America, with relatives in Africa–the Dorcasiidae; and in Asia–the Acavidae; all grouped in Acavoidea (e.g., [9]), or sometimes in Strophocheiloidea (e.g., [10]). South American members have traditionally been assigned to two families: Strophocheilidae and Megalobulimidae [5]. More recently these two families have been treated as subfamilies in Strophocheilidae, and, together with Acavidae and Dorcasiidae, placed in Rhytidoidea, which has five other families [11,12].

This paper has as main objective the formal introduction of the new strophocheilid taxa. The anatomical information presented on these new taxa will certainly be useful for future taxonomic revisions and for phylogenetic inferences. These aspects are part of a large revisional project in such this study is inserted. An initial phylogenetic treatment is provided in this paper based on species for which anatomy is reasonably well known in the literature, coupled with new information on the species herein erected. The main intention is not to present a "phylogeny of the Strophocheilidae", as anatomical investigation of many more species would be necessary, but as the basis for erection of the new taxa, and to provide a better scenario for a comparative discussion.

## Material and methods

### Material

A complete list of material examined follows each species' description. Material comprised of dry shells or shells and specimens preserved in 70% EtOH. All listed specimens were used to compose the descriptions; those that are only shells (sh) were examined at a conchological level; all complete specimens (shell and soft parts) (spm) were extracted and dissected. The Table 1 summarizes the total number of specimens in each studied species being shells, dissected individuals and measured shells, emphasizing that no statistical study is presently provided, the intention is only giving a notion of size. In the case of *Anthinus multicolor*, only the sample with dissected specimens is reported, but the MZSP collection have other also examined samples in shell characters.

**Table 1. Summary of number of specimens.**

| species | Shells examined | Specimens dissected | Shells measured |
| --- | --- | --- | --- |
| *Anthinus multicolor* | 9* | 7 | 5 |
| *Anthinus synchondrus* | 5 | 2 | 5 |
| *Anthinus vailanti* | 6 | 4 | 5 |
| *Anthinus morenus* | 18 | 10 | 6 |
| *Anthinus savanicus* | 11 | 7 | 5 |
| *Catracca uhlei* | 81 | 5 | 8 |

*Excluding other samples in MZSP collection.

## Methods and scope

Color photos were obtained by digital cameras, either hand-held (Canon Digital Rebel XTi with macro-lenz) or attached to the dissecting microscope (Zeiss AxioCam ERc 5s). Shell measurements were obtained with a digital caliper, for a minimum of 10 adult shells. The specimens were dissected by standard techniques [13] under dissecting stereomicroscopes, with the specimen immersed under the fixative. All drawings were obtained with the aid of a camera lucida; initially penciled, afterwards inked; usually drawings produced for each species include data derived from several specimens, as the intraspecific variation has shown insignificant. Thus, the anatomical drawings are an average (composite) of all examined specimens. The scales were obtained putting a ruler by side of the specimen. The type and voucher material are deposited in Museu de Zoologia da Universidade de São Paulo (MZSP) malacological collection. The specimens usually are relatively easily extracted from their shells, except for the holotype of *Anthinus synchondrus*, in this case a 'cesarean' (a small window) needed to be done in the last whorl. SEM work on the material is standard, with pieces metalized by gold, and examined in the Laboratory of Electron Microscopy of the MZSP. Anatomical terminology, particularly of the odontophore muscles, follows Simone [13]. The material studied and described in this work was collected by a team working for Femorale, a private company [www.femorale.com; http://www.femorale.com/femorale/index.asp]. The places of collection are not within protected areas, and as such collection activity did not require special permits. Nonetheless, the collections were made under general/permanent license IBAMA-Sisbio 10560–2, which permits extraction of wildlife samples for scientific purposes. As most of the studied material was collected by non-scientific expeditions, no further data beyond coordinates and place names were available. Thus, details on vegetation, climate, soil, rainfall, etc., were not available, but, when possible and relevant, these data were extracted from the literature, digital online resources, or official websites. For comparison of the presently studied species with those already known, the large MZSP collection was consulted, including the material assembled by Dr. José L.M. Leme, a previous curator and strophocheilid specialist. Collections of other European and American museums were also consulted in my trips seeking type specimens, some of them illustrated in a catalogue [6]. However, despite high effort, types of not all *Anthinus* species were found. In these cases, voucher material of previous revisions was preferentially studied. Despite having material for a total review of the genus *Anthinus* available, the previous Bequaert's review [4] is so well done that nothing actually new could be informed at shell level. Despite being relatively antique, Bequaert's review [4] is still valid at species' level, and few additional information was published on *Anthinus* after that. Thus, the present paper focuses on the description of the new taxa and the anatomical unpublished features. The type species, however, is included, extending to anatomical features and improving

the comparative taxonomic scenario. The taxonomical remarks on each species are all reunited in the Discussion item.

## Phylogenetic analysis

The phylogenetic methodology is the same as reported by Simone [13,14], that basically consists of the matrix mounted in Nexus, analyzed by programs TNT and PAUP. All of them resulted in a single cladogram shown below. Thus, beyond the species herein studied, the present preliminary phylogeny is based upon already published data of 14 species listed below, as well as additional examination of their voucher material deposited in MZSP in some unstudied structures, such as, e.g., the odontophore. The list of characters is in Appendix 1; and respective matrix in Appendix 2. It is important to emphasize that the shown phylogeny is not to be interpreted as "the phylogeny of the Strophocheilid". It has only the intention of demonstrating that the description of the new genus–*Catracca*–is necessary. In the literature, a small set of strophocheilids has their anatomy known in sufficient details for an initial phylogenetic inference. They are (1) *Mirinaba antoninensis* (Morretes, 1952) [15,16]; (2) *Mi. cadeadensis* (Morretes, 1952) [16]; (3) *Strophocheilus debilis* Bequaert, 1948 [5]; (4) *Megalobulimus lopesi* Leme, 1984 [17]; (5) *Me. grandis* (Martens, 1885) [17]; (6) *Me. parafragilior* Leme & Indrusiak, 1990 [18]; (7) *Me. proclivis* (Martens, 1888) [19]; (8) *Me. dryades* Fontenelle, Simone & Cavallari, 2021 [20]; (9) *Me. oblongus* (Müller, 1774) and (10) *Me. conicus* (Bequaert, 1948) [21]. As remote outgroups, a bulimulid (*Leiostracus carnavalescus* Simone & Salvador, 2016 [22]) and a simpulopsid (*Rhinus botocudus* Simone & Salvador, 2016 [22]), and close outgroups a pair of African dorcasiids (*Dorcasia alexandri* Gray, 1838 and *Trigonephrus porphyrostoma* (Melvill & Ponsonby, 1891) [23]) were chosen. All six species studied herein were also included.

## Nomenclatural acts

The electronic edition of this article conforms to the requirements of the amended International Code of Zoological Nomenclature, and hence the new names contained herein are available under that Code from the electronic edition of this article. This published work and the nomenclatural acts it contains have been registered in ZooBank, the online registration system for the ICZN. The ZooBank LSIDs (Life Science Identifiers) can be resolved and the associated information viewed through any standard web browser by appending the LSID to the prefix "http://zoobank.org/". The LSID for this publication is: urn:lsid:zoobank.org:pub: FC4DD323-EF6A-404B-9755-F124F9DBB6D4. The electronic edition of this work was published in a journal with an ISSN, and has been archived and is available from the following digital repositories: PubMed Central, LOCKSS, ResearchGate.

**Abbreviations in the Figures**: **aa**, anterior aorta; **ac**, albumen chamber; **ad**, albumen gland duct; **af**, anal folded subterminal portion; **ag**, albumen gland; **an**, anus; **as**, accessory genital gland; **au**, auricle; **bc**, bursa copulatrix; **bd**, bursa copulatrix duct; **bg**, buccal ganglion; **bm**, buccal mass; **br**, subradular membrane; **bv**, blood vessel; **cc**, cerebral commissure; **ce**, cerebral ganglion; **cd**, cerebral node; **cn**, cerebro-pedal and cerebro-pleural connectives; **cv**, pulmonary (efferent) vein; **da**, digestive gland anterior lobe; **dc**, dorsal chamber of buccal mass; **dd**, gastric duct to digestive gland; **df**, dorsal folds of buccal mass; **dg**, digestive gland posterior lobe; **di**, diaphragm or pallial floor; **dl**, left esophageal duct to digestive gland; **dr**, right esophageal duct to digestive gland; **eh**, epiphallus; **eo**, spermoviduct; **es**, esophagus; **ey**, eye; **fo**, free oviduct; **fp**, genital pore; **ft**, foot; **gm**, genital muscle; **go**, gonad; **hd**, hermaphrodite duct; **hp**, hump of free oviduct; **in**, intestine; **jw**, jaw; **kf**, kidney anterior fold; **ki**, kidney; **lf**, lateral fold of buccal mass; **m1–m10**, extrinsic and intrinsic odontophore muscles; **mb**, mantle border (edge); **mf**, mantle fold; **mj**, jaw and peribuccal muscles; **mo**, mouth; **mp**, prerectal muscle; **ne**, nephrostome; **nr**,

nerve ring; **oc**, odontophore cartilage; **od**, odontophore; **om**, ommatophore; **on**, optical nerve; **ou**, ommatophore muscle; **pa**, penis anterior chamber; **pc**, pericardium; **pe**, penis; **pf**, penis fold; **pg** pedal gland; **pm**, penis muscle; **pn**, pneumostome; **pp**, pedal ganglion; **pr**, penis aperture; **ps**, penis posterior chamber; **pt**, prostate; **pu**, pulmonary cavity; **pv**, pneumostome inner flap; **ra**, radula; **rn**, radular nucleus; **rs**, radular sac; **rt**, rectum; **sa**, salivary gland aperture; **sc**, subradular cartilage; **sd**, salivary gland duct; **se**, septum between esophagus and odontophore; **sf**, stomach inner fold; **sg**, salivary gland; **sh**, penis central sphincter; **sm**, secondary cephalic muscles; **sp**, sperm inner longitudinal fold; **sr**, seminal receptacle; **st**, stomach; **sw**, stomach muscular walls; **te**, ventral cephalic tentacle; **tg**, integument; **te**, tentacle; **tm**, tentacle muscle; **tn**, tentacular nerve; **un**, union of mantle border with nuchal surface; **ur**, urinary gutter; **ut**, uterus; **vd**, vas deferens; **ve**, ventricle; **vf**, vaginal fold; **vg**, vagina.

Additionally in the text, the following abbreviations are used: **L**, length; **sh**, empty dry shell; **spm**, complete specimen (shell and soft parts); **W**, width. Institutions: **MNRJ**: Museu Nacional da Universidade Federal do Rio de Janeiro; **MZSP**: Museu de Zoologia da Universidade de São Paulo; **USNM**: National Museum of Natural History, Smithsonian Institution.

## Results

### Comparative conchology and anatomy

**Systematics.** Genus *Anthinus* Albers, 1850

*Bulimus* (*Anthinus*) Albers, 1850[1]: 148; Martens, 1860 [3]: 189.

*Auris* (*Anthinus*): Pilsbry, 1895 [24]: 95.

*Gonyostomus* (*Anthinus*): Bequaert, 1948 [4]: 196; Oliveira et al., 1968 [8]: 10.

*Anthinus*: Morretes, 1949 [25]: 145; Leme, 1973 [5]: 332; Simone, 2006 [6]: 201.

**Diagnosis.** Shell of ~30 mm, apex blunt, aperture ~half of shell length. Protoconch of 3–4 whorls, sculptured by delicate reticulation of spiral and axial lines. Color mosaic of brown, beige and cream wide spots, with tendency of spiral organization in squared irregular spots. Head-foot lacking labial flange. Lung with not-protruded vessels, dark pigmented. Odontophore pair m5 ~half originated from cartilages and ~half from m4, free from m2. Pair of anterior (esophageal) ducts to digestive gland, located posteriorly, practically from stomach; both components (dr, dl) with about same caliber, left duct much longer. Prerectal pallial muscle (mp) usually present. Anus turned externally in pneumostome right corner. Seminal receptacle strongly curved. Spermoviduct accessory gland (as) present. Penis with single inner chamber; epiphallus short, rounded, vas deferens insertion subterminal, inner surface papillate; inner main chamber with main pair of strong longitudinal folds, sulcus between these folds continuous to epiphallus. Other details in [4,5].

**List of included species.** *Anthinus multicolor* (Rang, 1831) (type species); *A. albolabiatus* (Jaeckel, 1927); *A. miersi* (Sowerby, 1838); *A. turnix* (Gould, 1846); *A. synchondrus* new species; *A. vailanti* new species; *A. morenus* new species; *A. savanicus* new species.

**Taxonomic discussion.** see Discussion item.

***Anthinus multicolor* (Rang, 1831)** Figs 1–4.

Previous synonymy see Bequaert (1948 [4]: 196–197). Complement:

*Helix* (*Cochlogena*) *multicolor* Rang, 1831 [2]: 55 (pl. 3, fig. 1).

*Gonyostomus* (*Anthinus*) *multicolor*: Bequaert, 1948 [4]: 196–198 (pl. 4, fig. 5; pl. 9, figs 3, 7; pl. 13, fig. 6; Pl. 17, fig. 4).

*Anthinus multicolor*: Morretes, 1949 [25]: 145; Leme, 1973 [5]: 332; Abbott, 1989 [26]: 76 (fig.); Salgado & Coelho, 2003 [27]: 156; Simone, 2006 [6]: 202 (fig. 760).

*Gonyostomus multicolor*: Oliveira et al., 1968 [8]: 10; 1981 [28]: 341.

**Types.** whereabouts unknown [4].

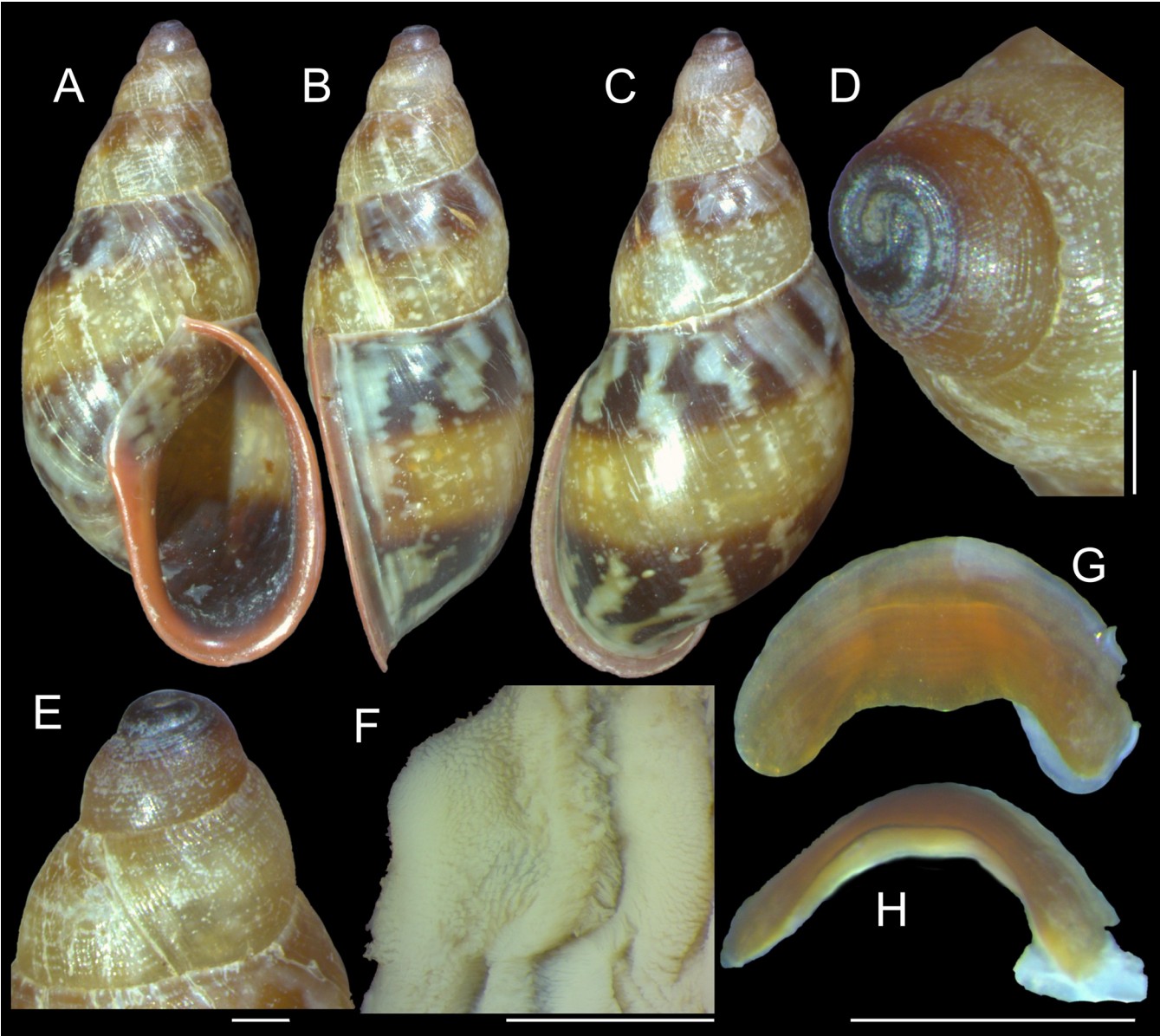

**Fig 1.** *Anthinus multicolor*, **shell and some anatomical structures, sample MZSP 152925.** (A-C) shell of specimen #2 (L 30.2 mm), frontal, right and dorsal views. (D) same, detail of apex, apical-slightly dorsal views. (E) same, protoconch and part of first teleoconch whorl in profile. (F) penis, inner surface, detail of middle and distal thirds showing papillae. (G) jaw, ventral view. (H) same, anterior view. Scales = 1 mm.

**Type locality.** São Paulo State and near Corcovado, Rio de Janeiro.

**Diagnosis.** Shell about 35 mm; color usual mosaic of brown and beige spots, lacking spiral alignment; spire angle ~40˚, protoconch of 2.5 whorls; peristome ~52% of shell length and ~55% of width; width ~46% of length. Mantle edge with simple, rounded fold. Reno-pericardial area small-sized. Pre-rectal pallial muscle absent. Odontophore with cartilages ~90% fused with each other; pair m4 ~50% originated from cartilage, remaining on m4. Jaw plate broad, thick. Radula with short, hook-like mesocone, rachidian different from neighboring teeth. Seminal receptacle widely curved, hermaphrodite duct inserting in its base, curved. Accessory genital gland in spermoviduct small; with pair of similar-shaped sperm-grooves. Free oviduct

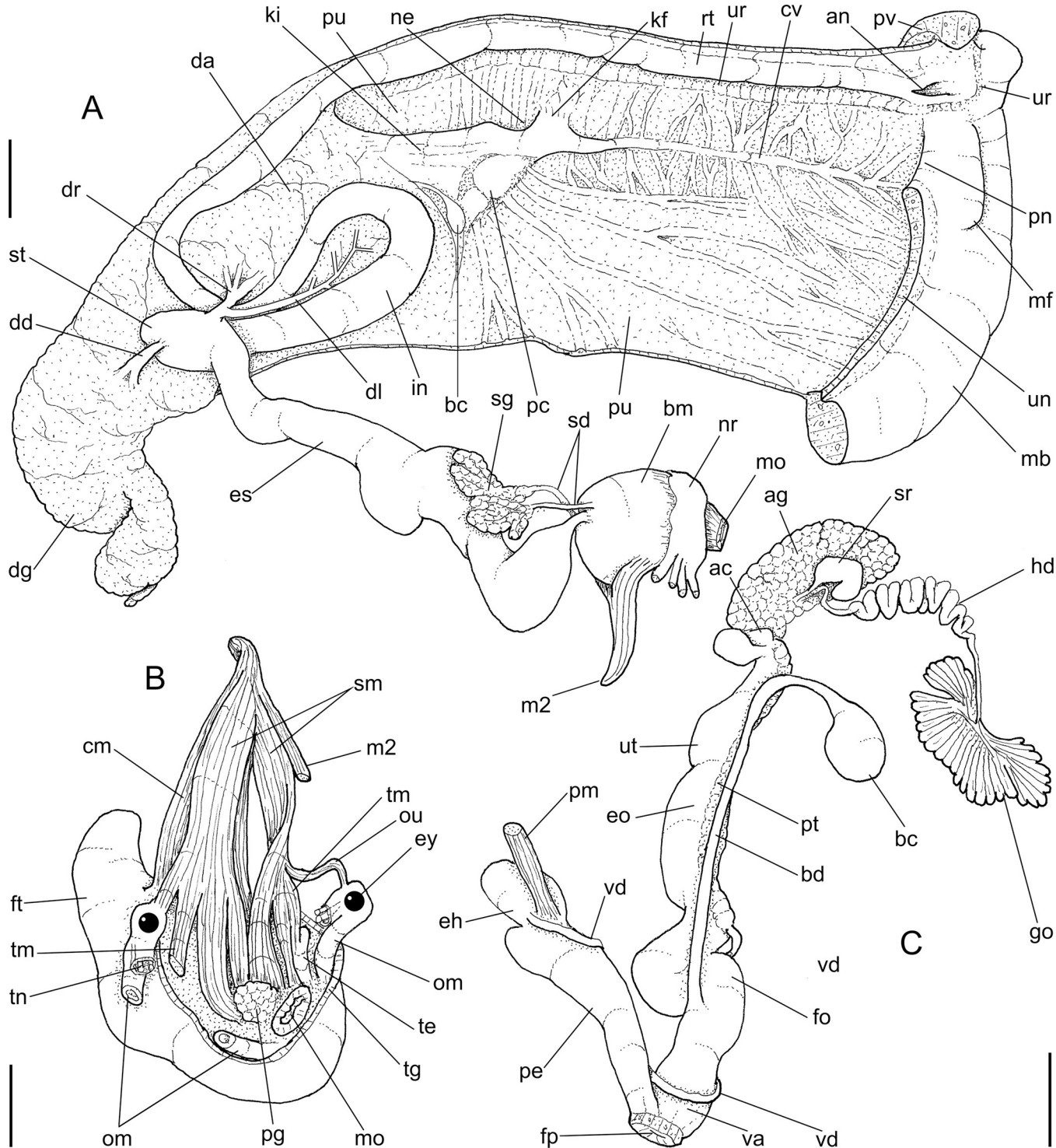

**Fig 2. *Anthinus multicolor*, anatomy.** (A) pallial cavity, partially uncoiled visceral mass and digestive system, mostly ventral view, inner lip of pneumostome sectioned and deflected upwards. (B) head-foot, dorsal view, most of dorsal integument and inner organs removed, main concern to retractile/columelar musculature, some right-anterior structures sectioned. (C) genital system, mostly dorsal view. Scales = 3 mm.

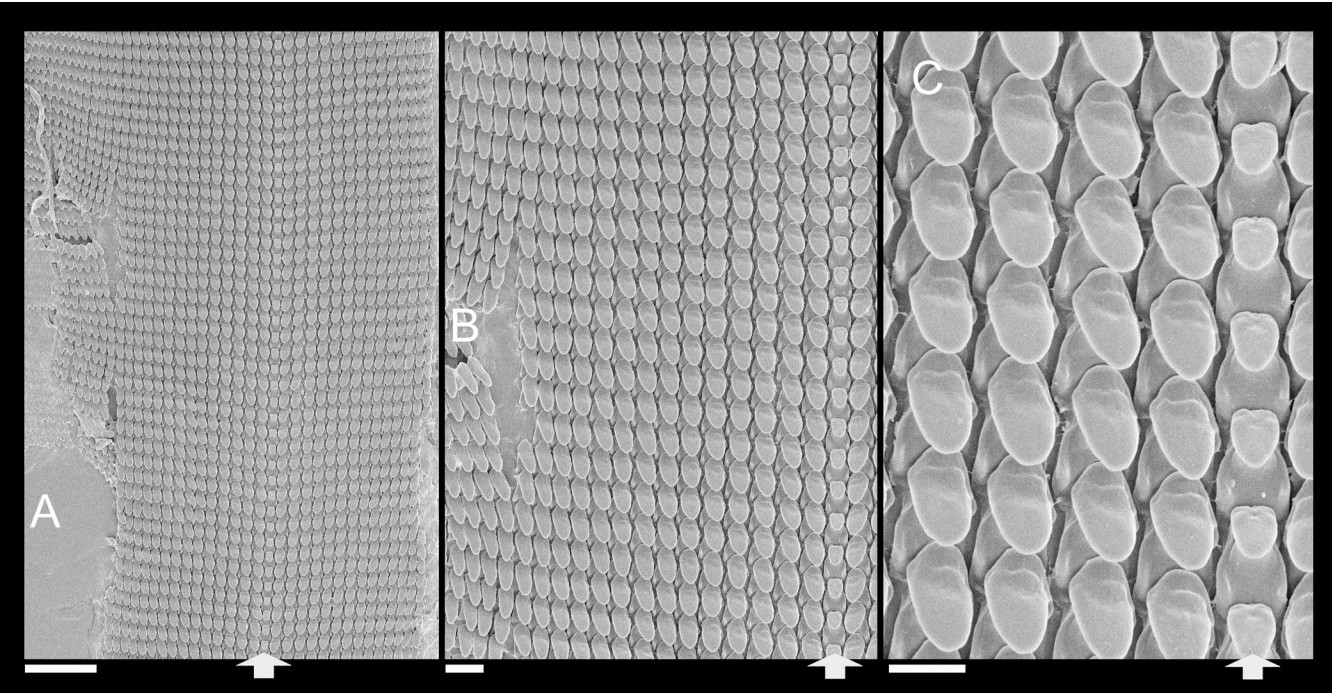

**Fig 3. *Anthinus multicolor*, radula in SEM, MZSP 152925.** (A) wide view, scale = 200 μm; (B) detail of central region, scale = 50 μm; (C) same, higher magnification, scale = 30 μm. White arrows indicating rachidian column.

with distal hump. Penis length ~50% of spermoviduct; penis muscle mostly inserted to penis' tip, with small branch to epiphallus and another small branch to adjacent vas deferens; epiphallus papillae tall, uneven, extending along entire penis chamber; large pair of penial inner folds wide and low, covered by papillae.

**Redescription. Shell.** (Fig 1A–1E) proper description in [4]. Complement: Adult shell around 35 mm, conical-oval; apex blunt; wider on last whorl; width ~46% shell length. Walls relatively thin. Color sometimes entirely brown ([6]: 202), but usually with reddish beige spiral band along inferior half of each spire whorl, or central region of body whorl (Fig 1A–1C); white spots randomly splayed, overlapping remaining colors, usually coalescent near superior suture, and also inferior third of last whorl (Fig 1B and 1C), barely forming irregular axial bands; small white punctiform spots splayed in inferior half of spire whorls and middle region of body whorl. Spire angle ~40˚. Protoconch of 2.5 whorls (Fig 1D and 1E), ~12% of length, width of ~6.1 mm, slightly taller than wide; first whorl smooth, gradually spiral narrow aligned punctuations appearing (Fig 1D and 1E), lacking axial cords except for relatively regular axial undulations, ~3 per mm, last protoconch whorl with spiral sculpture more spaced, ~8 in last protoconch whorl. Teleoconch of ~2 whorls, profile slightly convex; sculpture similar to that of protoconch last whorl, spiral punctuations (10 in penultimate whorl) and axial undulations; last 1.5 whorls lacking spiral punctuations, only having smooth growth lines (Fig 1C). Whorls profile weakly convex (Fig 1A–1C). Suture well-marked, not channeled (Fig 1C). Aperture slightly prosocline (~10˚ from longitudinal general axis) (Fig 1B), elliptic; ~52% of shell length, ~55% of shell width. Peristome continuous, reflected, thick, color red (Fig 1A); outer lip arched, lacking middle tooth; inner lip with inferior half almost straight, superior half as broad, weak callus with ~2/5 of peristome length (Fig 1A), weakly convex. Inferior half of

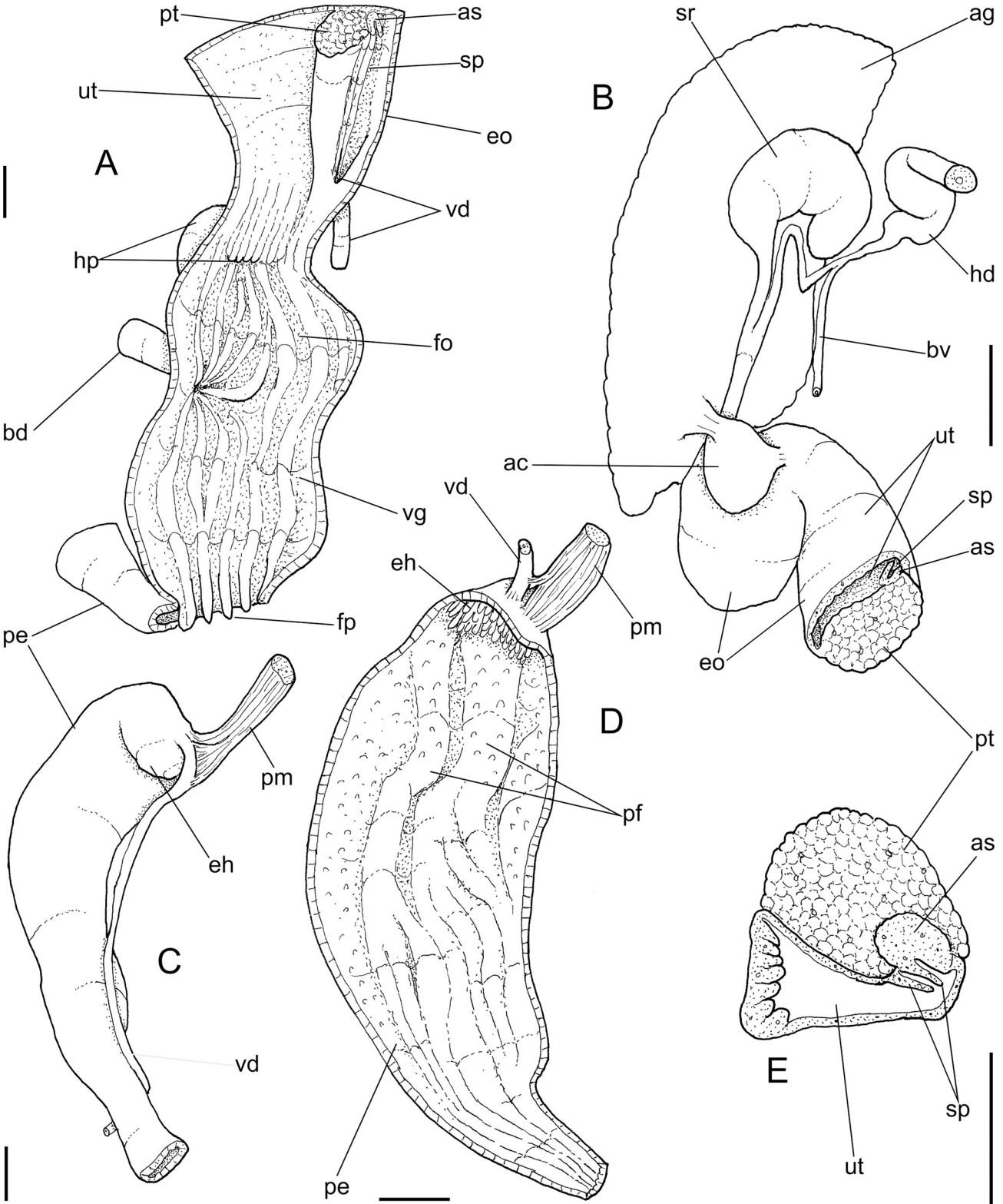

**Fig 4. *Anthinus multicolor*, anatomy.** (A) anterior half of genital tubes, opened longitudinally. (B) carrefour region of genitalia and posterior region of spermoviduct, albumen gland seen as transparent. (C) penis, ventral view. (D) same, opened longitudinally showing inner surface. (E) spermoviduct, transverse section in its middle region. Scales = 1 mm.

inner lip covering umbilicus. Umbilicus relatively wide, ~10% of inferior area. Body whorl ~70% shell length; uniform with spire.

**Head-foot.** (Fig 2B) of normal shape. Color uniformly pale. Columellar muscle thick, 1.5 whorls in length. Inner arrangement of columellar annexed muscles relatively complex. Main columellar bundle (cm) occupying ventral floor of haemocoel, relatively flat, wide slightly broader than half of foot width. Pair of secondary cephalic muscles (sm) with middle and anterior regions approximately double thickness and half width those of columellar main bundle (left muscle slightly broader than right one); originating gradually along dorsal surface of main columellar bundle, running anteriorly on main bundle; anterior third of each muscle with four parallel branches, two medial branches broad (~2/3 of its basal width) inserting in ventral region of perioral region. Branches 3 and 4 respectively tentacular and ommatophore muscles; ommatophore muscles (ou) narrower, mode lateral, inserting in tip of ommatophore; tentacular muscle (tm) longer, inserted in tip of tentacles. Pedal gland (pg) protruding in posterior region of buccal area.

**Mantle organs.** (Fig 2A) Mantle border thick, lacking pigments. Pneumostome (pn) protected by simple right ventral flap (pv), width ~1/5 of aperture length. Dorsal fold well developed (mf), occupying ~1/3 of dorsal mantle edge length; rounded. Pneumostome (pn) ~1/10 of shell aperture length, bearing exclusively air entrance and urinary gutter. Anus (an) separate aperture located at right, adjacent to pneumostome. Lung of 1.2 whorls in length, wide and elongated; right side ~1.5 times longer than left side. Pulmonary venation well-developed; posterior region of pulmonary vein (cv) protruded; left half only having longitudinal vessels, right half mostly having perpendicular vessels, except for oblique vessels in region preceding pneumostome. Pulmonary vein (cv) running longitudinally across pallial cavity roof medially or towards right, anterior end trifurcated. Reno-pericardial area beige, slightly triangular, located posteriorly within pallial cavity, its posterior abutting wall of visceral cavity, occupying ~10% of cavity length and ~40% of its width (details below). Rectum (rt) wide. Urinary gutter (ur) broad, smooth, lacking clear transverse folds; running along rectum; anterior urinary gutter surrounding left anal aperture.

**Visceral mass.** (Fig 2A) ~3 whorls in length. Both digestive gland lobes brown in color; anterior lobe (da) flattened, occupying ~1/5 of visceral volume, located just posteriorly to pallial cavity, continuous to kidney. Posterior lobe (dg) larger, extending 2 spiral whorls, occupying ~60% of visceral volume. Stomach ~1/15 of visceral volume, located between both digestive gland lobes, about 3/4 whorl posterior to pallial cavity (st). Digestive tubes (described below) surrounding anterior lobe of digestive gland. Gonad multi-lobed, cream color, encased between posterior lobe of digestive gland and columella, occupying ~1/3 whorl, ~1/10 of visceral volume.

**Circulatory and excretory systems.** (Fig 1A) Pericardium (pc) ~twice as long as wide, located obliquely between middle and left thirds of posterior end of pallial roof, appressed against right lateral side of kidney; occupying ~5% of lung area. Auricle located anteriorly, as continuation from pulmonary vein (cv); ventricle located posteriorly, larger. Kidney (ki) simple, dorso-ventrally flattened; size reported above; somewhat triangular, width ~2/3 of length; internally organized as two distinct regions–a longitudinal hollow cavity on left; filled by sponge-like renal tissue, clearly U-shaped (convexity left) in section. Nephropore (ne) small, longitudinal slit at anterior-left corner of kidney, directed towards right; protected by curved anterior projection of kidney (kf). Urinary gutter (ur) reported above.

**Digestive system.** (Fig 2A) Mouth (mo) and oral tube wide, muscular. Jaw plate (Fig 1G and 1H) thick; cutting edge blunt, sigmoid; lacking inner reinforcement, immersed in jaw thickness. Buccal mass spherical, occupying~1/5 of haemocoel volume. Dorsal surface of oral cavity with well-developed pair of dorsal folds, width of each ~1/3 of dorsal wall width;

separated from each other by dorsal chamber (dc) as wide as folds, but broader anteriorly. Odontophore (od) with ~60% of buccal mass volume. Odontophore muscles: **mj**, jaw and peribuccal muscles originating in outer-ventral surface of odontophore cartilages, running towards dorsal, splaying in dorsal wall of oral tube; **m1**, jugal muscles covering entirely haemocoelic structures, more concentrated close to mouth; **m1v**, small pair of ventral protractors jugal muscles, originating in ventral surface of haemocoel close to mouth, running towards posterior, inserting in ventral-posterior region of odontophore close to m2 insertion; **m2**, strong pair of retractor muscles of buccal mass, or radular muscles, originating as single bundle in columellar muscle posterior end, running anteriorly close to median line along ~60% of haemocoel length, inserting as two different bundles in ventro-posterior edge of odontophore, surrounding at some distance radular nucleus; **m3d**, thin layer of longitudinal fibers immersed in dorso-posterior wall of odontophore, preceding esophageal origin; **m4**, main pair of dorsal tensor muscles of radula, very thick, originating in postero-medial region of odontophore cartilages, surrounding outside and medially cartilages, inserting in subradular membrane in its region correspondent to buccal cavity; **m5**, pair of thick auxiliary dorsal tensor muscles of radula, originating half on posterior-medial surface of m4, and half on portion in postero-ventral region of odontophore cartilages, running towards median line covering m4, inserting in subradular membrane by side of m4 insertion; **m6**, horizontal muscle absent; **m7**, absent; **m10**, pair of narrow ventral odontophore protractor muscles, originating in ventro-anterior region of haemocoel, just ventral to mouth, running towards posterior along buccal mass length, inserting in latero-posterior surface of odontophore close to m2 insertions; **m11**, pair of narrow ventral tensor muscles of radula absent. Odontophore non-muscular structures: oc, pair of odontophore cartilages flattened, rather elliptical, anterior region slightly projected anteriorly close to median line, ~1.2 times longer than wide, fused with each other along ~90% in their anterior-medial edge, posterior end roughly rounded; **sc**, subradular cartilage, with expanding region in buccal cavity protecting subradular membrane. Radular sac short, not extending beyond odontophore.

**Radula.** (Fig 3) Slightly longer than odontophore; with rachidian teeth, and ~30 pairs of lateral/marginal teeth; no clear distinction between lateral and marginal teeth (Fig 3A), marginal teeth only slightly narrower than lateral teeth, and more inclined medially (Fig 3A and 3B: left side); all teeth with base as long as mesocone, articulating neighboring rows (Fig 3C); mesocone arched, curved inwards and posterior, ~1.5 longer than wide, apex rounded, base reinforcement slightly more developed, subterminal; no secondary cusps. **Rachidian** tooth (Fig 3B and 3C: arrow) as large as lateral teeth; base ~twice longer than wide, slightly flattened, barely triangular, with arched posterior edge articulating with neighbor tooth and pair of distal reinforcements; mesocone reduced, ~half size of those of lateral teeth, rounded, covering ~1/3 of base; width and length similar to those of lateral teeth. **Lateral teeth** similar to rachidian, except in being slightly longer, longer asymmetrical mesocone, arched towards medial region; base trapezoid, with medial concavity articulating with neighbor medial tooth and single distal reinforcement (Fig 3C). **Marginal teeth** starting with no clear boundary with lateral teeth (Fig 3A and 3B), occupying ~1/3 of each side; shaped similarly to lateral teeth, except for being slightly narrower and more inclined medially, row practically straight (Fig 3A: left side).

Salivary glands small, covering ~1/5 of esophagus length, in its anterior quarter region (Fig 2A: sg), forming two rounded, white, thin masses. Each salivary duct differentiable in middle and anterior side of glands, with ~1/12 of esophageal width (sd). Salivary duct running in both sides of esophageal origin, penetrating buccal mass wall in region close to buccal ganglia, running immersed in buccal dorsal wall along ~1/3 its length. Salivary ducts opening as small slits in located in posterior region of lateral edges of wide dorsal folds.

Esophagus ~1-whorl long, with thin, flaccid walls (Fig 2A: es); anterior 1/2 weakly broader, crop-like; inner surface simple, smooth. Stomach (st) narrow, curved, weakly bulging; position and size described above (visceral mass); gastric walls thin, weakly muscular; inner surface mostly smooth, but with chitinous inner cover well-developed. Esophageal insertion on right side, intestinal origin on left side, both close to columella. Duct to anterior lobe of digestive gland at short distance from esophagus and intestine intersection (dr) narrow, running shortly towards right, bifurcating; additional left duct (dl) originated in main duct base with ~1/3 of its width, running narrowly towards left and anterior along digestive gland lobe compressed by first intestinal loop (da), periodically possessing short ramifications along its length. Duct to posterior lobe of digestive gland located short distance from intestinal origin, posterior to above-described duct, directed towards opposite side (dd) as wide as anterior duct. Intestine (in) initially as wide as esophageal insertion, gradually narrowing up to ~half that width along its wide sigmoid loop in anterior lobe of digestive gland (in). Rectum and anus position described above (pallial cavity) (rt, an). Anus sessile, as slit in right end of mantle edge directly turned outside; inner surface with 8–10 simple longitudinal, relatively tall folds.

**Reproductive system.** (Fig 2C, 4) Gonad position described above (visceral mass), composed of 8–10 lobes with minute digitiform acini. Hermaphroditic duct (Fig 2C: hd) broad; coiled portions occupying anterior 2/3, with very strong coils; insertion strongly curved (Figs 2C and 4B: hd). Seminal receptacle (Figs 2C and 4B: sr) relatively large, softly and amply curved (convexity left, wide), insertion of hermaphroditic duct more anterior, at middle level of seminal receptacle duct. Fertilization complex simple, located at narrow and elongated base of seminal receptacle as duct of seminal receptacle; slightly shorter than length of receptacle. Fertilization complex totally immersed in albumen gland, inserting in posterior end of spermoviduct, by side of albumen gland duct, relatively wide, twice wider than receptacle's duct. Albumen gland (ag) solid, white, elliptical, more-or-less same size as gonad (~1/3 whorl). Albumen gland duct subterminal, connected to distal end of spermoviduct (Fig 4B), in lateral, small albumen chamber (Fig 4D: ac); narrowly connected to distal end of spermoviduct (Fig 4B: eo). Spermoviduct (Figs 2C, 24A and 24B: eo) of ~1.5 whorl in length, slightly narrower than albumen gland, ~10 times longer than wide; prostate wide (pt); glandular walls in uterus only along lateral side (Fig 4B and 4E: ut), remaining ~2/3 of uterine inner surface relatively thin-glandular. Presence of accessory genital gland (Fig 2B and 2E: as) lying along spermoviduct, flanking masculine furrows (sp), bulging along prostate (pt) side. Pair of masculine furrows lying along accessory genital gland (Fig 4B and 4E: sp). Vas deferens originating in simple aperture slightly posterior to end of uterine level (Fig 4A: vd). Vas deferens not coiled (Figs 2C and 4C: vd). Genital muscles not detectable. Penis slightly shorter than half of spermoviduct length (Fig 2C: pe); penis muscle inserted only between penis and epiphallus (Fig 2C: pm), with small branch connected to vas deferens in its region preceding its insertion (Fig 4C: pm). Epiphallus (eh) ~1/6 penis' length; vas deferens also inserted in middle level of outer epiphallus wall (Fig 4C: vd). Epiphallus inner surface with high layer of papillae (Fig 4D: eh). Internal penial arrangement of folds similar, but with fewer and wider folds, pair of more developed (main) folds (Fig 4D: pf) not so developed, being wider and lower; papillae slightly smaller than those inside epiphallus also lying along entire distal 2/3 of penis (Figs 2F and 4D).

**Central nervous system.** (Fig 2A: nr) More detailed in following studied species, being noteworthy cerebral commissure wide; cerebral node (cd) not so developed; all connectives slightly shorter and all ganglia slightly larger.

**Distribution.** Apparently restricted to Rio de Janeiro region. Remaining records to other states possibly are misidentifications (personal observation, upcoming study).

**Habitat.** Atlantic Rainforest.

**Measurements.** (in mm) MZSP 152925#2 (Fig 1A–1C): 30.2 by 13.3.

**Material examined.** BRAZIL. **Rio de Janeiro**; São Fidélis, Serra da Sapateira, 21˚39'S 41˚46'W, MZSP 152925, 9 specimens (2 with only shells) (Femorale leg., 2010).

**Taxonomic remarks.** See Discussion item.

*Anthinus synchondrus* **new species** (Figs 5–8).

ZooBank: urn:lsid:zoobank.org:act:64ABD2D2-71C4-49D7-AA69-4321AA8595EB.

**Types.** Holotype MZSP 152074. Paratypes: MZSP 152252, 2 shells, MZSP 152075, 1 shell from type locality. BRAZIL. **Minas Gerais**; Unaí (W.V. Matos col, iv/2020), Gueda, 16˚16'04"S 46˚39'43"W, 647–650 m altitude, MZSP 152236, 1 shell.

**Type locality.** BRAZIL. **Minas Gerais**; Unaí, Pedra da Fartura, 16˚31'37"S 46˚49'18"W, 628–750 m altitude [Weslley Vailant de Matos col, iv.2020].

**Etymology.** The specific epithet is derived from Greek words *syn*, a prefix meaning "together", and *chondros*, meaning "gristle, cartilage". This is an allusion to the total fusion of the odontophore cartilages (Fig 6D).

**Diagnosis.** Shell about 40 mm; color mosaic of brown-beige spots; spire angle ~35˚, protoconch of 3.2 whorls; peristome ~48% of shell length and ~50% of width; width ~54 of length. Mantle edge with projected fold. Reno-pericardial area large. Pre-rectal pallial muscle present. Odontophore with cartilages totally fused; pair m4 ~50% originated from cartilage, remaining on m4. Jaw plate narrow. Radula with elongated, spatula-like mesocone, rachidial slightly different from neighboring teeth. Seminal receptacle tightly curved, hermaphrodite duct inserting in its base perpendicularly. Accessory genital gland in spermoviduct large; with two similar sperm-grooves. Penis ~60% of spermoviduct; penis muscle with pair of small branches in epiphallus; epiphallus papillae low, uniform, restricted to it; pair of larger folds composed of transverse subfolds.

**Distinctive description. Shell.** (Fig 5A–5L) Adult shell around 40 mm, conical-oval; apex blunt; wider at middle of last whorl; width ~54% of shell length. Walls weakly thick. Dorso-ventrally weakly flattened (Fig 5E). Color light beige and brown squared spots, both predominating (Fig 5C) or slightly brown predominating (Fig 5H and 5J); brown spots slightly randomly distributed along 4–5 spiral, wide, rather isometric bands in last whorl, 2–3 spiral bands in penultimate whorl; usual concentration of brown pigment preceding peristome. Spire angle ~35˚. Protoconch of 3.2 whorls (Fig 5B, 5J and 5K), ~23% of length, width of ~7 mm, ~twice taller than wide; first whorl smooth, gradually axial and spiral narrow cords appearing (Fig 5K and 5L), initially both predominating, gradually axial cords slightly stronger, from suture to suture, interspaced slightly broader than each cord, ~10 per mm; transition to teleoconch clear, slightly prosocline (Fig 5B, 5H and 5K). Teleoconch sculpture similar to that of protoconch last whorl, as axial, uniform, complete cords and growth lines, gradually fading in last whorl (Fig 5C, 5H and 5J); of ~2.3 whorls. Whorls profile weakly convex (Fig 5G–5J) (Fig 5A–5C, 5F, 5I and 5J). Suture well-marked, weakly channeled (Fig 5J, 5K and 5L). Aperture slightly prosocline (~5˚ from longitudinal general axis) (Fig 5B), oval; ~48% of shell length, ~50% of shell width. Peristome well-reflected, thick, color variating from pale beige (Fig 5A and 5G) to white (Fig 5I); outer lip arched, lacking middle tooth; inner lip with inferior half widely concave, superior half as narrow, weak callus with ~1/3 of peristome length (Fig 5A and 5I), usually concave, rarely convex (Fig 5G). Inferior half of inner lip intensely covering umbilicus (Fig 5A, 5D, 5G and 5I). Umbilicus narrow (Fig 5B). Body whorl ~55% of shell length; usually uniform with spire (Fig 5A, 5G and 5I).

**Head-foot.** Similar features as *A. multicolor*, including columellar muscle having only two layers; i.e., secondary head muscles (ou, tm) originating direct from pair of secondary muscles (tm).

**Mantle organs.** (Fig 6A) most characters similar to *A. multicolor*. Distinctions and remarks following. Mantle edge dorsal fold well developed (mf), with elongated edge pointed to left.

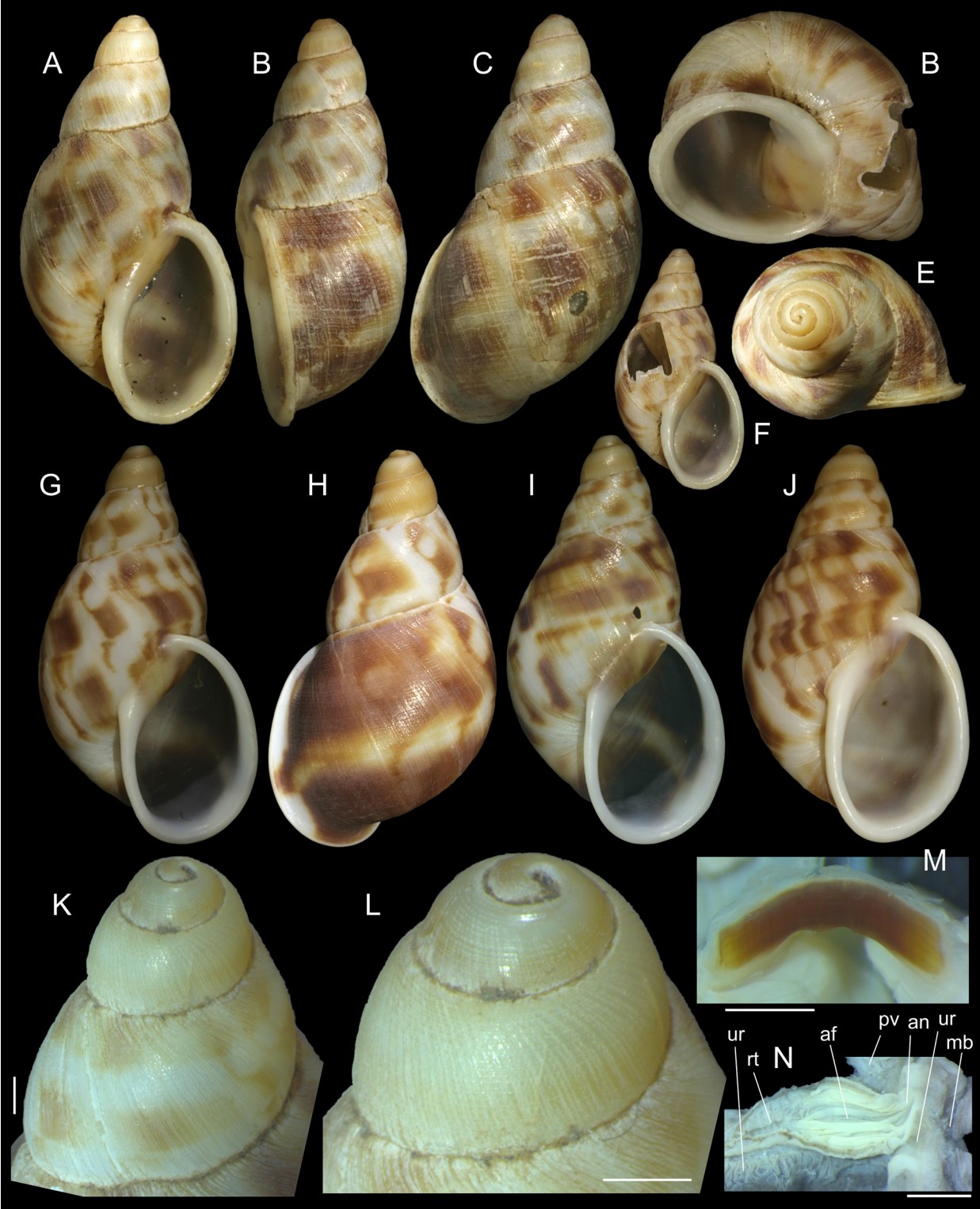

**Fig 5. *Anthinus synchondrus*, shell of types and some anatomical photos.** (A-F) holotype MZSP 152074 shell (L 36.9 mm). (A) frontal view. (B) right view. (C) dorsal view. (D) anterior-slightly ventral view. (E) apical view (W 19.7 mm). (F) frontal view after extraction of specimen by an artificial hole. (G-H) paratype MZSP 152236, (L 40.8 mm), frontal and dorsal views. (I-J) 2 paratypes MZSP 152252 shell, frontal views (L 36.4, 36.2 mm). (K) holotype, detail of protoconch and part of first teleoconch whorl, profile. (L) same, detail of apical region. (M) jaw of holotype in situ, ventral view. (N) detail of right-anterior corner of pallial roof of holotype, ventral view, distal portion of rectum sectioned longitudinally. Scales = 1 mm.

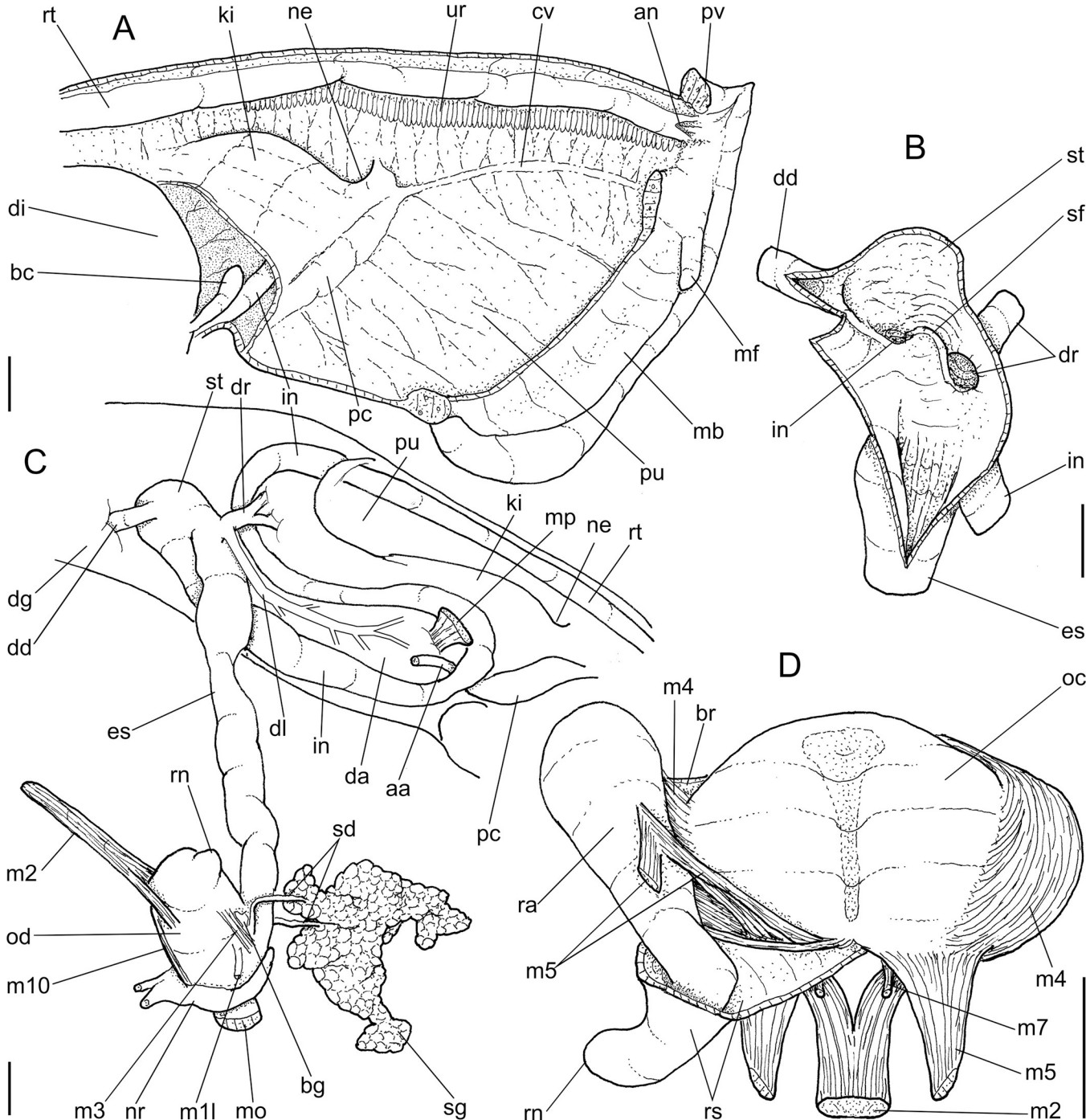

**Fig 6. *Anthinus synchondrus*, anatomy.** (A) pallial cavity roof, ventral view, pneumostome ventral lip sectioned and deflected upwards, portion of pallial floor also shown. (B) stomach opened longitudinally, ventral view. (C) digestive tubes mostly as in situ, ventral view, topology of some adjacent structures also shown. (D), odontophore, dorsal view, superficial layer of muscles and structures removed, left muscles deflected, radula deflected to left. Scales = 2 mm.

Anus directly opening externally (ventrally), preceded by right longitudinal folds (Fig 5N: af).
Urinary gutter (ur) slightly broader, with slightly taller perpendicular folds.
  **Visceral mass.** Similar features as preceding species.

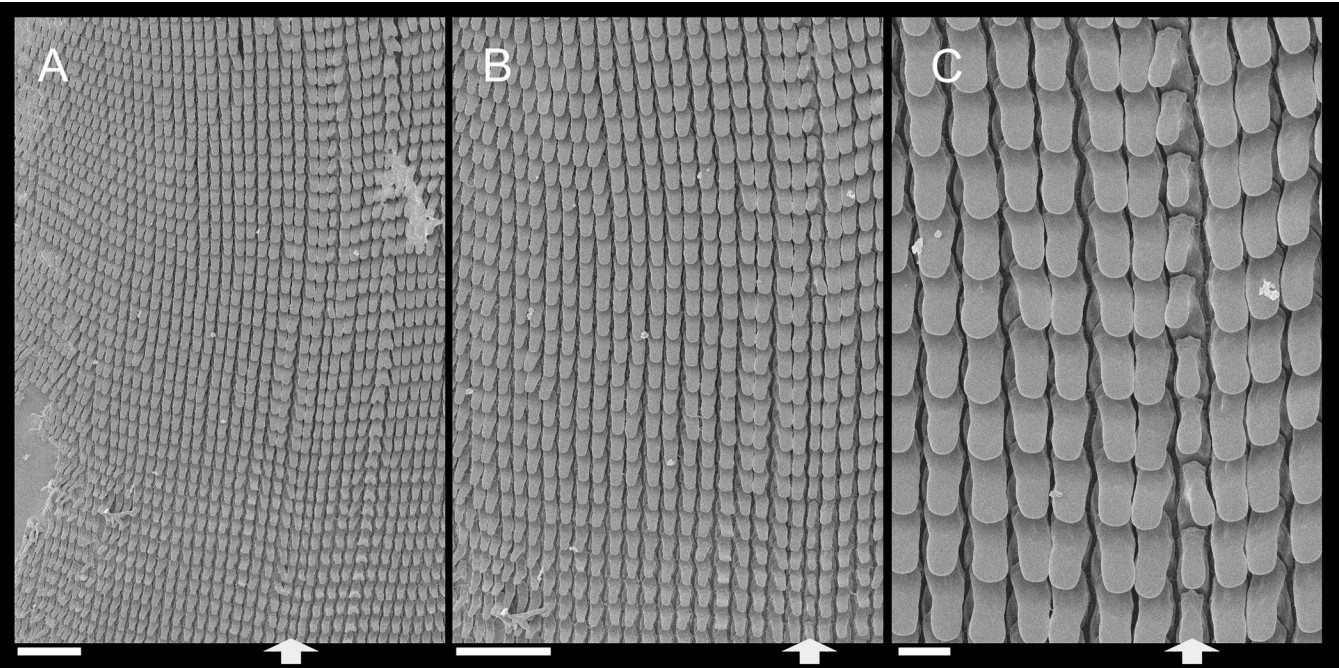

**Fig 7. *Anthinus synchondrus*, radula in SEM, holotype MZSP 152074.** (A) wider view, scale = 200 μm; (B) detail of central region, scale = 200 μm; (C) same, higher magnification, scale = 50 μm. White arrows indicating rachidian column.

**Circulatory and excretory systems.** (Fig 6A) Most features similar to *A. multicolor*; remarks following. Pericardium (pc) lightly more elongated, ~3-times as long as wide. Kidney (ki) tissue clearly U-shaped (convexity left). Nephropore (ne) also protected by curved anterior projection of kidney. Urinary gutter (ur) slightly broader, mainly along its middle portion. Narrow urinary gutter surrounding left edge of anus (an) also present.

**Digestive system.** (Figs 6C–8A) General features similar to those of *A. multicolor*, distinctions and interesting data following. Jaw slender, with uniform width along its length (Figs 5M and 8A: jw), weak central beak; transverse ribs very weak, almost imperceptible; commarginal sculpture wanting. Buccal mass and odontophore (Fig 6D) lacking m1a, m1v; **m3**, only detected in latero-dorsal region (Figs 6C and 8A: m3); **m2**, slightly thinner, insertion narrow on m4 base (Fig 6D); **m4**, very thick, relatively short; **m5**, thin and elongated, origin about half on m4 and half on posterior region of cartilages; **m7**, small narrow pair, origin in posterior end of cartilages, running inside radular sac, splaying along its inner surface. Pair of odontophore cartilages entirely fused with each other along their inner-ventral edge (Fig 6D: oc). **Radula** (Fig 7) with same general features of preceding species; with rachidian teeth, and ~40 pairs of lateral/marginal teeth; no clear distinction between lateral and marginal teeth (Fig 7C); differing in having longer, rather rectangular mesocone, ~3 times longer than wide, covering ~1/3 of posterior tooth (Fig 7B and 7C), mesocone apex rounded; long mesocone reinforced by longitudinal fold in inner concavity, continuous with base; base with similar characters as preceding species. **Rachidian** tooth (Fig 7A–7C: arrow) slightly narrower than lateral teeth, with mesocone with ~half size of neighboring mesocones (Fig 7C). **Lateral teeth** with weak medial inclination, rarely with bifid cutting edge (Fig 7A: right side). **Marginal teeth** starting with no clear boundary with lateral teeth; shaped similarly to lateral teeth, except for being weakly smaller and narrower, row slightly more inclined (Fig 7A: left side) along ~1/3 of each side. Salivary glands apertures as long longitudinal slits (Fig 8A: sa), located in posterior region

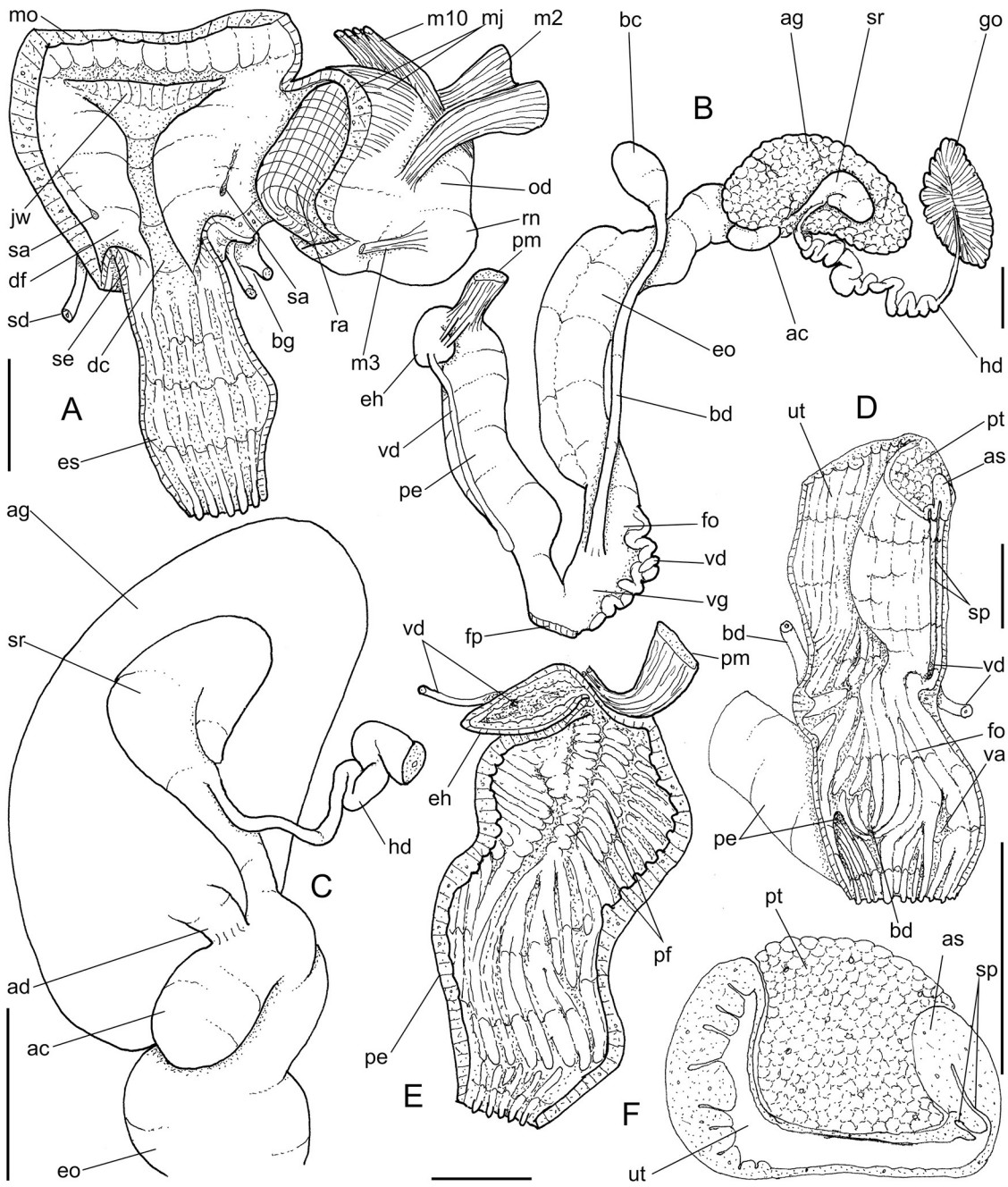

**Fig 8. *Anthinus synchondrus*, anatomy.** (A) foregut opened longitudinally, ventral view, odontophore deflected to right. (B) genital system, mostly dorsal view. (C) same, detail of carrefour, ventral view, seen if albumen gland was transparent. (D) genital system, anterior end opened longitudinally, penis only partially shown. (E) penis and epiphallus opened longitudinally. (F) spermoviduct, transverse section in its middle level. Scales = 2 mm.

of lateral edges of wide dorsal folds (df). Esophagus broader all along its length (Fig 6C: es). Stomach (st) ~50% larger, walls with considerable muscular walls; stomach inner surface (Fig 6B) with fold (sf) running between anterior duct to digestive gland (dr) and intestine (in) more developed. Esophageal-anterior duct to digestive gland also bifid (Fig 6C), left duct (dl)

almost as wide as right duct (dr), much longer. Anus turned externally (Fig 6A: an); inner surface also with 8–10 simple, longitudinal, subterminal, tall folds (Fig 5N: af).

**Reproductive system.** (Fig 8B–8F) General characters similar to those described for *A. multicolor*, distinctions and remarks following. Hermaphroditic duct (Fig 8B: hd) with broader coiled portions along its middle and distal thirds, with fewer but wider coils; insertion lacking curve (Fig 8C). Seminal receptacle (Fig 8B and 8C: sr) large, strongly and tightly curved (convexity left), forming distal blunt beak disposed pointing to hermaphroditic duct insertion. Spermoviduct (Fig 8B and 8C: eo) with glandular walls in uterus only in ventral side (Fig 8F: ut), remaining walls relatively thin-glandular. Presence of accessory genital gland (Fig 8D and 8F: as) lying along spermoviduct, flanking masculine furrows (sp), bulging along prostate (pt) side. Pair of masculine furrows narrow, one lying along accessory genital gland, and other along its medial edge, ~half deep (Fig 8D and 8F: sp). Vas deferens (vd) originating in simple junction of masculine furrows (Fig 8D: vd), no vaginal fold. Vas deferens relatively broad, intensely coiled in its portion on free oviduct and vagina (Fig 8B: vd). Penis slightly longer than half of spermoviduct length (Fig 8B: pe); penis muscle inserting terminally, short, broad (Fig 8B and 8E: pm), with pair of narrow branches connected to epiphallus. Epiphallus (eh) ~1/5 penis' length; vas deferens inserted subterminally on middle level of epiphallus wall (Fig 8E: vd). Epiphallus inner surface as thick glandular, uniform layer of papillae. Internal penial surface with single chamber (Fig 8E); arrangement of inner folds as in Fig 8E, with 8–10 longitudinal wide folds, almost no interspace, in basal half; these folds gradually coalescent in middle penial region, additionally appearing narrower transverse folds in distal region; these folds forming two aligned bulged portions in penis side, producing pair of longitudinal cancelled wide folds (Fig 8E: pf), separated by narrow furrow continuous with epiphallus aperture.

**Central nervous system.** Nerve ring located across buccal mass (Fig 6C: nr). Pair of cerebral ganglia (ce) almost fused; cerebral commissure slightly narrower than ganglia; each ganglion about as wide as adjacent esophageal section; several wide nerves originating in cerebral antero-lateral region. Cerebral node or gland (cd) located in postero-medial quadrant, with ~1/8 each ganglion's size. On each side, parallel, rather narrow connectives (cn) running between cerebral ganglion and ventrally located fused pedal and pleural ganglia (pp), accompanied by blood vessels (bv) issuing from anteriorly-directed aorta. At least six pairs of nerves originating from anterior side of pedal-visceral ganglion complex. Pair of statocysts not seen.

**Distribution.** So far known in region of Unaí, Minas Gerais, Brazil.

**Habitat.** Cerrado biome, semi-dry forest, 628–750 m altitude.

**Measurements.** (length and width in mm) Holotype MZSP 152074 (Fig 5A–5F): 36.9 by 19.7; paratype MZSP 152252 (Fig 5I and 5J): 36.4 by 19.6.

**Material examined.** Types (reported above).

**Taxonomic remarks.** see Discussion item.

*Anthinus vailanti* **new species** (Figs 9–12).

**ZooBank:** urn:lsid:zoobank.org:act:5CF7693D-4C22-4E66-9FCA-B0B821981E0B.

**Types.** Holotype MZSP 154382. Paratypes: MZSP 152118, 3 spm, MZSP 152244, 2 shells, MZSP 152119, all from type locality.

**Type locality.** BRAZIL. **Minas Gerais**; North of Brasilândia de Minas, 16˚59'29"S 46˚00'29"W, 630–635 m altitude [Weslley Vailant de Matos col, iv.2020].

**Etymology.** The specific epithet is in honor of Weslley Vailant de Mattos, from Alcobaça, Bahia, the field collector responsible for most of the new discoveries.

**Diagnosis.** Shell about 30 mm; color mostly monochromatic; spire angle ~50˚, protoconch of 3 whorls; peristome ~55% of shell length and ~60% of width; width ~60 of length. Mantle edge with pointed fold. Reno-pericardial area medium-sized. Pre-rectal pallial muscle present. Odontophore with cartilages ~90% fused with each other; pair m4 ~80% originated from

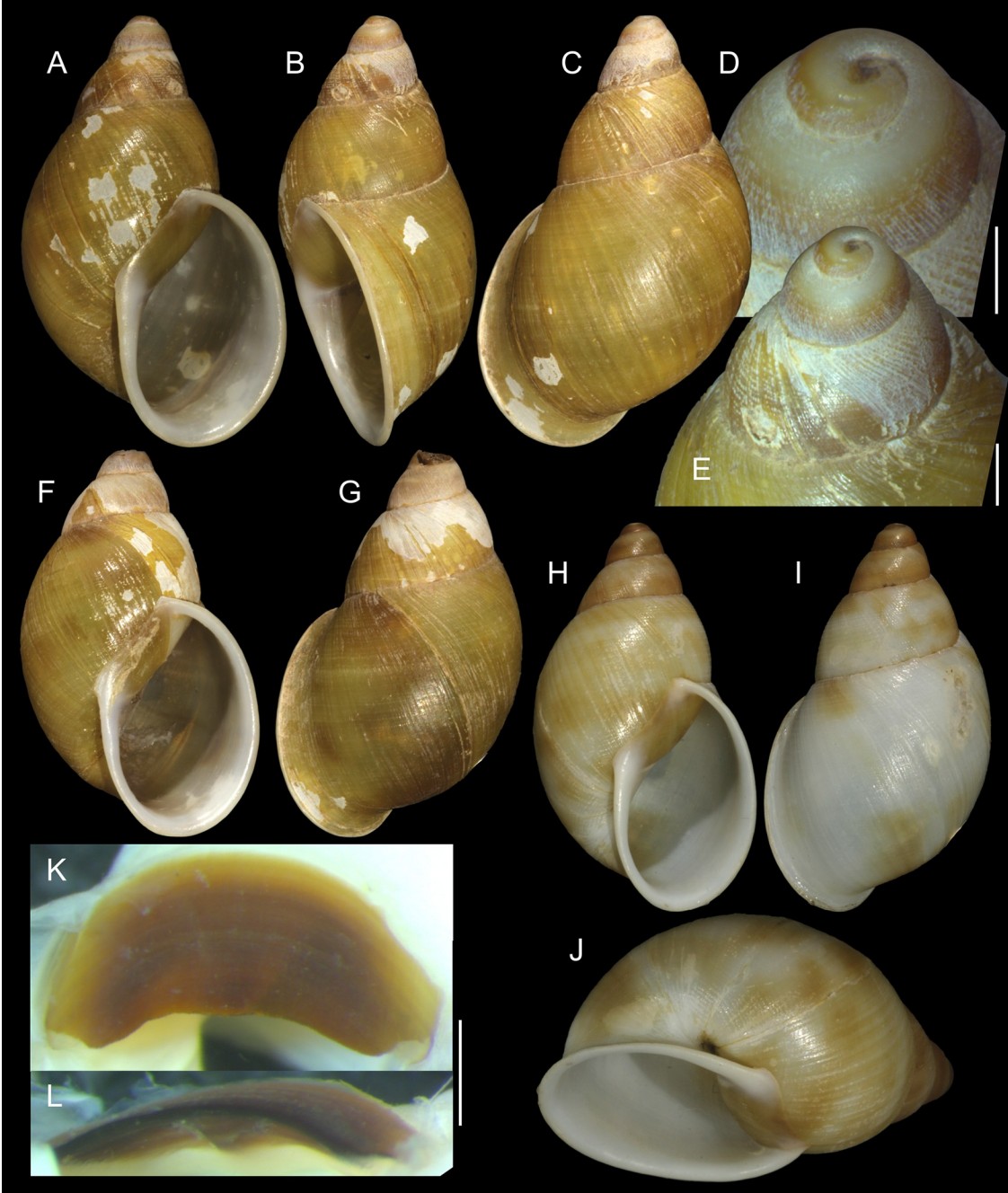

**Fig 9. *Anthinus vailanti*, shells and jaw.** (A-E) holotype MZSP 152891 (L 29.0 mm). (A) frontal view. (B) right view. (C) dorsal view. (D) detail of apex, right-slightly apical view. (E) protoconch and adjacent teleoconch whorl, right-slightly apical view. (F-G) paratype MZSP 152188 (L 27.9 mm) frontal and dorsal views. (H-J) paratype MZSP 152244 (L 29.5). (H) frontal view. (I) dorsal view. (J) anterior-slightly left view showing umbilicus. (K) holotype's jaw plate, ventral view. (L) same, anterior view. Scales = 0.5 mm.

cartilage, remaining on m4. Jaw plate broad, with central fold. Radula with elongated, spatula-like mesocone, rachidial slightly different from neighboring teeth. Seminal receptacle widely curved, hermaphrodite duct inserting in its middle region, perpendicularly. Accessory genital gland in spermoviduct large; with single, coiled sperm-groove. Penis ~60% of spermoviduct;

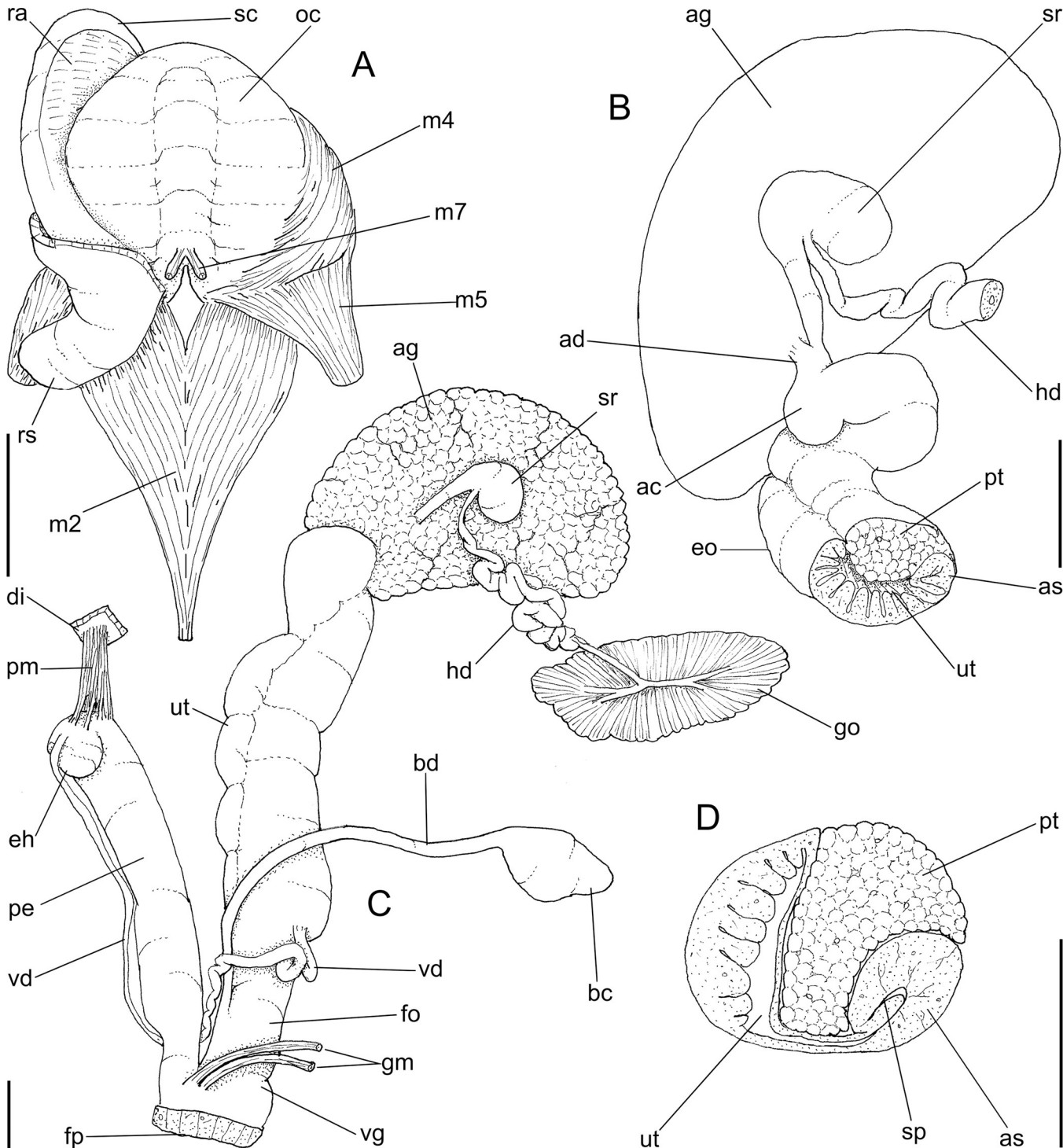

**Fig 10.** ***Anthinus vailanti*, anatomy.** (A) odontophore, dorsal view, superficial layer of muscles and structures removed, most muscles deflected, radular ribbon deflected to left. (B) middle region of genital system isolated, only topology of albumen gland shown. (C) genital system, mostly ventral view. (D) spermoviduct, transverse section in its middle level. Scales = 2 mm.

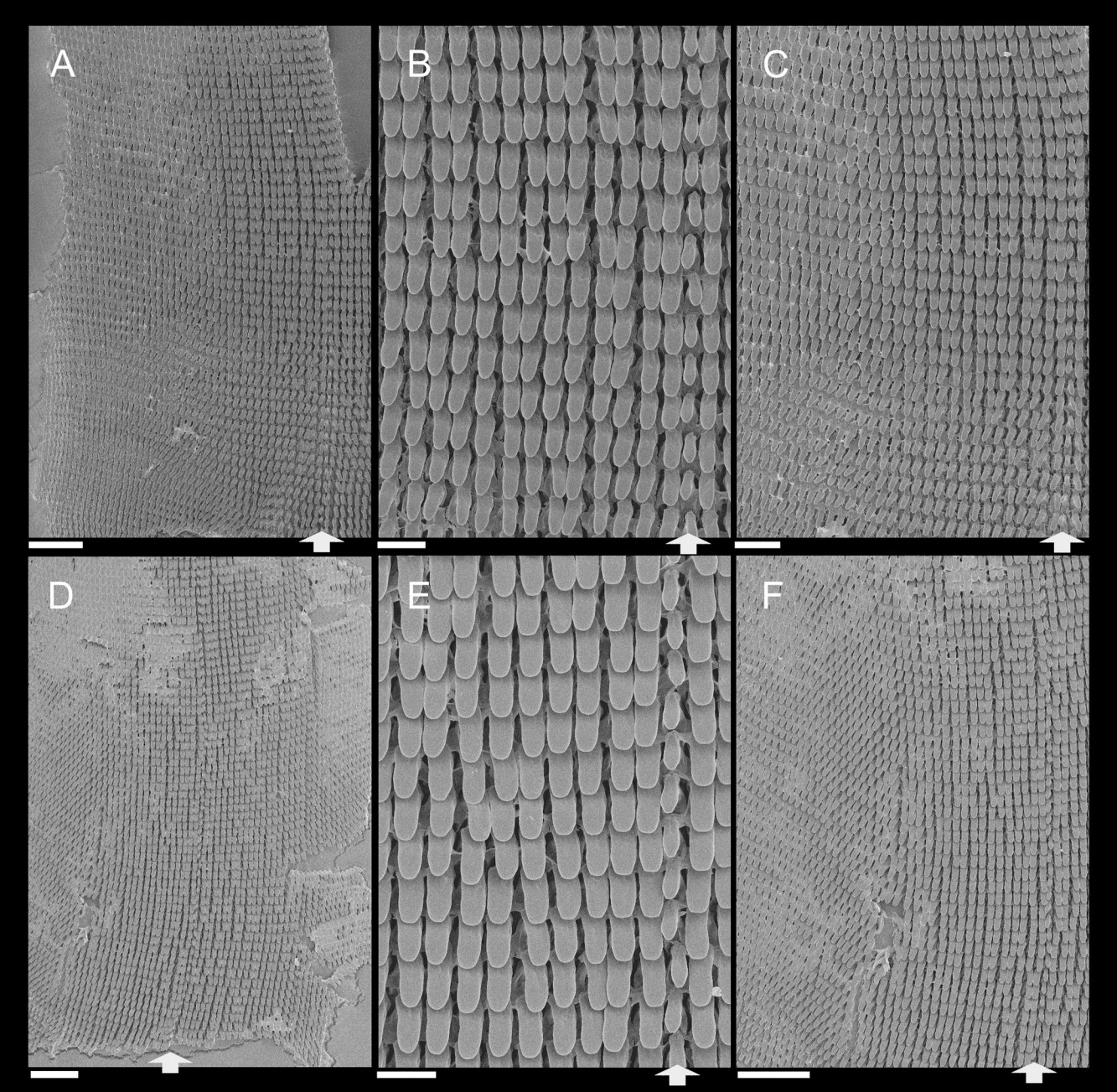

**Fig 11.** ***Anthinus vailanti*, radulae in SEM, Paratypes MZSP 152118, 2 specimens.** (A) wider vision, scale = 200 μm; (B) detail of central region, scale = 50 μm; (C) wide left view, scale = 100 μm; (D) wide view of another specimen, scale = 200 μm; (E) detail of central region, scale = 50 μm; (F) wide left view, scale = 200 μm. White arrows indicating rachidian column.

penis muscle with pair of small branches in epiphallus; epiphallus papillae of medium size, not-uniform, extending to penial chamber; pair of larger folds simple, fused with each other proximally.

**Distinctive description. Shell.** (Fig 9A–9J) Adult shell around 30 mm, conical-oval; apex blunt; wider at middle of last whorl; width ~60% of shell length. Walls thin. Dorso-ventrally weakly flattened (Fig 9J). Color uniform reddish brown in ~half of specimens (Fig 9A–9G); to

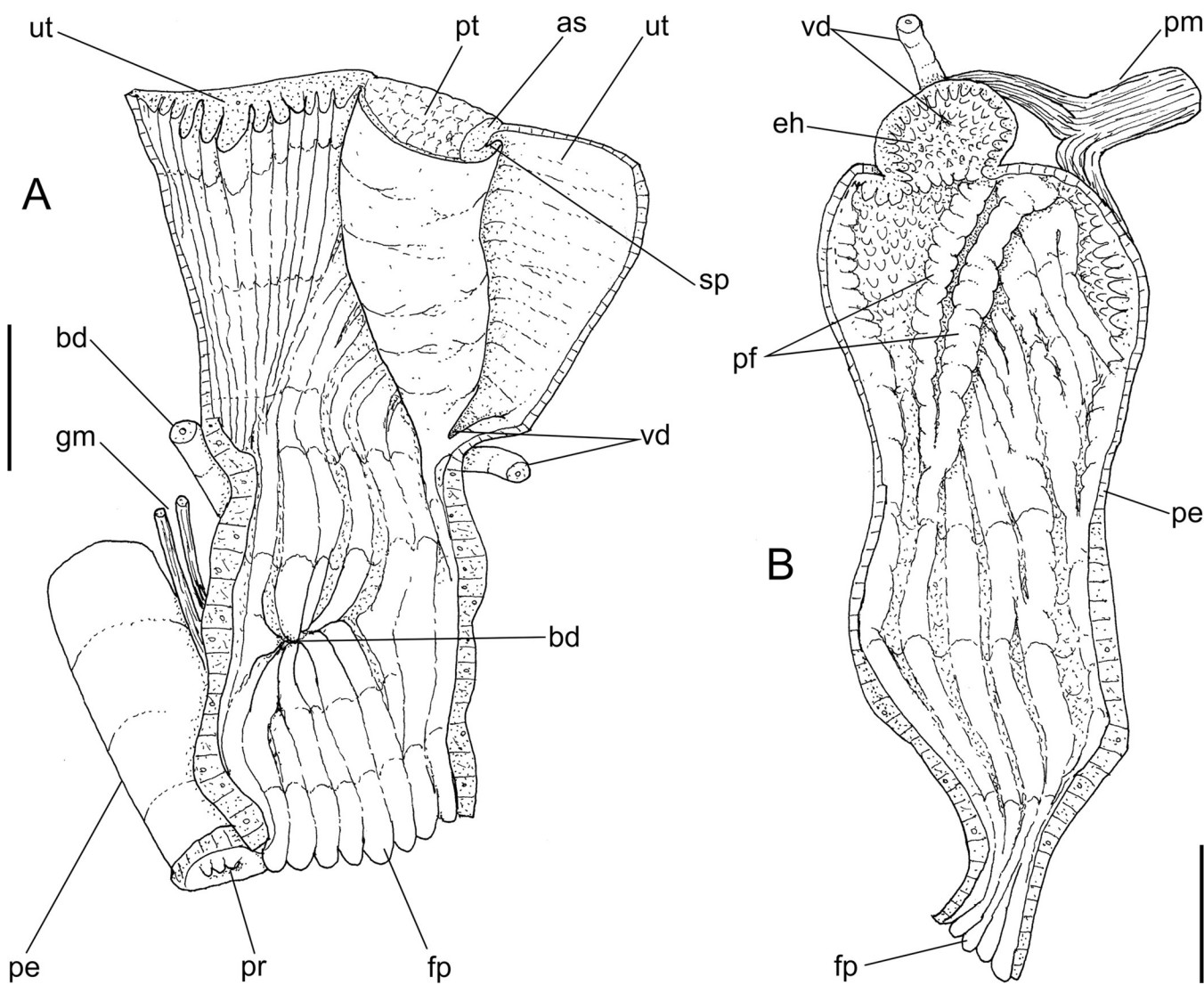

**Fig 12. *Anthinus vailanti*, anatomy.** (A) genital system, detail of anterior region opened longitudinally, penis and other adjacent structures only partially shown. (B) penis and epiphallus opened longitudinally. Scales = 2 mm.

sparse squared pale brown spots on uniform cream base in other half of specimens (Fig 9H–9J), spots less dense than preceding species, but with similar remaining features, with areas denser (Fig 9J) and areas almost lacking spots (Fig 9I). Periostracum heavy, deciduous in some areas, mainly in apex (Fig 9A–9G). First spire whorls white in superior half, gradually becoming pale brown in inferior half (Fig 9D and 9E). Spire angle ~50˚. Protoconch of 3.0 whorls (Fig 9B and 9E), ~20% of length, width of ~8.8 mm, slightly taller than wide; first whorl smooth, gradually axial and spiral narrow cords appearing (Fig 9D and 9E), initially both predominating, gradually axial cords slightly stronger, from suture to suture, interspaced slightly broader than each cord, ~6 per mm, last protoconch whorl with spiral sculpture more spaced, producing rectangular spaced with axial lines; transition to teleoconch clear, slightly prosocline (Fig 9E). Teleoconch sculpture similar to that of protoconch last whorl, as axial, uniform, complete cords and growth lines, gradually small punctuations appearing on axial lines before penultimate whorl, aligned in order to produce spiral punctuated lines, ~20 in penultimate

whorl (Fig 9E), lying next to peristome (Fig 9C, 9F and 9J); of ~2.2 whorls. Whorls profile weakly convex (Fig 9A–9J). Suture well-marked, weakly channeled (Fig 9I). Aperture slightly prosocline (~20˚ from longitudinal general axis) (Fig 9B), elliptic; ~55% of shell length, ~60% of shell width. Peristome reflected, thick, color white (Fig 9A, 9F and 9H); outer lip arched, lacking middle tooth; inner lip with inferior half weakly concave, superior half as narrow, weak callus with ~2/3 of peristome length (Fig 9A, 9F and 9H), almost straight. Inferior half of inner lip covering umbilicus. Umbilicus relatively wide (Fig 9J). Body whorl ~80% of shell length; usually uniform with spire (Fig 9F–9J) to slightly broader (Fig 9A–9C).

**Head-foot.** Similar features as *A. multicolor*, except for more developed pedal glands, bulging in haemocoel floor.

**Mantle organs.** Most characters similar to *A. multicolor*. Except for urinary gutter ~5% broader,

**Visceral mass.** Similar features as preceding species.

**Circulatory and excretory systems.** Most features similar to *A. multicolor*; except for renal anterior fold proportionally smaller, ~half size. Renal glandular lobe also U-shaped, but with ventral branch with double thickness of dorsal branch.

**Digestive system.** General features similar to those of *A. multicolor*. Remarks following. Jaw plate (Fig 9K and 9L) much broader and thicker; cutting edge blunt, with median transverse reinforcement (Fig 9L) in its ventral side. Buccal mass possessing m1v pair. Pair of salivary ducts slightly broader. Odontophore (Fig 10A) differing in having **m5** pair only ~20% originating from cartilages, remaining originating on m4; **m7** pair originated in dorsal-inner surface of posterior region of cartilages; odontophore cartilages (oc) only ~90% fused, with small posterior remaining notch. **Radula** (Fig 11) similar to that of *A. synchondrus*, with ~50 pairs of lateral/marginal teeth (Fig 11A and 11D); distinctions following. **Rachidian** tooth (Fig 11B and 11E: arrow) further smaller and narrower; mesocone slightly claviform. **Lateral teeth** not inclined, pointing posteriorly only; mesocone slightly more elongated, covering ~1/3 of neighboring posterior tooth. **Marginal teeth** inclined outside, located in more arched row (Fig 11C and 11D: left side). Anus subterminal folds taller and stronger, 4–5 in number only; occupying ~10% of rectum length.

**Reproductive system.** (Fig 10B–10D, 12A and 12B) General characters similar to those described for *A. multicolor*, distinctions and remarks following. Hermaphroditic duct (Fig 10C: hd) also broad; coiled portions occupying almost its entire length, except for short stretch at both ends; insertion lacking curve (Fig 10B). Seminal receptacle (Fig 10B and 10C: sr) relatively small, strongly and widely curved (convexity left), insertion of hermaphroditic duct in middle of concavity. Spermoviduct (Fig 10B and 10C: eo) with narrower prostate (pt); glandular walls in uterus only along latero-ventral side (Fig 10D and 12A: ut), remaining ~1/3 of uterine inner surface relatively thin-glandular. Accessory genital gland (Figs 10B, 10D and 12A: as) also present and large. Only single detectable masculine furrow lying along accessory genital gland (Figs 10D and 12A: sp). Vas deferens originating in simple aperture in end of uterine level (Fig 12A: vd), also lacking vaginal fold. Vas deferens also coiled in its portion on feminine tubes, but less intensely (Fig 10C: vd). Presence of pair of genital muscles (Fig 10C: gm) in ventral side of penis base. Penis slightly longer than half of spermoviduct length (Fig 10C: pe); penis muscle also with pair of narrow branches connected to epiphallus (Figs 10C and 12B: pm). Epiphallus (eh) ~1/6 penis' length; vas deferens also inserted subterminally on middle level of epiphallus wall (Fig 12B: vd). Epiphallus inner surface with high layer of papillae (Fig 12B: eh). Internal penial arrangement of folds similar, but with pair of more developed (main) folds (Fig 12B: pf) narrower and slightly taller; papillae similar to those inside epiphallus also lying at side of these main folds (Fig 12B).

**Central nervous system.** With similar attributes as preceding species.

**Distribution.** So far known from type locality.

**Habitat.** Cerrado biome, semi-dry forest, 630–635 m altitude.

**Measurements.** (in mm) Holotype MZSP 154382 (Fig 9A–9E): 29.0 by 17.7; paratypes MZSP 152118 (Fig 9F and 9G): 27.9 by 17.7; MZSP 152244 (Fig 9H–9J): 29.5 by 16.5.

**Material examined.** Types (reported above).

**Taxonomic remarks.** see Discussion item.

*Anthinus morenus* **new species** (Figs 13–15).

**ZooBank:** urn:lsid:zoobank.org:act:A34D18E0-47B6-40C8-A3C8-7FF3367E3AE5.

**Types.** Holotype MZSP 151891. Paratypes: MZSP 152047, 12 spm from type locality (3 with only shells), MZSP 152048, 2 shells. BRAZIL. **Minas Gerais**; Paracatu, Gruta Sapezal, 16˚48'03"S 46˚53'47"W, 700–710 m altitude, MZSP 152195, 3 shells (Weslley Vailant de Matos col., iv.2020).

**Type locality.** BRAZIL. **Minas Gerais**; Paracatu, São José do Sapezal, 16˚48'03"S 46˚53'47"W, 700–710 m altitude [Weslley Vailant de Matos col, iv.2020].

**Etymology.** The specific epithet is in apposition, a Latinization based on the Portuguese word "moreno", meaning dark-brown colored, an allusion to the dark shell color.

**Diagnosis.** Shell about 35 mm; color usual mosaic of brown and beige spots; spire angle ~50˚, protoconch of 3 whorls; peristome ~51% of shell length and ~51% of width; width ~60 of length. Mantle edge with simple, rounded fold. Reno-pericardial area small-sized. Pre-rectal pallial muscle present. Odontophore with cartilages ~50% fused with each other; pair m4 ~40% originated from cartilage, remaining on m4. Jaw plate narrow, lacking central beak. Radula with elongated, spatula-like mesocone, rachidial rather similar to neighboring teeth. Seminal receptacle tightly curved, hermaphrodite duct inserting in its base, perpendicularly. Accessory genital gland in spermoviduct large; with pair of thick-walled sperm-grooves. Penis ~50% of spermoviduct; penis muscle mostly inserted to epiphallus; epiphallus papillae tall, restricted to it; single, large inner fold.

**Description. Shell.** (Fig 13A–13H, 13J and 13K) Adult shell around 35 mm, conical-oval; apex blunt; greatest width on last whorl; width ~60% shell length. Walls relatively thin. Color beige, with three spiral, equidistant brown bands up to penultimate whorl, five in last whorl; mosaic of sparse, apparently random transverse (axial) spots, irregular in size, normally limited to two neighboring spiral bands, but sometimes extending further (Fig 13C and 13K). Spire angle ~50˚. Slightly flattened dorso-ventrally, dorso-ventral axis ~80% of latero-lateral axis (Fig 13E). Protoconch of 3 whorls (Fig 13B, 13F, 13G and 13K), ~20% of length, width of ~7 mm; first whorl smooth, gradually axial narrow cords appearing (Fig 13G and 13H), from suture to suture, interspaced similar to each cord, ~10 per mm; transition to teleoconch clear, slightly prosocline (Fig 13B, 13F, 13G and 13K). Teleoconch of ~2.5 whorls; sculpture similar to that of protoconch last whorl in its first whorl, as axial, uniform, complete cords and growth lines (Fig 13F); after first teleoconch whorl axial cords gradually disappearing, being substituted by rather irregular axial undulations and minute spiral aligned punctuations (Fig 13C, 13F and 13J), ~25 spiral lines in penultimate whorl, interval between lines equivalent to their width up to double of it; spiral punctuated lines relatively uniformly covering surface up to peristome (Fig 13B, 13C and 13K). Whorls profile slightly convex. Suture well-marked, slightly channeled (Fig 13B, 13F and 13J). Aperture slightly prosocline (~7˚ from longitudinal general axis) (Fig 13B), elliptic; ~51% of shell length, ~51% of shell width. Peristome reflected, thick, pure white (Fig 13A and 13B) to pale beige (Fig 13J); outer lip arched, lacking tooth (Fig 13A and 13J); inner lip uniformly arched, concave; callus narrow, in superior third (Fig 13A and 13J). Inferior 2/3 of inner lip partially covering umbilicus. Umbilicus relatively narrow (Fig 13D). Body whorl ~84 shell length; usually looking wider than if spire had uniform growth (Fig 13C and 13K).

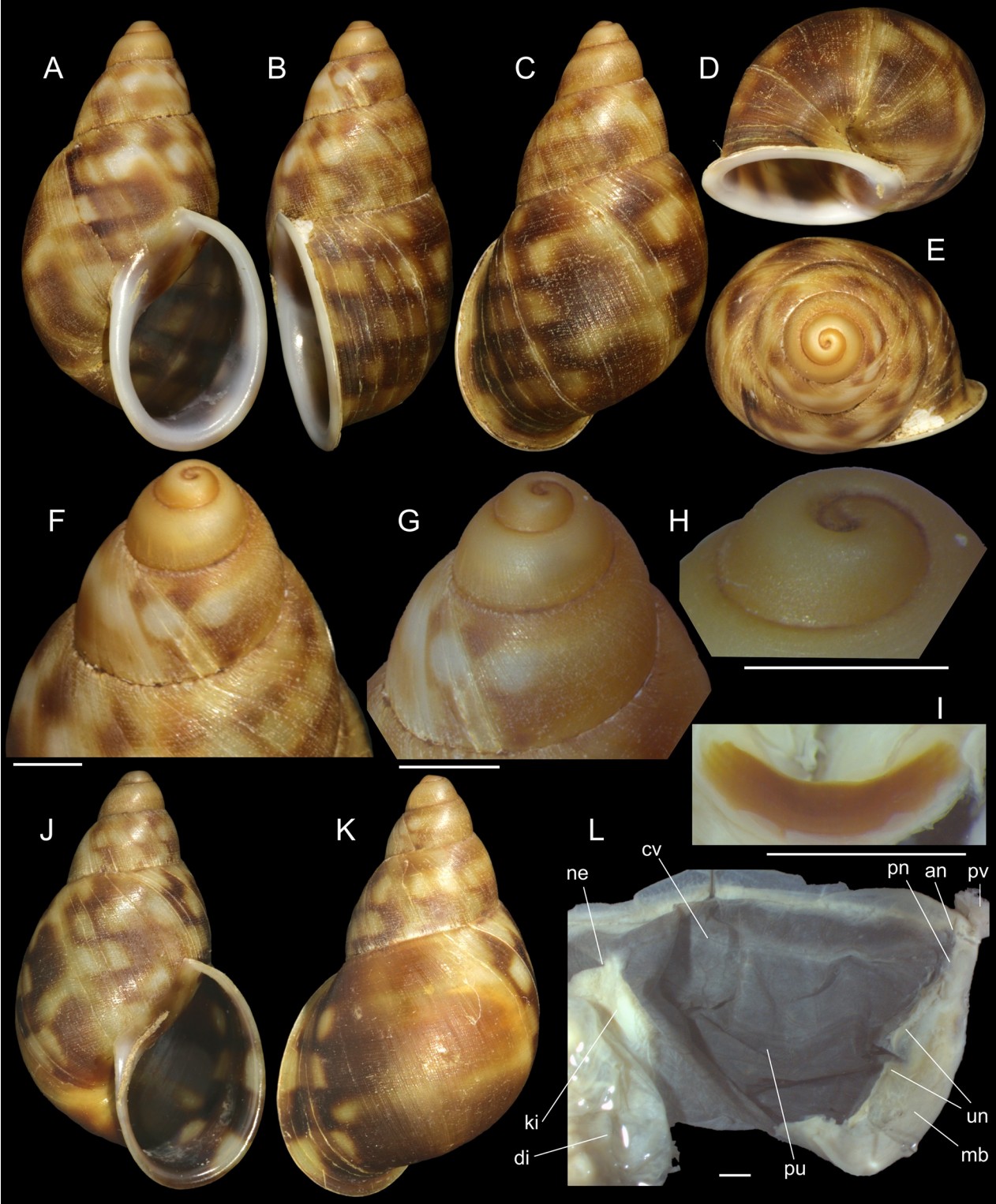

**Fig 13. *Anthinus morenus*, shell of some types and anatomy.** (A-H) Holotype MZSP 152891 (L 35.8 mm). (A) Frontal view. (B) Right view. (C) Dorsal view. (D) Anterior view (W 21.0 mm). (E) Apical view. (F) Detail of protoconch and fist teleoconch whorl, profile. (G) Same, higher magnification. (H) Same, detail of apex. (I-L) Paratype MZSP 152047#1. (I) Jaw, in situ, ventral view. (J) Shell, frontal view (L 34.8 mm). (K) Shell, dorsal view. (L) Pallial cavity, ventral view, dorsal part of mantle edge and deflected to right, part of mantle floor also shown. Scales = 2 mm.

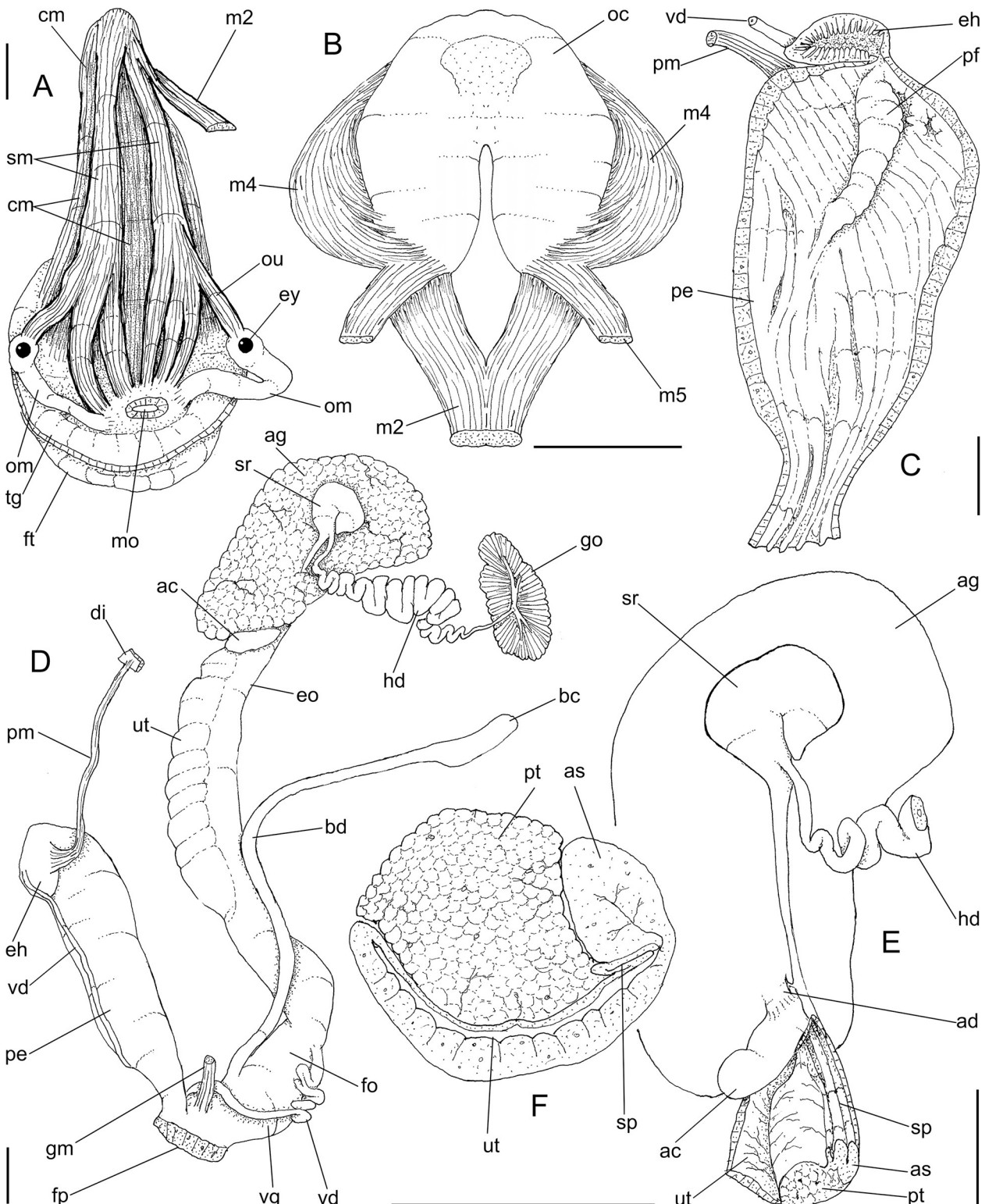

**Fig 14. *Anthinus morenus*, anatomy.** (A) Head-foot, head, digestive and genital structures removed, dorsal integument removed, main concern to main head-foot musculature. (B) Odontophore, dorsal view, superficial layer of muscles and structures removed, most muscles deflected. (C) Penis, sectioned longitudinally to show inner surface. (D) Genital system, mostly ventral view. (E) Middle region of genital system isolated, only topology of albumen gland shown. (F) Spermoviduct, transverse section in its middle level. Scales = 2 mm.

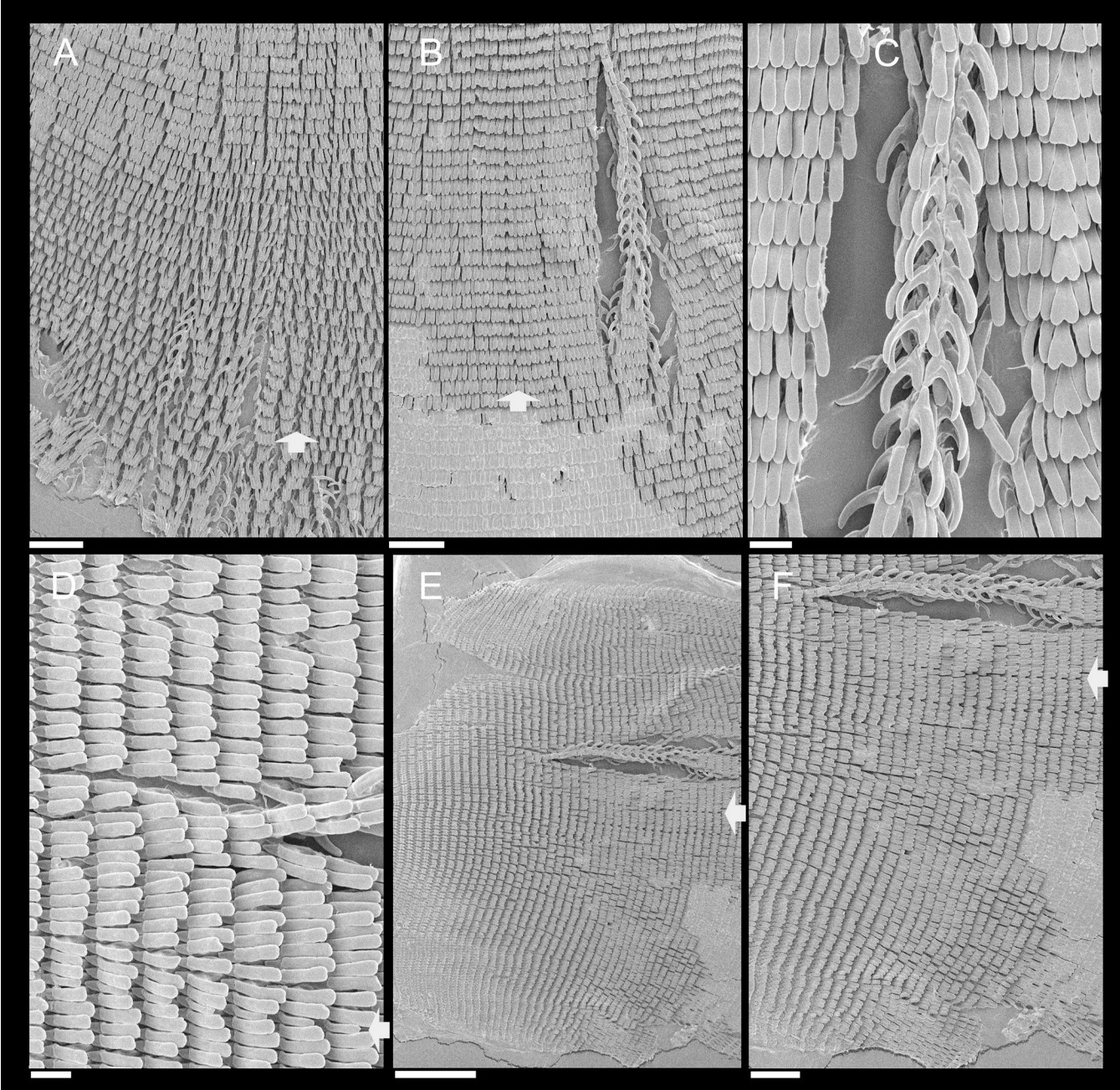

**Fig 15.** *Anthinus morenus*, **Radula in SEM, paratype MZSP 152047.** (A) wide view, scale = 200 μm; (B) detail of central region, scale = 200 μm; (C) detail of damaged region, with teeth in profile, scale = 50 μm; (D) detail of right region, scale = 50 μm; (E) wide view, scale = 500 μm; (F) detail of left region, scale = 200 μm. White arrows indicating rachidian column.

**Head-foot.** (Fig 14A) of normal shape, with characters similar to *A. multicolor*. Differing in pair of secondary cephalic muscles thicker, broader, and separated from each other almost up to their posterior end. Pedal gland not protruding in haemocoel.

**Mantle organs.** (Fig 13L) most features similar to *A. multicolor*, differences following. Pallial roof uniformly pigmented dark (pu). Except for pulmonary vein (cv), all remaining pulmonary vessel not protruded, only visible with light passing through roof. Secondary mantle edge

fold (mf) smaller, not pointed at left. Lung longer, almost 2 whorls. Reno-pericardial structures smaller, occupying ~10% of roof area. Urinary gutter smooth, lacking transverse folds.

**Visceral mass.** of similar attributes of preceding species.

**Circulatory and excretory systems.** (Fig 13L) with same features as preceding species, except in being ~40% smaller.

**Digestive system.** General features similar to those of preceding species. Remarks following. Jaw plate (Fig 13I) narrow, relatively thinner, including its cutting edge. Pair of salivary glands ~30% less developed. Odontophore (Fig 14B) differing in having **m5** pair with narrower origin, ~40% originating from cartilages, remaining originating on m4; **m7** absent; odontophore cartilages (oc) only ~50% fused, with long posterior notch separating both. **Radula** (Fig 15) mostly similar to *A. synchondrus*, with ~60 pairs of lateral/marginal teeth; distinctions and remarks following. Mesocone inner reinforcement clearly shown in Fig 15C (central column, with teeth in profile), running along mesocone, becoming thicker and attaching to base. All teeth narrower and more elongated, ~4–5 times longer than wide (Fig 15B and 15F); some rare column bifid (Fig 15C: right side). **Rachidian** tooth of difficult visualization (Fig 15A, 15B and 15E), ~1/3 narrower and shorter than lateral tooth, and slightly dislocated anteriorly in relation to its row (Fig 15D: arrow). **Lateral teeth** with very elongated mesocone, widely curved posteriorly (Fig 15D). **Marginal teeth** with no clear boundary with lateral teeth, as gradual diminishment in lateral regions (Fig 15E and 15F, inferior region).

**Reproductive system.** (Fig 14C–14F) General characters similar to those described for preceding species, distinctions and remarks following. Hermaphroditic duct (Fig 14D: hd) also broad; coiled portions occupying almost its entire length, except for short stretch at both ends (Fig 14E). Seminal receptacle (Fig 14D and 14E: sr) relatively small, strongly and tightly curved (convexity left), insertion of hermaphroditic duct in receptacle base, lacking curve (Fig 14E). Spermoviduct (Fig 14D and 14E: eo) with glandular walls in uterus along its entire outer side (Fig 14E and 14F: ut). Accessory genital gland (Fig 14E and 14F: as) also present and large. Pair of masculine furrows narrow, one lying along accessory genital gland, and other along its medial edge, ~half deep (Fig 14E and 14F: sp). Vas deferens originating in simple aperture in end of uterine level, also lacking vaginal fold. Vas deferens also coiled in its portion on feminine tubes, but less intensely (Fig 14D: vd). Presence of single genital muscle (Fig 14D: gm) in ventral side of vaginal base. Penis ~half of spermoviduct length (Fig 14D: pe); penis muscle mostly inserted in epiphallus (Fig 14D: pm) with additional small portion connected to penis distal end. Epiphallus (eh) ~1/4 penis' length; vas deferens also inserted subterminally in middle level of epiphallus wall (Fig 14C and 14D: vd). Epiphallus inner surface with high layer of papillae (Fig 14C: eh). Internal penial arrangement of folds similar, but with single more developed (main) fold (Fig 14C: pf) broad and slightly taller; penial wall substantially thick muscular, inner set of folds (Fig 14C) softer and simpler.

**Central nervous system.** With equivalent attributes as preceding species; except having longer cerebral commissure.

**Distribution.** Paracatu region, Minas Gerais.

**Habitat.** Cerrado biome, semi-dry forest, 700–710 m altitude.

**Measurements.** (length and width in mm) Holotype MZSP 152891 (Fig 13A–13H): 35.8 by 21.0; paratype MZSP 152047#1 (Fig 13I–13L): 34.8 by 20.8.

**Material examined.** Types (reported above).

**Taxonomic remarks.** see Discussion item.

***Anthinus savanicus* new species** (Figs 16–19).

**ZooBank:** urn:lsid:zoobank.org:act:7816D208-22FE-4C19-BEE5-ED2DD68B3FAD.

**Types.** Holotype MZSP 154391. Paratypes MZSP 153897, 6 spm, MZSP 152171, 4 shells, all from type locality.

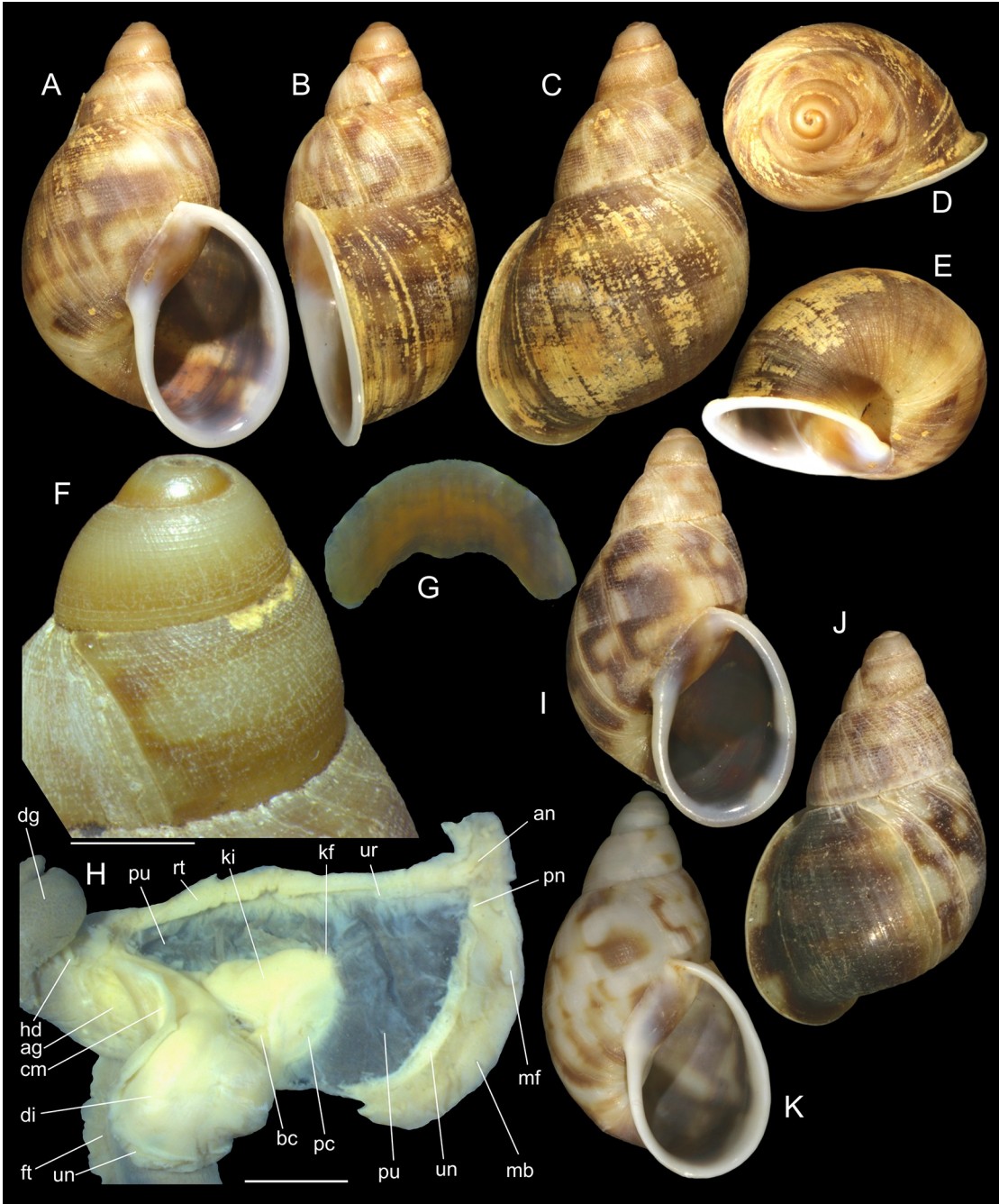

**Fig 16. *Anthinus savanicus*, shell of types and anatomical details.** (A-H) holotype MZSP 154391 (L 32.5 mm). (A) shell, frontal view. (B) right view. (C) dorsal view. (D), apical view. (E) inferior (umbilical) view. (F) protoconch and part of first teleoconch, profile, scale = 2 mm. (G) jaw, ventral view (L 1.7 mm). (H) almost entire specimen, mostly dorsal view, pallial cavity removed and deflected right (ventral view), head-foot deflected below, visceral mass slightly uncoiled, scale = 5 mm. (I-J) paratype MZSP 153897 (L 31.5 mm), frontal and dorsal views. (K) paratype MZSP 152171 (L 35.3 mm), frontal view.

**Type locality.** BRAZIL. **Goiás**; north of Formosa, 15˚21'50"S 47˚27'08"W [Weslley Vailant de Matos col, ix.2012].

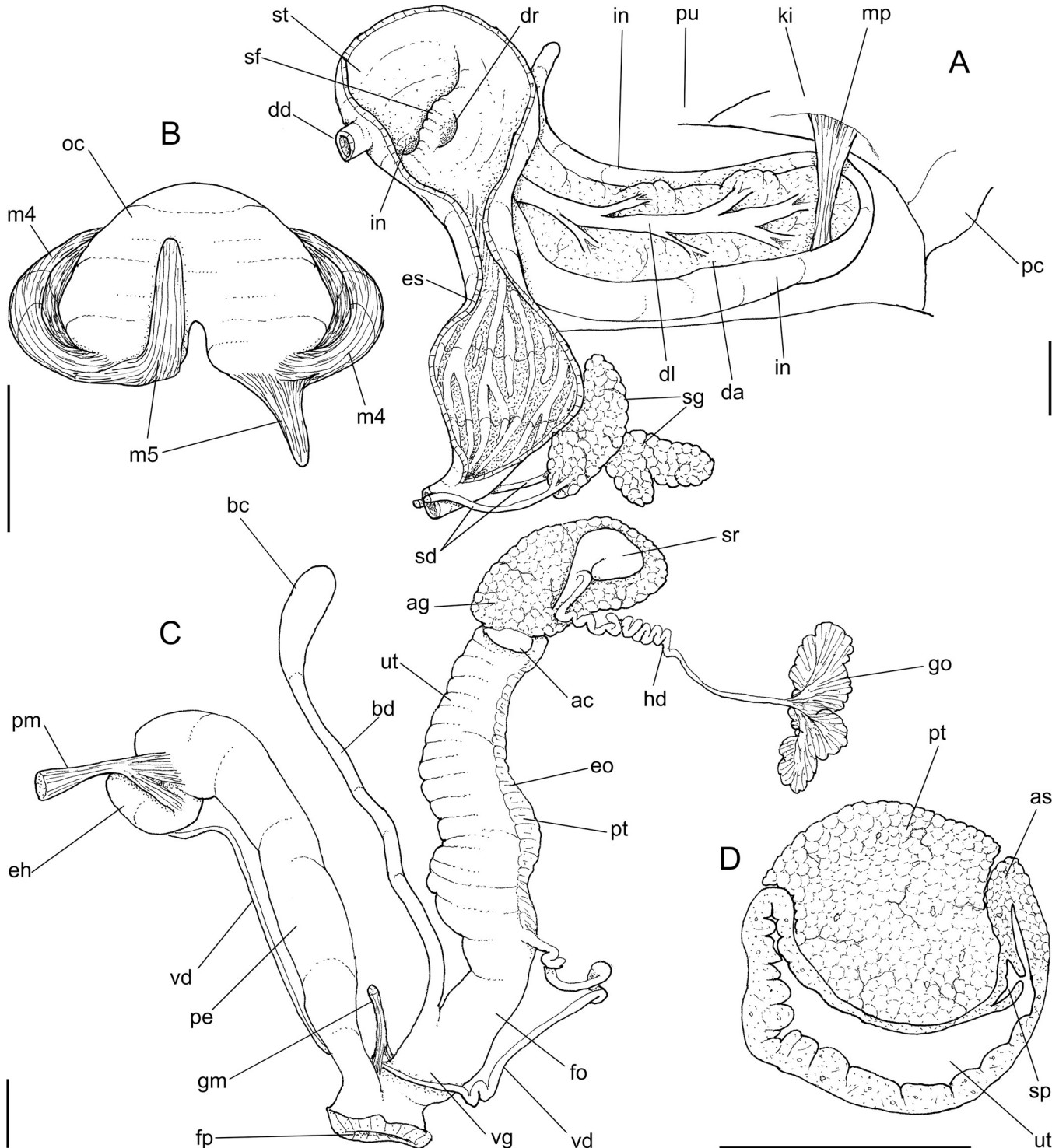

**Fig 17. *Anthinus savanicus*, anatomical drawings.** (A) midgut as in situ, ventral view, some adjacent structures or their topology also shown, stomach and esophagus opened longitudinally. (B) odontophore, dorsal view, outer layer of structures removed, pair of odontophore cartilages deflected, both m4 and left m5 (right in Fig) also deflected. (C) genital structures, mostly ventral view, most vas deferens (dv) detached. (D) spermoviduct, transverse section of its middle level. Scales = 2 mm.

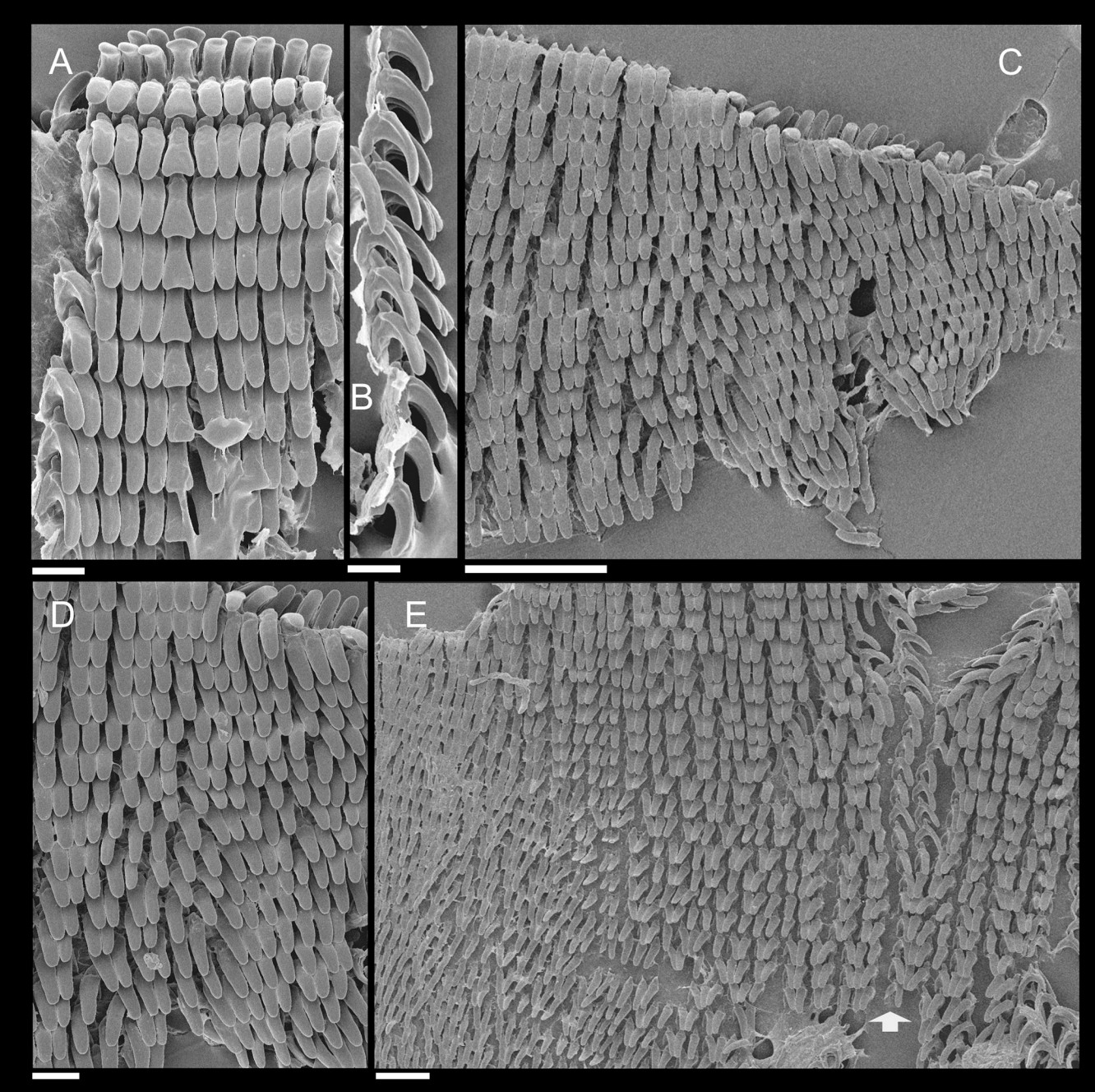

**Fig 18. *Anthinus savanicus*, radulae in SEM.** (A) holotype MZSP 154391. Detail of central region, scale = 50 μm; (B) same, detail of column in profile, scale = 50 μm; (C) same, detail of right region, scale = 200 μm; (D) paratype MZSP 153897, detail of right region, scale = 50 μm; (E) same, central region, scale = 100 μm. White arrows indicating rachidian column.

**Etymology.** The specific epithet is derived from the regional biome, called *Savana Aciden-tada* (rugged savanna).

**Diagnosis.** Shell about 32 mm; color usual mosaic of brown and beige spots; spire angle ~50°, protoconch of 2.8 whorls; peristome ~48% of shell length and ~51% of width; width ~60% of length. Mantle edge with projected fold. Reno-pericardial area large-sized. Pre-rectal

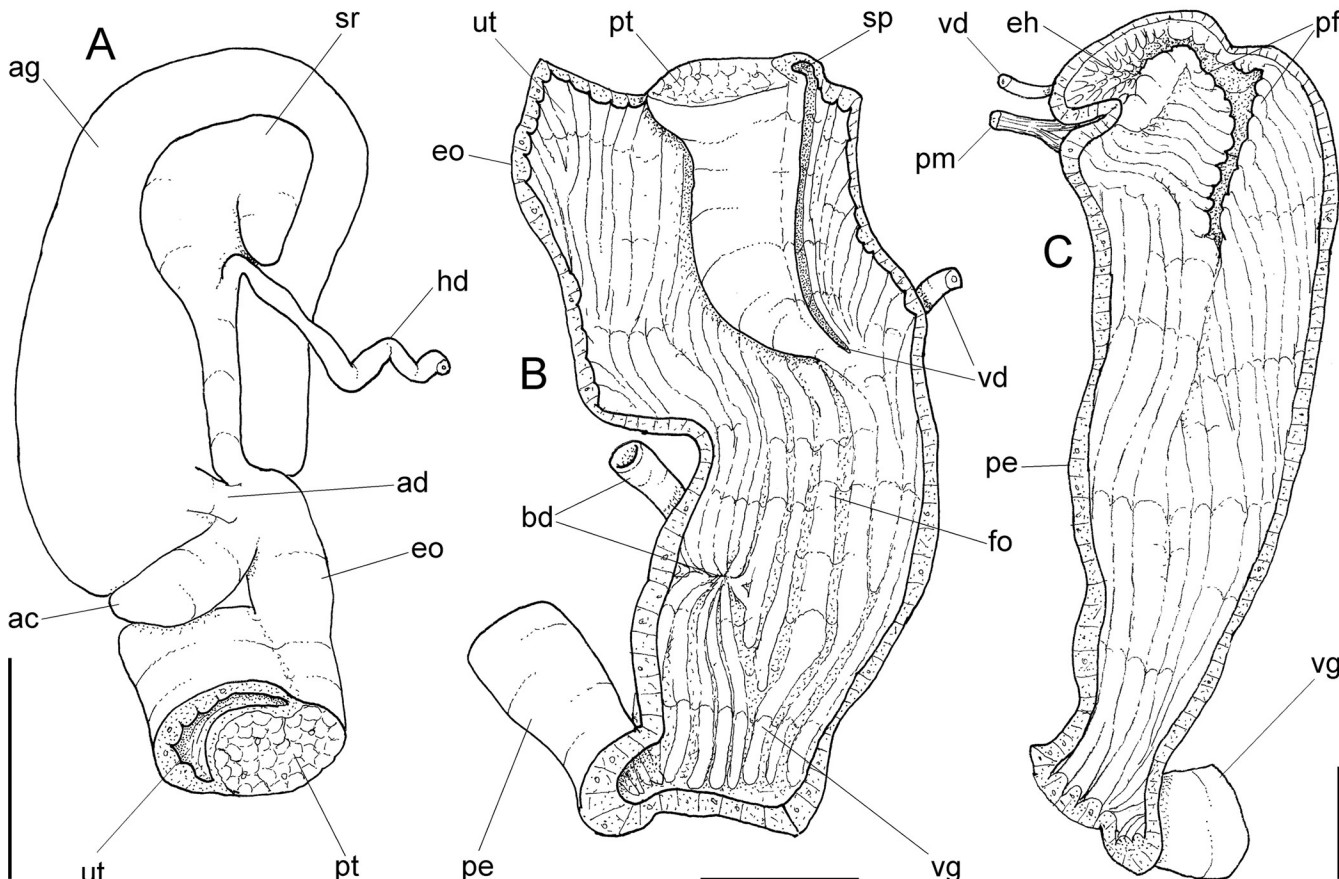

**Fig 19. *Anthinus savanicus*, anatomy, genital system.** (A) carrefour region, ventral view, seen if albumen gland (ag) was transparent, spermoviduct (eo) transversely sectioned in its posterior region; (B) anterior end, ventral view, entirely opened longitudinally, portions of adjacent structures also shown. (C) penis, ventral view, entirely opened longitudinally, portions of adjacent structures also shown. Scales = 2 mm.

pallial muscle present. Odontophore with cartilages ~85% fused with each other; pair m4 ~50% originated from cartilage, remaining on m4. Jaw plate broad, thick. Radula with elongated, spatula-like mesocone, rachidial rather similar to neighboring teeth. Seminal receptacle tightly curved, hermaphrodite duct inserting in its base, curved. Accessory genital gland in spermoviduct small; with pair of sperm-grooves, one being T-shaped. Penis length ~80% of spermoviduct; penis muscle mostly inserted in penis distal end, with thick branch to epiphallus; epiphallus papillae tall, restricted to it; large pair of penial inner folds at end of secondary folds.

**Distinctive description. Shell.** (Fig 16A–16F and 16I–16K) Adult shell around 32 mm, conical-oval; apex blunt; wider at middle of last whorl; width ~60% of shell length. Walls slightly thick. Dorso-ventrally slightly flattened (Fig 16D and 16E). Color light beige and brown squared spots, both predominating; brown spots randomly distributed along 4–5 spiral, wide, rather isometric bands in last whorl, 2–3 spiral bands in penultimate whorl; usual concentration of brown pigment preceding peristome (Fig 16C and 16J). Spire angle ~50˚. Protoconch of 2.8 whorls (Fig 16B, 16D, 16F and 16J), ~19% of length, width of ~7.7 mm, slightly taller than wide; 5–6 narrow, spiral lines present since first whorl, equally spaced, gradually more spiral and axial narrow lines appearing (Fig 16F), initially spiral lines predominating, gradually in last whorl both predominating, gradually axial cords slightly stronger, from suture

to suture, interspaced by similar width as lines', ~7 per mm; transition to teleoconch clear, slightly prosocline (Fig 16B, 16F and 16J). Teleoconch sculpture similar to that of protoconch's last whorl, as axial, uniform, complete cords and growth lines; cords containing punctuations, aligned with their neighboring cords producing spiral punctuated cords (Fig 16A, 16B, 16I and 16J); ~18 lines in penultimate whorl. Suture well-marked, weakly channeled (Fig 16J and 16K). Aperture slightly prosocline (~10˚ from longitudinal general axis) (Fig 16B), elliptical; ~48% of shell length, ~51% of shell width. Peristome reflected, thick, color white (Fig 16A, 16I and 16K); outer lip arched, lacking middle tooth; inner lip with inferior half shallowly concave, superior half as narrow, weak callus with ~2/5 of peristome length (Fig 16A, 16I and 16K), usually convex. Inferior half of inner lip intensely covering umbilicus. Umbilicus relatively wide (Fig 16E). Body whorl ~80% of shell length; uniform with spire (Fig 16C, 16J and 16K).

**Head-foot.** (Fig 16H) Similar features as preceding species.

**Mantle organs.** (Fig 16H) most characters similar to preceding species. Distinctions and remarks following. Pallial cavity ~0.75 of whorl. Mantle edge dorsal fold (mf) with very elongated edge pointed to left. Urinary gutter (ur) slightly narrow, transverse folds low.

**Visceral mass.** Similar features as preceding species.

**Circulatory and excretory systems.** (Fig 16H) Most features similar to preceding species; except in being proportionally larger, occupying ~40% of pallial cavity (pc, ki).

**Digestive system.** (Fig 17A and 17B) General features similar to those of preceding species, distinctions and interesting data following. Jaw strong, well-arched, with central inner reinforcement (Fig 16G), weak central beak; transverse ribs very weak, almost imperceptible; commarginal sculpture wanting. Buccal mass and odontophore (Fig 17B) lacking m1a, m1v and m3d; **m3p**, well-developed, extending along base of esophageal origin; **m4**, very thick, relatively short; **m5**, thin and short, origin about half on m4 and half on posterior region of cartilages; **m7**, absent. Pair of odontophore cartilages ~85% fused, with small posterior medial notch separating both cartilages (Fig 17B: oc). **Radula** (Fig 18) virtually similar to preceding *A. morenus*, with ~50 pairs of lateral teeth; but with all teeth slightly shorter and wider (Fig 18A, 18C and 18D), mesocone reinforcement (Fig 18B) and bifid column (Fig 18A: center) also detected. **Rachidian** tooth (Fig 18E: arrow) similar to lateral teeth, except in being slightly smaller. **Marginal teeth** also turned outside, but proportionally larger, more similar to lateral teeth (Fig 18E: left side). Salivary ducts narrow, elongated (Fig 17A: sd). Esophagus broader all along its length, inner folds longitudinal, rather irregular (Fig 17A: es). Stomach (st) large, walls considerably muscular; stomach inner surface (Fig 17A) lacking developed fold, only having small transverse folds (sf) close to duct to digestive gland (dl). Esophageal-anterior left duct (dl) also broad and long.

**Reproductive system.** (Figs 17C, 17D and 19A–19C) General characters similar to those described for preceding species, distinctions and remarks following. Hermaphroditic duct (Fig 17C: hd) with coiled portions only in distal half, with fewer and simpler coils; insertion with strong curve (Figs 15A and 17C: hd). Seminal receptacle (Figs 17C and 19A: sr) large, strongly and tightly curved (convexity left), forming distal blunt beak pointing to hermaphroditic duct insertion. Spermoviduct (Figs 17C, 19A and 19B: eo) with glandular walls in uterus relatively low and splayed (Fig 17D: ut). Accessory genital gland relatively smaller (Figs 17D and 19B: as), only present in basal 2/3 of spermoviduct, distal third lacking this gland (Fig 19A); masculine furrows composed of pair of separated folds flanking wide groove (Figs 17D and 19B: sp). Vas deferens (vd) originating in simple junction of masculine furrows (Fig 19B: vd), no vaginal fold. Vas deferens relatively broad, coiled only in some sparce portions (Fig 17C: vd); becoming very narrow along penial region. Penis almost as long as spermoviduct length (Fig 17C: pe); penis muscle inserting as pair of relatively same-sized wide branches attached both, to epiphallus and to distal penial end (Fig 17C: pm). Epiphallus (eh) ~1/6 penis' length; vas deferens

inserted subterminally on middle level of epiphallus wall. Epiphallus inner surface as thick glandular, uniform layer of papillae (Fog. 10C: eh. Internal penial surface with single chamber (Fig 19C); arrangement of inner folds as in Fig 19C, with 8–10 longitudinal wide, low folds, almost no interspace, along entire penis length; these folds distally as pair of longitudinal multilobed folds (Fig 19C: pf) restricted to single side, with wide, smooth groove separating them; this groove continuous to epiphallus aperture (Fig 19C).

**Central nervous system.** with equivalent attributes as preceding species, except for cerebral pair of ganglia more closely located, with wider commissure; and for slightly more antero-posteriorly elongated cerebral and pedal pair of ganglia.

**Distribution.** So far known the type locality.

**Habitat.** Cerrado biome.

**Measurements.** (length and width in mm) Holotype MZSP 154391 (Fig 16A–16C): 32.5 by 20.3; Paratype MZSP 153897 (Fig 16I–16L) 31.5 by 18.7; Paratype MZSP 152171 (Fig 16K): 35.3 by 20.1.

**Material examined.** Types (reported above).

**Taxonomic remarks.** see Discussion item.

**Genus *Catracca* new genus.** **ZooBank.** urn:lsid:zoobank.org:act:C9AEAD6E-EA30-4584-A884-AB967AE8CE6A.

**Diagnosis.** shell of relatively thick walls; spire tall, with blunt tip; narrowly umbilicate. Protoconch of three whorls; first whorl smooth; narrow close-placed axial cords uniformly disposed in remaining whorls, from suture to suture. Peristome deflected, callus narrow. Color uniform, lacking spots. Pallial cavity lacking septum, weak vascular network. Kidney with anterior projection protecting nephrostome. Odontophore with pair m5 originated part from pair m4 and part from pair m2. Stomach with walls relatively muscular. Seminal receptacle balloon-like, non-curved. Spermoviduct lacking accessory genital gland. Epiphallus short, rounded, amply connected to penis. Penis internally comprising two chambers separated by strong and complex folded, transverse, ring-like, its distal edge comprising tall fold with well-developed sphincter.

**Type species.** *Catracca uhlei* new species.

**List of included taxa.** *Catracca uhlei*. **Etymology.** The genus' name is in apposition and is derived from the Portuguese word for ticket gate, *catraca*, as a make-up word; anyhow, its central region usually has a reinforced core sustaining the bars, this piece resembles the described shell. Gender: feminine.

**Taxonomic discussion.** see Discussion item.

*Catracca uhlei* new species (Figs 20–24).

**ZooBank:** urn:lsid:zoobank.org:act:F8EEDDB8-F57F-40C2-831E-5F667C529BCF.

**Type specimens.** Holotype MZSP 151900. Paratypes: MZSP 151901, 17 shells, MNRJ 23647, 1 shell, USNM 1661730, 1 shell, all from type locality. BRAZIL. **Minas Gerais**; Itacarambi (Weslley Vailant de Matos col), entrance of Parque Nacional do Peruaçu (ii.2019), 15°10'30"S 44°13'24"W, 530 m altitude, MZSP 152161, 21 shells, MZSP 152222, 15 shells, MZSP 152242, 11 shells, Vargem Grande, 15°00'50"S 44°04'34"W, 466 m altitude, MZSP 151871, 3 specimens (7.ii.2020); Januária, border of Parque Nacional do Peruaçu, 14°57'55"S 44°04'21"W, 485 m altitude, MZSP 151893, 1 specimen, MZSP 151892, 10 shells (i.2020).

**Type locality.** BRAZIL. **Minas Gerais**; West of Itacarambi, 15°00'38"S 44°07'10"W, 540–550 m altitude [Weslley Vailant de Matos col, ii.2019].

**Etymology.** The specific epithet is in honor of Mauricio Sergio Uhle, São Paulo, a shell collector and sponsor of expeditions.

**Description.** Shell. (Fig 20A–20I) Adult shell around 45 mm, conical-oval; apex blunt; wider on last whorl; width ~1/2 shell length. Walls relatively thick. Color uniform light beige

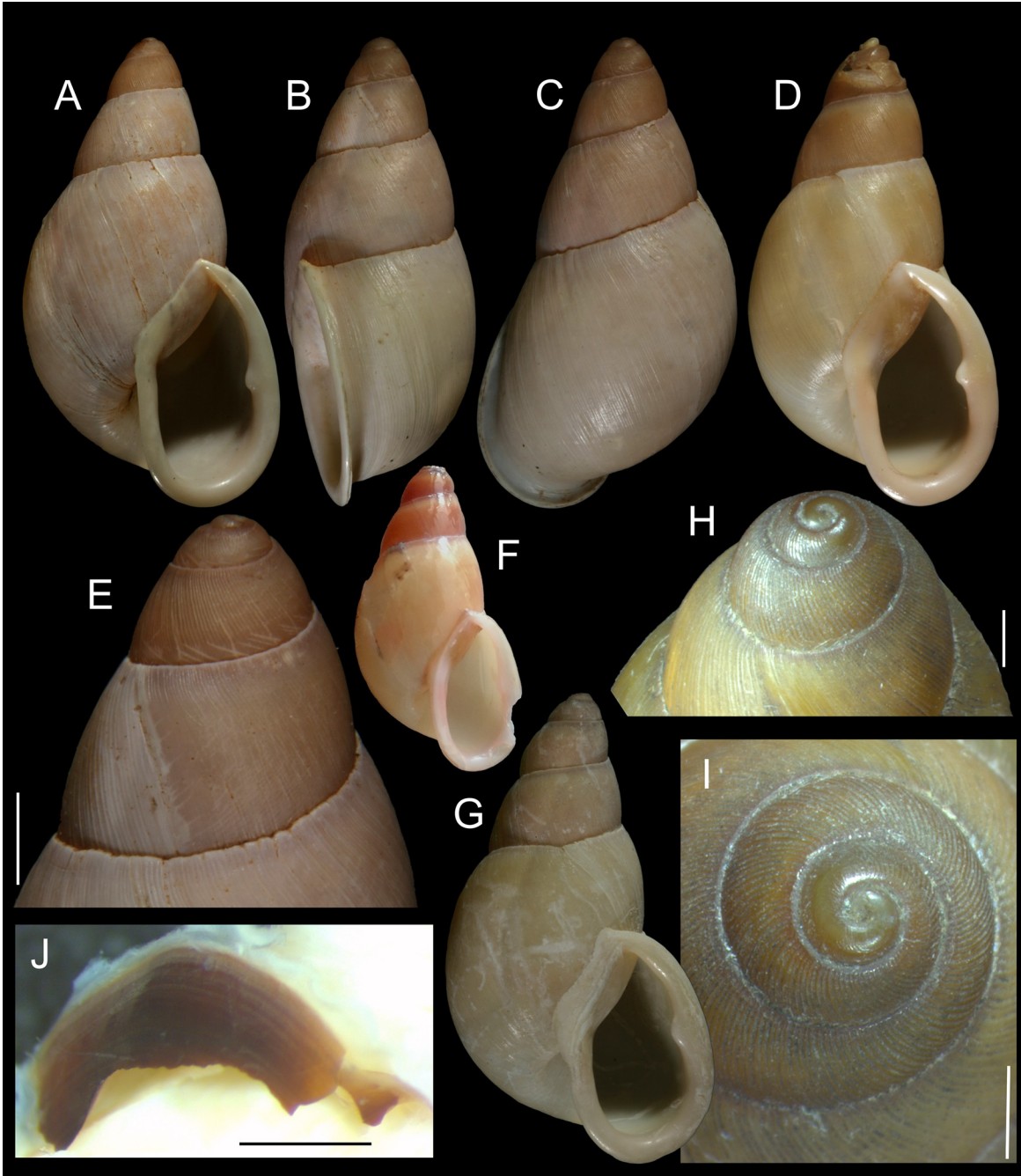

**Fig 20. *Catracca uhlei*, shells and jaw.** (A-C) Holotype MZSP 151900 (L 44.2 mm). (A) Frontal view. (B) right view. (C) dorsal view. (D) Paratype MZSP 151901#1 (L 42.8 mm), frontal view; (E) holotype protoconch in profile, scale = 3 mm. (F) paratype 151893, shell of one of dissected specimens, frontal view (L 47.1 mm). (G) Paratype MZSP 151901#2 (L 43.0 mm). (H) Paratype MZSP 151871, young specimen, detail of apex, profile-slightly apical view, scale = 2 mm. (I) Same, apical view, scale = 2 mm. (J) Jaw plate, ventral view (slightly broken in left end, right in Fig), scale = 1 mm.

or yellowish pale beige; some few specimens with reddish-brown tone in inferior third of spire whorls (Fig 20F). Spire angle ~40˚. Protoconch of 3 whorls (Fig 20B and 20E), ~23% of length, width of ~9 mm; first whorl smooth, gradually axial narrow cords appearing (Fig 20H and

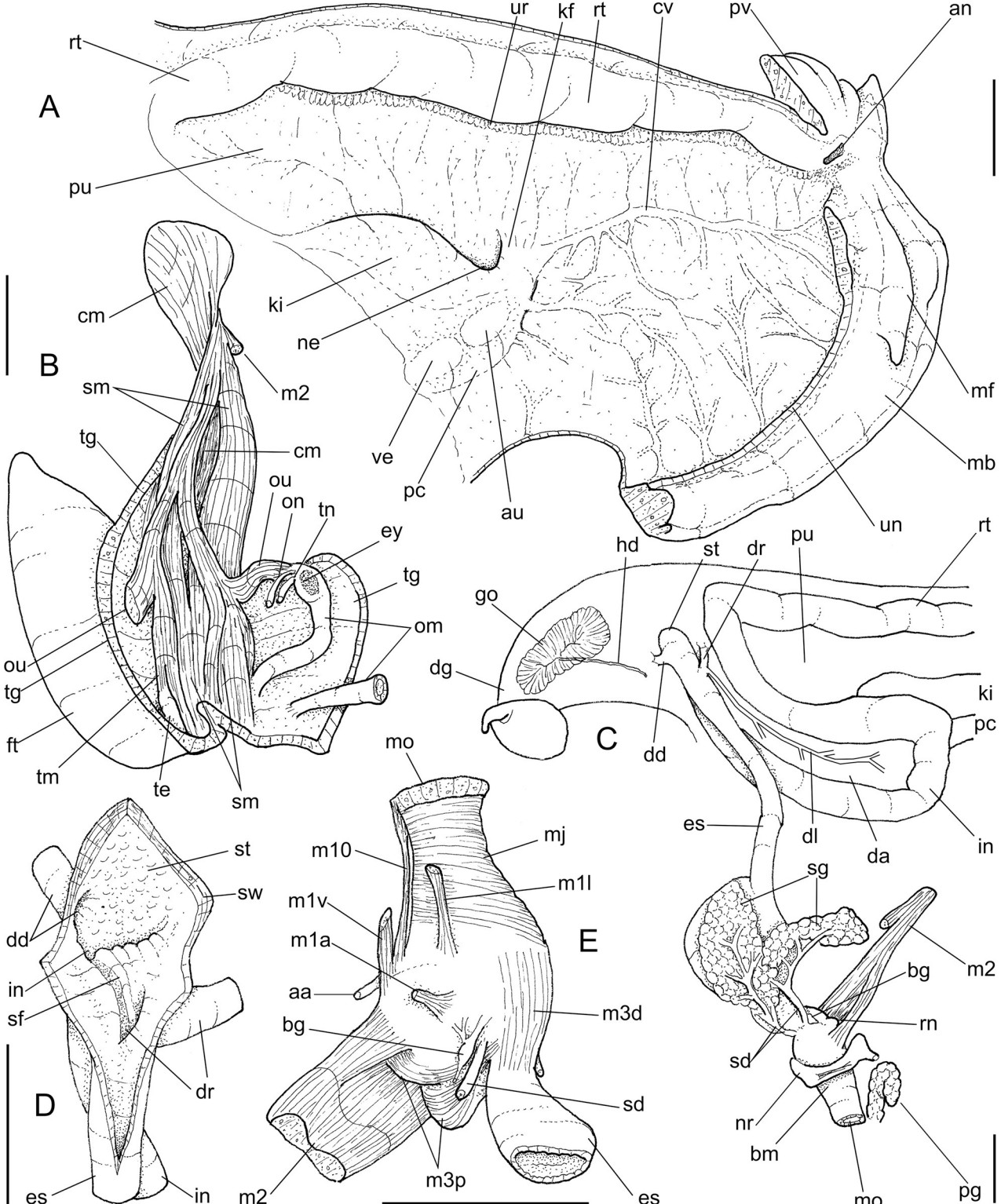

**Fig 21. *Catracca uhlei*, anatomy.** (A) Pallial cavity roof, ventral view, pneumostome ventral lip sectioned and deflected upwards. (B) Head-foot, head, digestive and genital structures removed, anterior integument sectioned and deflected to left (right in Fig), main concern to main head-foot musculature. (C) Fore- and midgut, mostly ventral view, shown as in situ, topology of some adjacent structures also shown. (D) Stomach, ventral view, mostly sectioned longitudinally to show inner surface. (E) Buccal mass, left view. Scales = 5 mm.

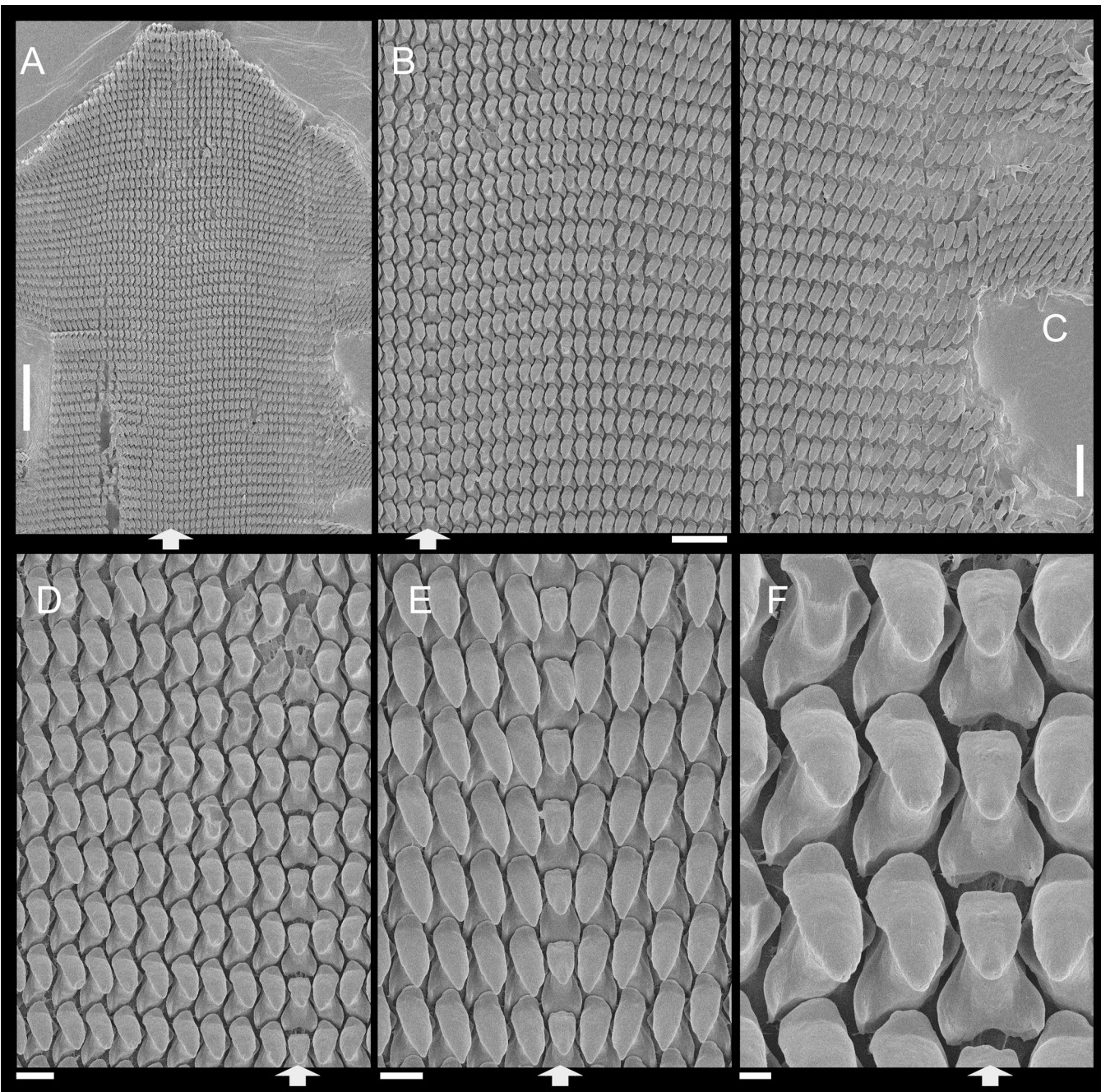

**Fig 22. *Catracca uhlei*, radula in SEM, holotype MZSP 151893.** (A) wide view, scale = 300 μm; (B) detail of central region, scale = 100 μm; (C) detail of lateral region, scale = 100 μm; (D) detail of central region, scale = 50 μm; (E) same, higher magnification, scale = 30 μm; (F) same, higher magnification, scale = 10 μm. White arrows indicating rachidian column.

20I), running from suture to suture, interspaces similar to each cord's width, ~10 per mm; transition to teleoconch distinct, slightly prosocline (Fig 20E and 20H). Teleoconch of ~2 whorls, profile slightly convex; sculpture similar to that of protoconch last whorl, as axial, uniform, complete cords and growth lines, slightly more irregular on last whorl (Fig 20A, 20C, 20D and 20G). Suture well-marked, slightly channeled (Fig 20B, 20C, 20E and 20G). Aperture

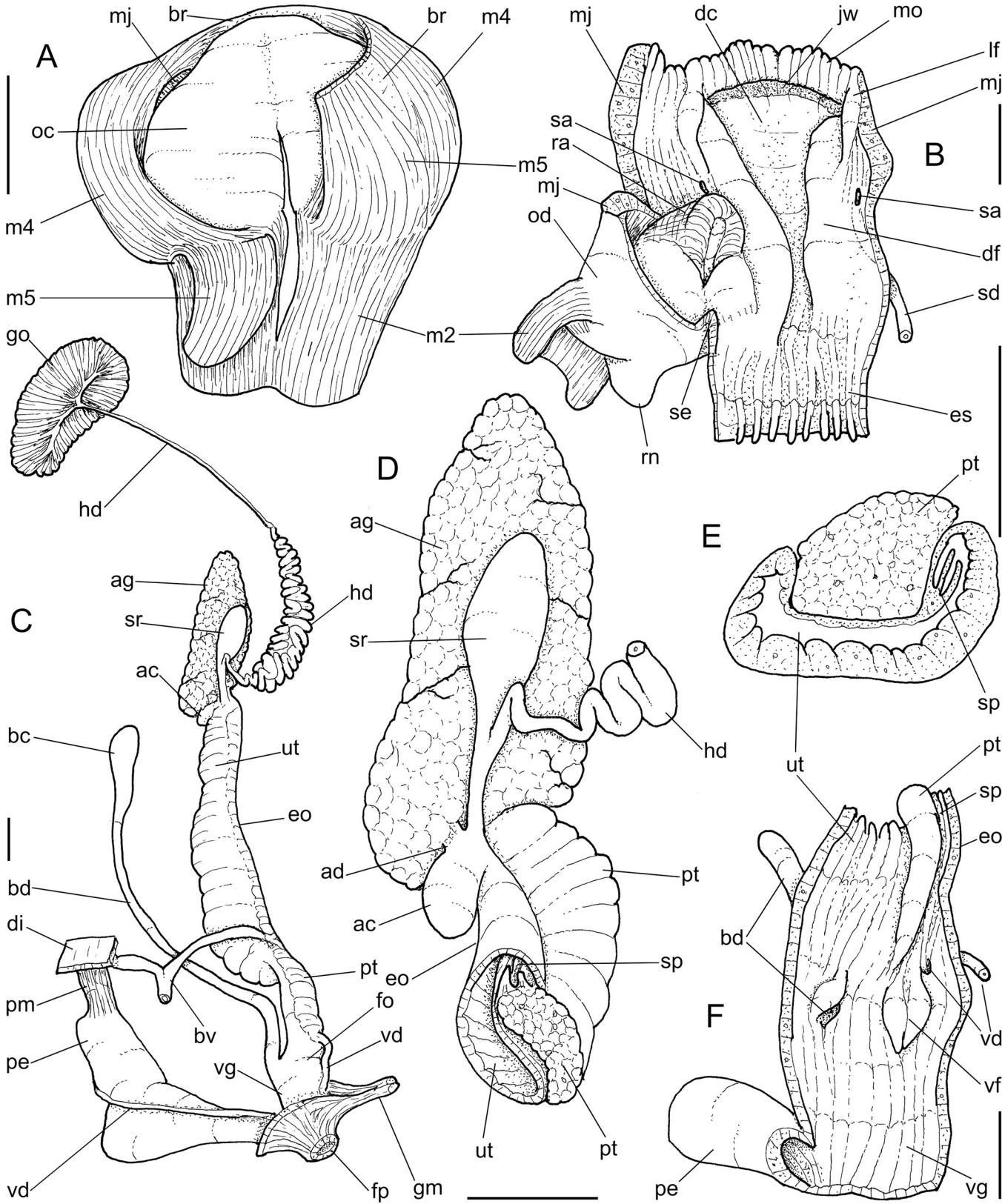

**Fig 23. *Catracca uhlei*, anatomy.** (A) Odontophore, dorsal view, superficial layer of muscles and structures removed, left muscles deflected. (B) Buccal mass, ventral view, sectioned longitudinally, odontophore deflected to left. (C) Genital system, mostly ventral view. (D) Middle region of genital system isolated. (E)

Spermoviduct, transverse section in its middle level. (F) Anterior region of genital system, ventral view, female portion sectioned longitudinally to show inner surface. Scales = 2 mm.

slightly prosocline (~10° from longitudinal general axis) (Fig 20B), oval; ~45% of shell length, ~50% of shell width. Peristome continuous, reflected, thick, same color as remaining shell; outer lip arched, usually with small, blunt tooth medially (Fig 20A, 20D and 20G) absent in younger adults (Fig 20F); inner lip with inferior half almost straight, superior, parietal half appressed to body whorl, with thick callus, sometimes planar in last whorl (Fig 20D and 20F) sometimes flanked by shallow outer furrow (Fig 20A and 20F). Inferior half of inner lip partially covering umbilicus. Umbilicus relatively wide (Fig 20A, 20D and 20G), ~10% of inferior area. Body whorl ~2/3 shell length; usually regularly increasing in size (Fig 20A, 20D and 20F), more rarely wider (Fig 20G).

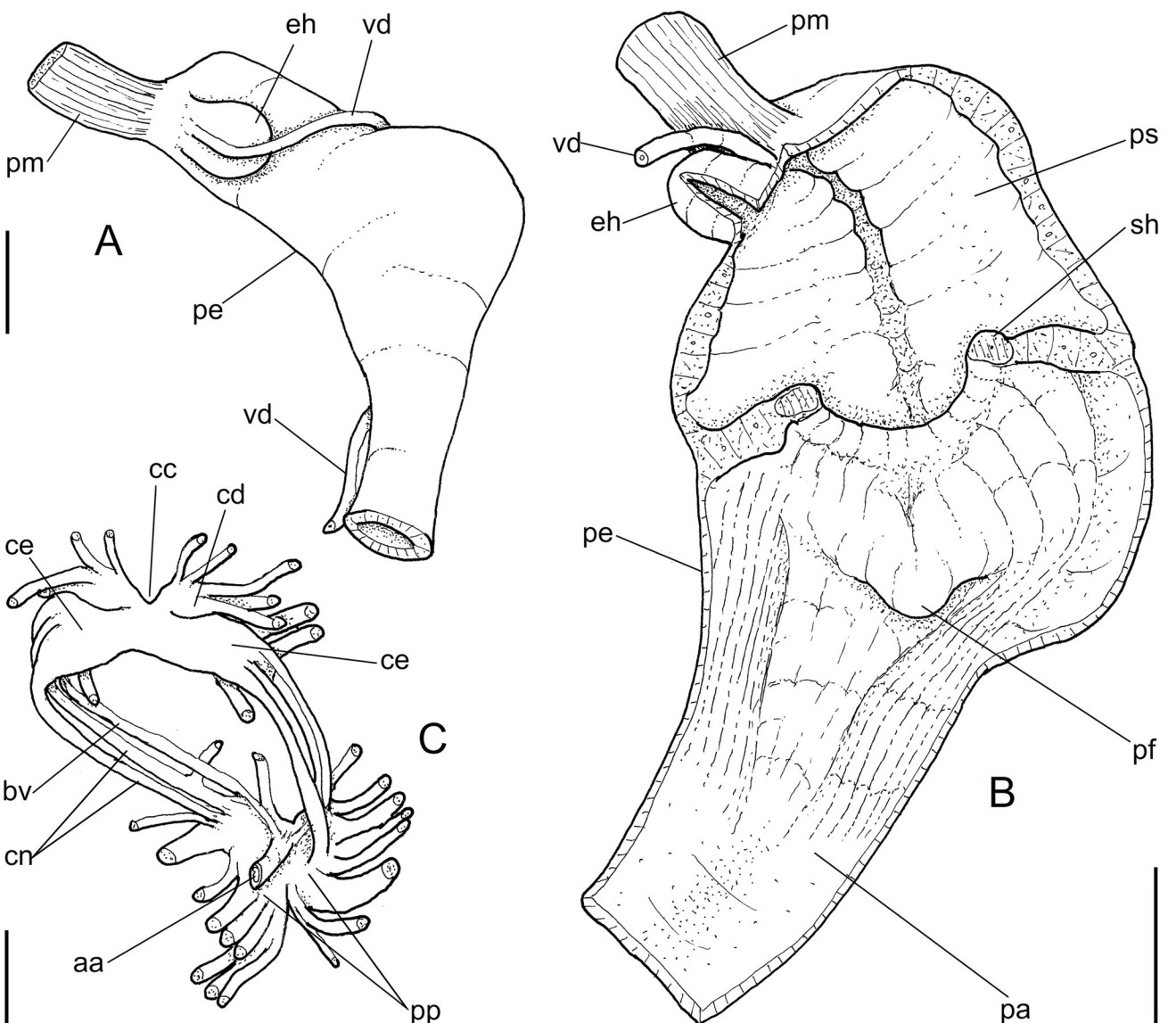

**Fig 24. *Catracca uhlei*, anatomy.** (A) Penis, dorsal view. (B) Same, sectioned longitudinally to show inner surface. (C) Nerve ring, dorsal view. Scales = 2 mm.

**Head-foot.** (Fig 21B) of normal shape. Color uniformly pale. Columellar muscle thick, 1.5 whorls in length. Inner arrangement of columellar annexed muscles relatively complex. Main columellar bundle (cm) occupying ventral floor of haemocoel, relatively flat, wide slightly broader than half of foot width. Pair of secondary cephalic muscles (sm) with middle and anterior regions approximately double thickness and half width those of columellar main bundle (left muscle slightly broader than right one); originating gradually along dorsal surface of main columellar bundle, running anteriorly on main bundle, inserting in ventral region of perioral region. Pair of tentacular/ommatophore muscles originating between posterior and middle thirds of main columellar muscle bundle, between both secondary cephalic muscles; gradually thickening and bifurcating towards anterior; strong bifurcation halfway between their detected origin and anterior end, dorso-lateral branches becoming short and strong ommatophore muscles (ou), inserting near tip of ommatophore (om); ventro-medial branches running towards anterior, dorsally to secondary cephalic muscles, halfway newly bifurcating, lateral branch becoming short tentacular muscles (tm), inserting in tip of tentacles (te); medial branch running further towards anterior, inserting dorsally to insertion of secondary cephalic muscles. Pedal gland large, curved, protruding in ventral region of buccal area (Fig 21C: pg).

**Mantle organs.** (Fig 21A) Mantle border thick, lacking pigments. Pneumostome (pn) protected by simple right ventral flap (pv), width ~1/5 of aperture length. Dorsal fold well developed (mf), occupying ~1/3 of dorsal mantle edge length; left end pointed, projected beyond its base. Pneumostome (pn) ~1/10 of shell aperture length, bearing exclusively air entrance and urinary gutter. Anus (an) separate aperture located at right, adjacent to pneumostome. Lung of 1.5 whorls in length, wide and elongated; right side ~1.5 times longer than left side. Pulmonary vessels conspicuous, but low; 6–7 stronger vessels perpendicularly disposed at right; 6–7 main vessels at left radially disposed; all pulmonary vessels visible, but low, i.e., weakly protruded. Pulmonary vein (cv) running longitudinally across pallial cavity roof medially or towards right, anterior end trifurcated. Reno-pericardial area beige, slightly triangular, located posteriorly within pallial cavity, its posterior abutting wall of visceral cavity, occupying ~30% of cavity length and ~40% of its width (details below). Rectum (rt) wide. Urinary gutter (ur) narrow, running along rectum; anterior urinary gutter surrounding left anal aperture.

**Visceral mass.** (Fig 21C) ~3 whorls in length. Both digestive gland lobes brown in color; anterior lobe (da) flattened, occupying ~1/5 of visceral volume, located just posteriorly to pallial cavity, continuous to kidney. Posterior lobe (dg) larger, extending 2 spiral whorls, occupying ~60% of visceral volume. Stomach ~1/15 of visceral volume, located between both digestive gland lobes, about 3/4 whorl posterior to pallial cavity (st). Digestive tubes (described below) surrounding anterior lobe of digestive gland. Gonad multi-lobed, cream color, encased between posterior lobe of digestive gland and columella, occupying ~1/3 whorl, ~1/10 of visceral volume.

**Circulatory and excretory systems.** (Fig 21A) Pericardium (pc) ~twice as long as wide, located obliquely between middle and left thirds of posterior end of pallial roof, appressed against right lateral side of kidney; occupying ~5% of lung area. Auricle (au) located anteriorly, as continuation from pulmonary vein (cv); ventricle (ve) located posteriorly, larger. Kidney (ki) simple, dorso-ventrally flattened; size reported above; somewhat triangular, width ~2/3 of length; internally organized as two distinct regions–a longitudinal hollow cavity on left; filled by sponge-like renal tissue on right ~2/3 from posterior to anterior ends, being gradually broader towards anterior. Nephropore (ne) small, longitudinal slit at anterior-left corner of kidney, directed towards right; protected by curved anterior projection of kidney (kf). Urinary gutter (ur) composed of small folds perpendicular to rectum axis, flanking almost entire rectum left edge, starting at short distance from pulmonary postero-right end, ending in pneumostome as narrow urinary groove flanking left edge of anus (an).

**Digestive system.** (Figs 21C–23B) Oral tube wide, muscular (Fig 21E: mj). Radula features (Fig 22) below. Jaw simple, thick, dark brown, exposed portion horseshoe-shaped, thicker at middle (Fig 20J); surface having only coaxial low undulations and growth lines; located ai anterior end of buccal dorsal chamber (Fig 23B: jw). Buccal mass spherical, occupying~1/5 of haemocoel volume (Fig 21C: bm). Dorsal surface of oral cavity with well-developed pair of dorsal folds (Fig 23B: df), width of each ~1/3 of dorsal wall width; separated from each other by dorsal chamber (dc) as wide as folds, but broader anteriorly. Odontophore with ~60% of buccal mass volume (Figs 21E and 23B: od). Odontophore muscles (Figs 21E–23B): **mj**, jaw and peribuccal muscles originating in outer-ventral surface of odontophore cartilages (Figs 23A: mj), running towards dorsal, splaying in dorsal wall of oral tube (Fig 21E: mj); **m1**, jugal muscles covering entirely haemocoelic structures, more concentrated close to mouth; **m1a**, small pair of lateral muscles, origin in latero-ventral region of haemocoel, running dorsally short distance, inserting in lateral surface of odontophore; **m1l**, small pair of lateral protractor muscles, origin in lateral region of buccal area, running towards posterior along 2/3 buccal mass length, inserting in lateral region of odontophore, anterior to m1a; **m1v**, small pair of ventral protractors jugal muscles, originating in ventral surface of haemocoel close to mouth, running towards posterior, inserting in ventral-posterior region of odontophore close to m2 insertion (Fig 21E: m1v); **m2**, strong pair of retractor muscles of buccal mass, or radular muscles, originating as single bundle in columellar muscle posterior end (Fig 21B: m2), running anteriorly close to median line along ~60% of haemocoel length, inserting as two different bundles in ventro-posterior edge of odontophore, surrounding at some distance radular nucleus (Fig 23A: m2); **m3d**, thin layer of longitudinal fibers immersed in dorso-posterior wall of odontophore, preceding esophageal origin (Fig 21E); **m3p**, thin layer of dorso-ventral fibers covering posterior odontophore wall, ventral to esophagus (Fig 21E); **m4**, main pair of dorsal tensor muscles of radula, thick, originating in postero-medial region of odontophore cartilages (Fig 23A: m4), surrounding outside and medially cartilages, inserting in subradular membrane in its region correspondent to buccal cavity; **m5**, pair of thick auxiliary dorsal tensor muscles of radula, originating on posterior-medial surface of m4 and short portion in postero-ventral region of odontophore cartilages (Fig 23A: m5), running towards median line covering m4, inserting in subradular membrane by side of m4 insertion; **m6**, horizontal muscle absent (Fig 23A); **m7**, absent; **m10**, pair of narrow ventral odontophore protractor muscles, originating in ventro-anterior region of haemocoel, just ventral to mouth, running towards posterior along buccal mass length (Fig 21E: m10), inserting in latero-posterior surface of odontophore close to m2 insertions; **m11**, pair of narrow ventral tensor muscles of radula absent. Odontophore non-muscular structures (Fig 23A and 23B): **oc**, pair of odontophore cartilages flattened, rather elliptical, anterior region slightly projected anteriorly close to median line, ~1.2 times longer than wide, fused with each other along ~half in their anterior-medial edge (Fig 23A: oc), posterior end roughly rounded; **sc**, subradular cartilage, with expanding region in buccal cavity protecting subradular membrane. Radular sac short, not extending beyond odontophore.

**Radula.** (Fig 22) Slightly longer than odontophore; with rachidian teeth, and ~45 pairs of lateral/marginal teeth; no clear distinction between lateral and marginal teeth (Fig 22A), marginal teeth only slightly narrower than lateral teeth, and more inclined medially (Fig 22C: right side); all teeth with base as long as mesocone, articulating neighboring rows (Fig 22D and 22F); mesocone arched, curved inwards and posterior, ~twice longer than wide, apex bluntly pointed; no secondary cusps. **Rachidian** tooth (Fig 22E and 22F: arrow) as large as lateral teeth; base ~twice longer than wide, slightly flattened, barely triangular, with arched posterior edge articulating with neighbor tooth and pair of distal reinforcements; mesocone ~half of base, symmetrical, located slightly dislocated posteriorly from adjacent lateral teeth row (Fig 22E and 22F). **Lateral teeth** similar to rachidian (Fig 22B and 22D), except in being slightly

longer, asymmetrical, arched towards medial region; base trapezoid, with medial concavity articulating with neighbor medial tooth and single distal reinforcement (Fig 22F). **Marginal teeth** starting with no clear boundary with lateral teeth (Fig 22A and 22C) occupying ~1/3 of each side; shaped similarly to lateral teeth, except for being slightly narrower and more inclined medially (Fig 22C).

Salivary glands covering esophagus in its region preceding its anterior quarter (Fig 21C: sg), forming two elongated, white, thin masses. Each salivary duct differentiable in middle and anterior side of glands, with ~1/12 of esophageal width (Fig 21C, 21E and 4B: sd). Salivary duct running in both sides of esophageal origin, penetrating buccal mass wall in region close to buccal ganglia (Figs 21E and 23B: sd), running immersed in buccal dorsal wall along ~1/3 its length (Fig 23B: sd). Salivary ducts opening as small slits in middle level of dorsal folds, on their lateral side (Fig 23B: sa).

Esophagus ~1-whorl long, with thin, flaccid walls (Fig 21C: es); anterior 1/3 clearly broader, crop-like; posterior 2/3 narrower; inner surface simple, with 5–10 simple longitudinal folds. Stomach (Fig 21C: st) narrow, curved, almost not bulging; position and size described above (visceral mass); gastric walls relatively thick muscular (Fig 21D: sw); inner surface mostly smooth (Fig 21D), except for narrow longitudinal fold along smaller curvature (sf), connecting esophageal duct to digestive gland with intestinal origin; series of low, right folds flanking anterior side of longitudinal fold. Esophageal insertion on right side, intestinal origin on left side, both close to columella. Duct to anterior lobe of digestive gland at short distance from esophagus and intestine intersection (Fig 21C and 21D: dr) wide, running shortly towards right, bifurcating; additional left duct (Fig 21C: dl) originated in main duct base with ~1/3 of its width, running narrowly towards left and anterior along digestive gland lobe compressed by first intestinal loop (da), periodically possessing short ramifications along its length. Duct to posterior lobe of digestive gland located short distance from intestinal origin, posterior to above-described duct, directed towards opposite side (Fig 21C and 21D: dd) slightly wider than anterior duct. Intestine (in) initially as wide as esophageal insertion, gradually broadening up to ~double that width along its wide sigmoid loop in anterior lobe of digestive gland (Fig 21C: in). Rectum and anus position described above (pallial cavity) (Fig 21A: rt, an). Anus sessile, as slit in right end of mantle edge directly turned outside; inner surface with 8–10 simple longitudinal, relatively tall folds.

**Reproductive system.** (Figs 23C–24B) Gonad position described above (visceral mass), composed of 8–10 lobes with minute digitiform acini (Fig 21C: go). Hermaphroditic duct (Fig 23C: hd) narrow and weakly coiled in posterior half, abruptly becoming wider (up to 3-times wider), more intensely coiled in anterior half, coiling diminishing only close to insertion (Fig 24D: hd). Seminal receptacle (Fig 23C and 23D: sr) large, oval, balloon-like sac, ~twice longer than wide, ~three times wider than hermaphroditic duct. Fertilization complex simple, located at narrow and elongated base of seminal receptacle (Fig 23D) as duct of seminal receptacle; slightly shorter than length of receptacle. Fertilization complex totally immersed in albumen gland, inserting in posterior end of spermoviduct, by side of albumen gland duct, relatively wide, twice wider than receptacle's duct. Albumen gland (Fig 23C and 23D: ag) solid, white, elliptical, more-or-less same size as gonad (~1/3 whorl). Albumen gland duct subterminal, connected to distal end of spermoviduct (Fig 23D: ad), in lateral, large albumen chamber (Fig 23D: ac); widely connected to distal end of spermoviduct (Fig 23D: eo). Spermoviduct (eo) of ~1.5 whorl in length, slightly narrower than albumen gland, ~10 times longer than wide. Prostate gland occupying ~3/4 of spermoviduct surface and ~1/2 its volume (Fig 23C and 23E: pt). Uterus occupying ~3/4 of spermoviduct space, external walls thick-glandular (Fig 23C and 23E: ut), inner surface mostly covered by wide oblique folds (Fig 23D: ut). Sperm groove entirely composed of two tall folds (Fig 23D and 23E: sp), becoming vas deferens some

distance posterior to genital pore (~1/15 of remaining spermoviduct length) (Fig 23C and 23F: vd), with ~1/10 of anterior spermoviduct width, running outside of penis shield (Fig 23C). Vagina ~1/20 spermoviduct length (Fig 23C and 23F: vg); inner surface simple, with 8–10 longitudinal, low, wide folds (Fig 23F: vg). Bursa copulatrix almost as long as spermoviduct length; bursa duct ~half of adjacent spermoviduct width in its origin, gradually narrowing towards posterior end (Fig 23C: bd); bursa oval, ~1/3 of albumen gland size (Fig 23C: bc), located encased between pericardium and adjacent intestinal loop. Penis ~3/4 of spermoviduct length, ~2/3 its anterior width (Fig 23C: pe); penis muscle inserting terminally, short, broad (Figs 23C and 24A: pm), with no branch on epiphallus. Epiphallus ~1/7 penis' length, tip rounded, located laterally (Fig 24A: eh) inner surface with smooth, widely opening to penis posterior chamber (Fig 24B: eh). Vas deferens inserted subterminally in penis tip, in intersection with epiphallus (Fig 24A and 24B: vd). Internal penial surface with two clear chambers (Fig 24B), strong transverse septum dividing penis in about similar-sized posterior (ps) and anterior (pa) chambers; transverse septum having strong sphincter in edge (sh), and proximal blunt projection bulging in a side of anterior chamber (pf); this transverse septum swelling middle portion of penis externally (Figs 23C and 24A); posterior penis chamber with wide furrow running from opposite side of epiphallus aperture, up to region close to transverse septum (Fig 24B: ps); anterior penis chamber with strong, wide longitudinal fold located at base of blunt projection (pf), tapering towards anterior; low, wide longitudinal folds covering entire surface of anterior penis chamber (pa). Genital pore with radial arrangement of muscles, with pair of more developed muscles (Fig 23C: gm) running dorsally and posteriorly, connecting with middle region of columellar muscle.

**Central nervous system.** (Fig 24C) Nerve ring located across buccal mass (Fig 21C: nr). Pair of cerebral ganglia (ce) almost fused; cerebral commissure slightly narrower than ganglia; each ganglion about as wide as adjacent esophageal section; several wide nerves originating in cerebral antero-lateral region. Cerebral node or gland (cd) located in postero-medial quadrant, with ~1/8 each ganglion's size. On each side, parallel, rather narrow connectives (cn) running between cerebral ganglion and ventrally located fused pedal and pleural ganglia (pp), accompanied by blood vessels (bv) issuing from anteriorly-directed aorta. At least six pairs of nerves originating from anterior side of pedal-visceral ganglion complex. Pair of statocysts not seen.

**Distribution.** Minas Gerais region around Parque Nacional Cavernas do Peruaçu, so far from Itacarambi (north) to São João da Ponte (south).

**Habitat.** Limestone outcrops, Cerrado forest, 466–759 m altitude.

**Measurements.** (length and width in mm) holotype MZSP 151900 (Fig 20A–20C): 44.2 by 23.5 (Fig 20A–20C); paratypes MZSP 151901#1 (Fig 20D): 42.8 by 20.7; #2: 43.0 by 24.4 (Fig 20G); MZSP 151893: 47.1 by 25.3 (Fig 20F).

**Material examined.** Types (reported above). BRAZIL. **Minas Gerais** (W.V. Matos col); Itacarambi, Vargem Grande, 15˚00'50"S 44˚04'34"W, 466 m altitude, MZSP 151869, 3 shells (7. ii.2020); São João da Ponte, near Olímpio Campos, 15˚50'44"S 44˚00'03"W, 759 m altitude, MZSP 151811, 1 shell (i.2020).

**Taxonomic remarks.** see Discussion item.

## Discussion

The taxon descriptions presented above do not include comparative remarks, which are reported in this item. However, the descriptions are presented in a comparative scenario with the preceding ones, evidencing the main differences amongst the taxa herein studied. Besides, there are the diagnoses, which bring the set of character necessary to identify each taxon. The present section is, thus, more focused on the most interesting features and those more relevant

to taxonomy and present/future phylogenetic approaches to resolving relationships among South American strophocheilids.

## Generic debate

As reported above, *Anthinus* appears to encompass the smallest strophocheilids [6], and those with a slightly "Bulimulus"-like, bulimoid shell, i.e., a fusiform shell with a relatively small aperture. These features easily differentiate the genus from the remaining cofamiliar genera that are bigger and with proportionally larger aperture. *Catracca*, introduced here, actually breaks this rule, as it is similar to *Anthinus* in these features, so much so that samples were initially identified as belong to that genus. Despite the *Catracca* shell lacking the colorful pattern of *Anthinus*, the discovery of *A. vailanti*, which also possesses a rather monochromatic shell, affects the generic separation only based on conchology. The anatomical features were fundamental to determine that these are distinct genera.

The set of characters reported here in diagnoses of both genera are sufficient to individualize them amongst the strophocheilids. However, it is recognized that the remaining genera do not have the same level of details known. Despite Leme [5] having redefined all strophocheilid (including megalobulimines) genera, including their anatomy, the level of details was not the same as those reported herein. Contrary to Leme's [5] definition of *Anthinus*, some characteristics were not confirmed in the present study. In particular, the pair of colored bands of the antero-dorsal region of the animal were not found. Further, all examined *Anthinus* species possessed the genital accessory gland (e.g., Figs 4E, 10D and 14F: as) that was considered absent, and being an exclusive feature of the megalobulimines. Other characteristics reported in Leme's [5] diagnosis are accurate, such as the weakly muscular stomach lacking inner folds (e.g., Fig 6B), the curvature of the seminal receptacle (e.g., Figs 8C and 24B: sr) (which is also shared with other strophocheilines), and the shape of the epiphallus (e.g., Fig 4C: eh). Other interesting characteristics that now can be added to the *Anthinus* diagnosis include the low profile of the pulmonary vessels, usual for species of smaller size, dark pigmented (e.g., Fig 13L); the double origin of the odontophore muscle pair m5 (e.g., Fig 6D); the posterior insertion of the anterior (esophageal) duct to the digestive gland, inserted practically into the stomach (e.g., Fig 6B: dr), its bifurcation, with both components of similar caliber (e.g., Fig 6C: dr, dl); the presence of the pallial, pre-rectal muscle (another supposed exclusive character of the megalobulimines); and the penis inner arrangement of folds forming a pair of stronger posterior folds, the furrow between which is continuous with the epiphallus aperture.

The new genus *Catracca* shares some characteristics with *Anthinus*, distinguishing both from other strophocheilids, such as the relatively small shell aperture; the small but distinct umbilicus (Fig 20A and 20D); the configuration of the robust head musculature (Fig 21B); the posterior position of the anterior duct of the digestive gland (Fig 21C, despite the left branch, dl, is much narrower); and of the nerve ring (Fig 24C). However, *Catracca* has a set of exclusive characters that justify its introduction, reported in its diagnosis above, and discussed further below. In some aspects, the *Catracca* protoconch resembles those of *Megalobulimus* Miller, 1878 in being blunt, wide and with uniform axial sculpture (Fig 20E, 20H and 20I). It differs, however, in being much more elongated and of smaller absolute size. The other strophocheilid genera have the protoconch with delicate reticulated sculpture and much more acuminate [4,6]; *Catracca* actually has the bluntest perspective, with a proportionally widest protoconch amongst the strophocheilines, producing a somewhat mamillated shell. From the anatomical diagnostic characters of *Catracca*, which particularly distinguish it from *Anthinus*, the more interesting are the thick muscular stomach (Fig 21D: sw), which in muscular development is inferior only to the gizzard-like stomach of *Megalobulimus* [5]; the odontophore pair m5

entirely originated from the pair m4; the sac-like, non-curved seminal receptacle (Fig 23D: sr); the absence of accessory genital gland in the spermoviduct (Fig 23E); and by the pair of vaginal folds (Fig 23F: vf). However, the more outstanding distinctive feature of *Catracca* is its inner penial arrangement (Fig 24B). The large transverse septum with well-developed sphincter in its edge (sh) is unique amongst the strophocheilids. A structure approaching this was only found only in *Mirinaba* Morretes, 1955, particularly in *M. planidens* (Michelin, 1831) ([5]: fig 54). However, in the latter, the transverse septum is much lower, thicker, and lacks the sphincter. Besides, the *Mirinaba* copulatory organ has an elongated epiphallus, more similar to those of *Megalobulimus*. The epiphallus in *Catracca* is most similar to that of *Anthinus*, short and rounded (Fig 23A: eh); it differs, however, in lacking inner papillae cover (e.g., Figs 14C and 20C: eh). Moreover, in *Catracca* the vas deferens inserts in the border between the epiphallus and penis (Fig 23A and 23B), while in *Anthinus* the vas deferens inserts in the middle level of the outer epiphallus wall (e.g., Fig 8E: vd). The penis muscle insertion is restricted to the penis wall in *Catracca* (Fig 23A), while some branches of it insert on the epiphallus in *Anthinus* (e.g., Figs 8B and 17C: pm).

Interestingly, in *Anthinus*, the radula of the type species *A. multicolor* looks more like that of *Catracca uhlei* than those of remaining studied congeners. *A. multicolor* and *C. uhlei*, share short, blunt mesocones, particularly those of the rachidian teeth (Figs 3C and 22F: arrow), while the mesocone of the remaining *Anthinus* species are much more elongated, flattened, spatula-like, with the rachidian more similar to lateral. The radulae of both genera differ from those of the megalobulimines in lacking broad mesocones, of almost the same size as the bases. The radulae of the other strophocheilid genera are not known.

The presence of supposedly megalobulimine characteristics in *Anthinus*, such as the pre-rectal pallial muscle (e.g., Fig 17A: mp) and the accessory genital gland (e.g., Fig 17D: as); as well as the megalobulimine supposed exclusive characters in *Catracca*, such as the protoconch (Fig 20E) and the sac-like seminal receptacle (Fig 23D: sr) are of difficult interpretation in respect to generic interrelationships. In the provisional phylogenetic treatment below, the accessory genital gland resulted in a convergence between megalobulimines and *Anthinus*. The same result (convergence) was obtained for the regular narrow axial cords in the protoconch shared between *Catracca* and some megalobulimines. The sac-like seminar receptable, on the other hand, appears to be plesiomorphic in the family.

**Specific debate.** A gallery of shells of all known *Anthinus* species are assembled in Fig 25A–25H, 26A–26H, based on representatives (mostly holotypes–D–H), as well as of *Catracca uhlei* (Fig 25I and 26I holotype). In that, the most outstanding feature is the monochromia of *A. vailanti* (Fig 25F and 26F) and of *C. uhlei*, which already sets them apart from the remaining that possess characteristic beige-brown spots or blotches forming weakly-defined, spiral interrupted bands, i.e., pale spots or blotches on the beige-brown background. However, some specimens of *A. vailanti* possess some evidence of such banding (Fig 9H–9I). After that, the present debate focuses on *Anthinus* species, considering that the diagnosis of each species already brings the set of characters exclusive of each species that supported their description. The present discussion focuses on the characters, every numerical parameter is based on a N reported in the Table 1. The Table 2 summarizes the main conchological differences among the species studied herein. The present text, thus, only has additional information. Related to the shell elongation, *A. multicolor* is the slenderer (width/length W/L index = 46%) (Fig 1A–1C, 25C), with a group orbiting W/L ~50%, such as *A. synchondrus* (Fig 25E– 54%), *A. albolabiatus* (Fig 25A– 49%), *A. miersi* (Fig 25B– 51%) and *A. turnix* (Fig 25D– 55%); while another group has shells broader, encompassing species with W/L = 60%, such as *A. vailanti*, *A. morenus* (Fig 25G) and *A. savanicus* (Fig 25H). The protoconch absolute size, in mm, is also an important parameter, as usually is relatively constant within each species; the largest

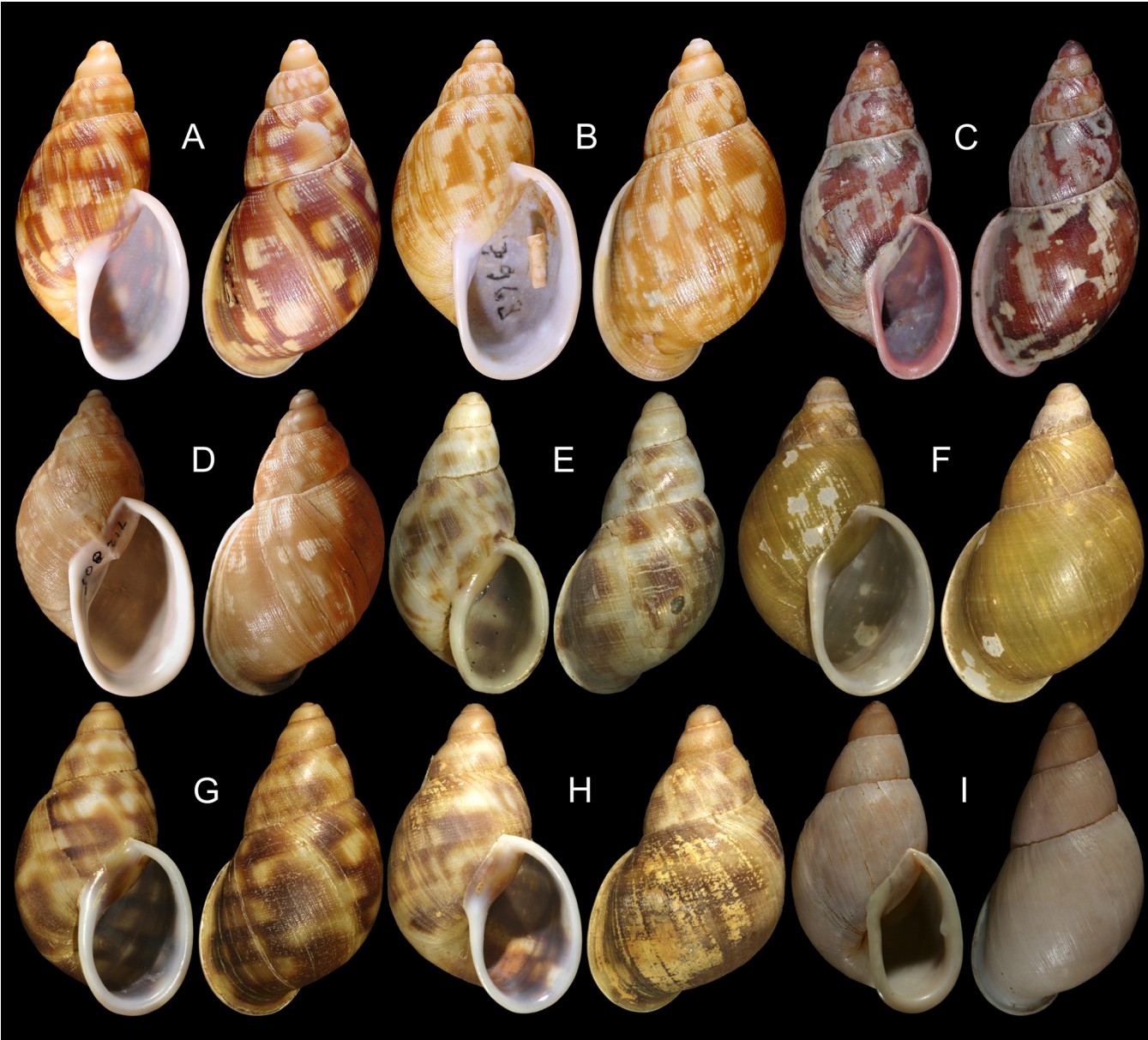

**Fig 25. Gallery of shells of all *Anthinus* species, frontal and dorsal views.** (A). *A. albolabiatus*, MZSP 7958 (L 50 mm). (B) *A. miersi*, MZSP 7963 (L 53 mm). (C) *A. multicolor*, MZSP 29620 (L 26 mm). (D) *A. turnix*, holotype USNM 712805 (L 50 mm). (E) *A. synchondrus* holotype MZSP 152074 (L 36.9 mm); (F) *A. vailanti* holotype MZSP 152891 (L 29.0 mm). (G) *A. morenus* holotype MZSP 152891 (L 35.8 mm). (H) *A. savanicus* holotype MZSP 154391 (L 32.5 mm). (I) *Catracca uhlei* holotype MZSP 151900 (L 44.2 mm). A to D extracted from Simone (2006).

protoconch is that of *C. uhlei*, with 9 mm of diameter; it is followed by that of *A. vailanti*, with 8.8 mm; in the range of ~7 mm there are *A. savanicus*– 7.7 mm, *A. albolabiatus*– 7.5 mm, and *A. synchondrus* and *A. morenus* with 7.0 mm; the range of ~6 mm starts with *A. turnix*– 6.8 mm. *A. miersi*– 6.3 mm, ending with the smallest protoconch of *A. multicolor*– 6.1 mm. In relation to the aperture, the amplest is that of *A. turnix*, with aperture length occupying 60% if shell length, and 67% of shell width; it is followed by *A. miersi*– 58%-60% respectively; the smallest proportional peristome is that of *C. uhlei*– 45%-50%, followed by *A. synchondrus*– 48%-50% and *A. savanicus*– 48%-51%; the remaining species are in the range of peristome size

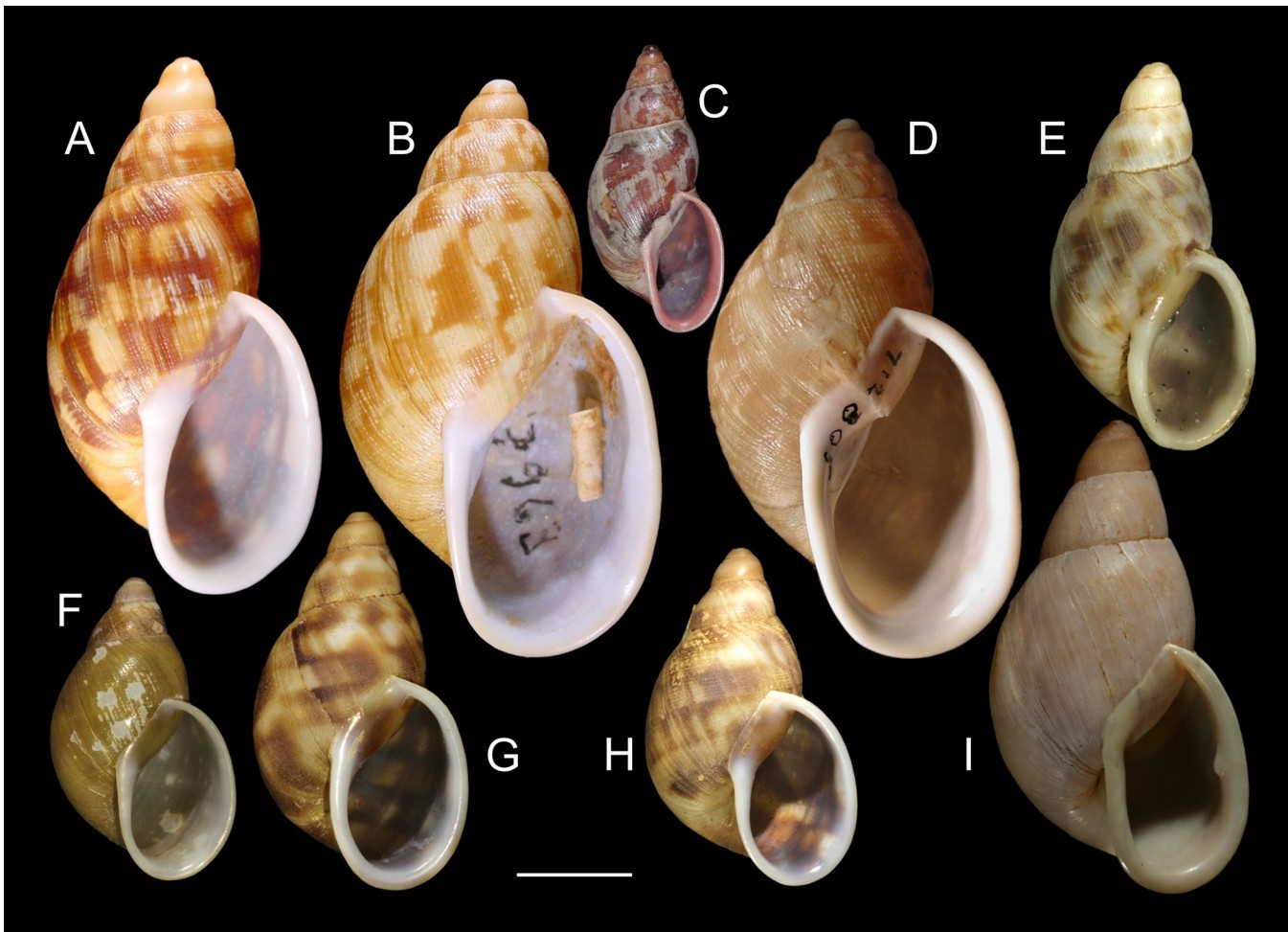

**Fig 26. Gallery of shells of all *Anthinus* species in same scale, frontal views.** (A). *A. albolabiatus*, MZSP 7958. (B) *A. miersi*, MZSP 7963. (C) *A. multicolor*, MZSP 29620. (D) *A. turnix*, holotype USNM 712805. (E) *A. synchondrus* holotype MZSP 152074. (F) *A. vailanti* holotype MZSP 152891. (G) *A. morenus* holotype MZSP 152891. (H) *A. savanicus* holotype MZSP 154391. (I) *Catracca uhlei* holotype MZSP. A to D extracted from Simone (2006). Scale = 10 mm.

about half of the shell's size, as follows: *A. morenus*– 51%-51%, *A. albolabiatus*– 52%-54%, *A. multicolor*– 52%-55%, and *A. vailanti*– 55%-60%. The shell's absolute size is also an interesting parameter, with *A. miersi* being the larger, with an average of 53 mm, followed by *A. turnix*– 50 mm; the smallest species is *A. vailanti*, with average size of 30 mm, followed by *A.*

**Table 2. Summary of main conchological differences among the species studied in this paper (all measures are approximations of average measures, see Table 1).**

| species | Shell size (mm) | Shell color | Spire angle | Protoconch whorls | Peristome x length | Width x length | Apex |
|---|---|---|---|---|---|---|---|
| *Anthinus multicolor* | 35 | mosaic | 40˚ | 2.5 | 52% | 46% | narrow |
| *Anthinus synchondrus* | 40 | mosaic | 35˚ | 3.2 | 48% | 54% | narrow |
| *Anthinus vailanti* | 30 | Monoch-romatic | 50˚ | 3 | 55% | 60% | narrow |
| *Anthinus morenus* | 35 | mosaic | 50˚ | 3 | 51% | 60% | narrow |
| *Anthinus savanicus* | 32 | mosaic | 50˚ | 2.8 | 48% | 60% | narrow |
| *Catracca uhlei* | 45 | Monoch-romatic | 40˚ | 3 | 45% | 50% | blunt |

*savanicus*– 32 mm, and *A. morenus*–*A. multicolor*, both with 35 mm; *A. synchondrus*– 40 mm, and *A. albolabiatus*– 45 mm, are intermediary. The average size of *C. uhlei* is 45 mm.

Regarded to *Anthinus multicolor*, a more detailed discussion on the loss of the types, geographic distribution, and shell variation, is found elsewhere [4]. In the original description [29], Rang reported that the examined material came from Brazil, particularly Saint-Paul province (the State of São Paulo), and from near Corcovado, which today is a place inside the city of Rio de Janeiro, State of Rio de Janeiro. Bequaert [4] reported the species as occurring in these two states (Rio de Janeiro and São Paulo), and figured a topotype ([4]: pl. 4, figs. 3, 7; MCZ 74441, from Corcovado), which is similar to the original illustration ([29]: pl 3, fig. 1). The specimen has irregular dark spots in its shell, lacking apparent spiral alignment as usually have the other species (Fig 25). The shell illustrated by Simone [6] (Fig 25C) was collected in Teresópolis, located close to Serra dos Órgãos, contiguous with the Corcovado rainforest. It also shows the same spot patterns of the previous illustrations [4,29]. The presently studied sample were collected in São Fidélis, a place ~80km far from Corcovado, but also apparently in the same biome, as it is close to the Desengano State Park. The shells of the presently studied sample (Fig 1A–1C) also have the irregular, not spirally aligned spots, but they possess a wide spiral clear band along the middle level of the last whorl. This band is not present in all specimens of that lot, and, thus, is interpreted as variation. The non-alignment of the dark spots is, thus, an interesting species' distinctive character.

Related to anatomy, beyond the species described herein, there is some information only on *A. miersi* (radula, pallial roof and genitalia [7]) and the generic statement by Leme [5]. The anatomical discussion on the other species is, thus, precluded, and even on *A. miersi* the details are not equivalent. The mantle edge of the studied *Anthinus* has the usual pattern of most pulmonates, which includes a small left fold from the pneumostome (e.g., Fig 21A: mf, which is pointed to the left); this fold has shown relatively constant in each species and of taxonomical value, in *A. synchondrus* and in *A. savanicus*, it has an elongated projection (Figs 6A and 16H: mf); it is pointed in *A. vailanti*, and simply rounded in *A. morenus* and *A. multicolor* (Figs 13L and 22A). The pallial cavity is interestingly darkly pigmented in all species (e.g., Fig 134H: pu), in which the pulmonary vessels are difficult to discern. The different pattern of lung vascularization is evident in the drawings represented for each species. Furthermore, the relative size of the reno-pericardial area varies: *C. uhlei* and *A. vailanti* (Fig 21A), considered of medium size (~1/4-1/5 of total cavity area); while *A. synchondrus* (Fig 6A) and *A. savanicus* (Fig 16H) can be considered large, occupying ~1/3 of lung area; and *A. morenus* (Fig 13L), and *A. multicolor* (Fig 2A), can be considered small (about 1/5 of lung area). The pallial pre-rectal muscle, as referred above, is supposedly a megalobulimine characteristic [5], but found in all here studied *Anthinus* examined here (Figs 6C and 17A: mp) except for *A. multicolor* (Fig 2A). In the digestive system, the jaw plate shows interesting comparative data; it is deeper and broad in *C. uhlei* (Fig 20J), *A. vailanti* (Fig 9K–9L), *A. savanicus* (Fig 16G), *A. multicolor* (Fig 1G and 1H), and possibly in *A. miersi* [7]; however, it is shallow and wide in *A. synchondrus* (Fig 5M), and *A. morenus* (Fig 13I); moreover, a ventral additional medial fold reinforces the jaw plate exclusively in *A. vailanti* (Fig 9L), and the jaw lacks a median anterior projection or beak only in that of *A. morenus* (Fig 13I). Related to the odontophore intrinsic and extrinsic musculature, they have shown lightly similar; the most variable pairs are the m1, m3 and m7, which can be modified, present or absent according to the species (see descriptions), however, the variation found in pair m5 appears to be more interesting; it is 100% originated from m4 in *C. uhlei*, but it is ~50% originated from m4 and 50% from cartilages in *A. synchondrus*, *A. savanicus* and *A. multicolor*, it is ~60% originated from m4 in *A. morenus*, and ~20% in *A vailanti* (Fig 10A). The degree of odontophore cartilages fusion along the ventral-inner edge is also of taxonomic interest, both cartilages are ~50% fused in *C. uhlei* (Fig 23A) and *A. morenus* (Fig 14B); ~90%

fused in *A. vailanti* (Fig 10A) and *A. multicolor*; ~85% fused in *A. savanicus* (Fig 17B); while the cartilages are entirely fused in *A. synchondrus* (Fig 6D). The radular teeth's mesocone is short, compared to those of the other genera of the family, in *C. uhlei* (Fig 22), *A. multicolor* (Fig 23), and *A. miersi* [7]; however, it is elongated, spatula-like in *A. synchondrus* (Fig 7), *A. vailanti* (Fig 11), *A. morenus* (Fig 15), and *A. savanicus* (Fig 18). This mesocone elongation looks something exclusive of this subgroup of *Anthinus* amongst the known strophocheilids. An additional character is easily differentiating the rachidian tooth and its neighbors; the species with short mesocone, the differentiation between both is easy, while those with elongated mesocone, the rachidian is very similar to the neighboring teeth, being of difficult differentiation; from those, *A. synchondrus* and *A. vailanti* with certain effort it is possible to point out which is the rachidian column; however, this is very difficult in *A. morenus* and *A. savanicus*. Another interesting feature of the examined species is the posterior position of the anterior duct to the digestive gland, called the esophageal duct, which connects to the anterior lobe of the digestive gland. This esophageal duct originates practically from the stomach, while at least in *Megalobulimus*, it is far from the stomach; this feature is not known in the other strophocheilid genera. Beyond the odd position of the esophageal duct to digestive gland, there is the strong bifurcation of it, i.e., it bifurcates shortly after its origin. This characteristic is found in *Catracca* and in *Anthinus*, being only different in size; both branches have relatively similar caliber in *Anthinus* (e.g., Fig 6C: dr, dl), while the left branch is narrower in *Catracca*; this does not occur at least in *Megalobulimus*, but this is unknown in other related genera.

The genital system usually is very informative in heterobranchs, but it has shown relatively conservative in strophocheilids, differing only in small details reported below. The presence of the supposedly exclusive megalobulimine accessory genital gland in spermoviduct (e.g., Fig 8D and 8F: as) of the *Anthinus* is extraordinary. The seminal receptacle offers some distinction of the examined species, being balloon-like in *Catracca*, as well in megalobulimines [5], and strongly curved in *Anthinus*; however, the seminal receptacle is amply, widely curved in *A. vailanti* (Fig 10B: sr) and *A. multicolor* (Fig 4B); while it is tightly curved in *A. synchondrus* (Fig 8C), *A. morenus* (Fig 14E), and *A. savanicus* (Fig 19A). The insertion of the hermaphrodite duct (hd) in the seminal receptacle is in its base in all studied species, except for *A. vailanti*, in which it is inserted in the middle of its concave edge; this insertion is strongly curved in *C. uhlei*, *A. savanicus* and *A. multicolor*, while it is straight, rather perpendicular in *A. synchondrus*, *A. vailanti* and *A. morenus*. The penis relative size is ~80% of the spermoviduct in *A. savanicus* (Fig 17C: pe), ~60% in *A. synchondrus* (Fig 8B) and *A. vailanti* (Fig 10C), and ~50% in *C. uhlei* (Fig 23C), *A. morenus* (Fig 14D), *A. multicolor* (Fig 2C), and *A. miersi* [7]. The accessory genital gland in the spermoviduct is relatively small in *A savanicus* (Fig 4B: as) and *A. multicolor*, while it is large in the other three species. The number and form of the sperm-grooves in the spermoviduct is also variable according to the species; they are double and similar-sized in *C. uhlei* (Fig 23E: sp) and *A. multicolor* (Fig 4E), they are also double, but asymmetrical in *A. synchondrus* (Fig 8F), it is single and with a spiral fold in *A. vailanti* (Fig 10D), they are double but of different thickness in *A. morenus* (Fig 14F), they are double, but one of them is T-shaped in *A. savanicus* (Fig 17D). In the vaginal-free oviduct cavity, *C. uhlei* has a double vaginal fold (Fig 4F: vf), while *A. multicolor* has a hump (Fig 4C: hp); the internal pattern of folds is different in each species, and are schematized in their respective drawings (Figs 4A, 8D, 12A and 19B). The insertion of the penis muscle shows also interesting distinctions, it is only inserted in the penis wall in *C. uhlei* (Fig 24A: pm), it has two small branches additionally connected to epiphallus in *A. synchondrus* (Fig 8B), and *A. vailanti* (Fig 10C), it has a larger component connected to epiphallus in *A. morenus* (Fig 14D), it has a single and broad branch to epiphallus in *A. savanicus* (Fig 17C), and it has a small branch connected to epiphallus and another to vas deferens in *A. multicolor* (Fig 4C and 4D). As told above, all

*Anthinus* have a papillae cover in epiphallus lumen, but the papillae are small and uniform in *A. synchondrus* (Fig 8E), of medium size in *A. vailanti* (Fig 12B), and tall in *A. morenus* (Fig 14C), *A. savanicus* (Fig 19C) and *A. multicolor* (Fig 4D); additionally, the papillae extend further along the penial chamber in *A. multicolor*, and along an adjacent area in *A. vailanti*; besides, the papillae of *A. multicolor* are not uniform. The internal pattern of penial folds is also distinctive, *A. morenus* is the only species in having a single and simple large posterior fold (Fig 14C: pf); all remaining *Anthinus* have a pair of parallel folds in a side: *A. synchondrus* has these folds composed of a series of transverse threads (Fig 8E); *A. vailanti* is the only in having this pair fused in the proximal region (Fig 12B); *A. savanicus* has this pair formed by the end of the secondary folds of the remaining penial inner surface (Fig 19C); while *A. multicolor* has these folds wide and covered by papillae (Fig 4D). Interestingly, this pair of folds has its intermediary furrow continuous with the aperture of the epiphallus. All these internal structures are exteriorized through copulation, including the complex structure of *Catracca* (Fig 24B). To understand how they work during copula is a fascinating issue for future investigation.

**Anatomical and broader taxonomical debate.** As referred above, details of the some remaining strophocheilid genera are still unknown, and what is known is mostly focused in megalobulimines. This precludes further inferences on the intrinsic and extrinsic interrelationships and phylogeny of the taxa. Leme (e.g., [5,10,23]) had that as a main professional goal, but it remains unfinished. The intention is to find a character or a set of characters supporting each taxon, which, based on lack of studies, so far is impossible. Megalobuliminae, firstly introduced as a family [5] really looks like a subdivision of Strophocheilidae, possessing the largest species (regarded as an anti-predatory strategy [30]), and supported at least by the pulmonary septum, buccal flange, pre-rectal septum in intestine, pre-rectal pallial muscle and accessory genital gland in the spermoviduct [5]. However, the presence of these two last characteristics in *Anthinus*, here presented, shows that these concepts must be reevaluated. Within strophocheilids, only *Megalobulimus* (so far the single genus of Megalobuliminae, with almost a hundred species), and *Anthinus* (this study) have samples with anatomical description sufficiently detailed for supporting more detailed comparative analyses like phylogeny. The other six genera [5,6] are totally unknown or only have some few structures known.

The central nervous system (nerve ring) has shown relatively conservative in both genera studied here, all of them are similar (e.g., Fig 24C), varying only in some small details and proportions, as reported above. At least in its main features, the nerve ring of the studied species is very similar to that of the megalobulimines [10] and even of the Bulimulidae (e.g., [31]).

A phenotypical base to understand the relationship of the Strophocheilidae with the other related families is also difficult for the same reason–lack of anatomical knowledge. At the time Strophocheilidae was only closely related to Dorcasiidae (African) and Acavidae (Asian) within Acavoidea [9], a group that apparently lays large eggs, produced by a thick walled, glandular spermoviduct, the phenotypic analysis was difficult. The scenario became more complicated with the present conception of these families as part of Rhytidoidea, with five additional families, based on molecular approaches [11,12,32]. The plan is to perform studies on other genera of all these families in a similar level of details of the present one, to produce a morphological database enough for a phylogenetic analysis. Only then a confrontation of the present taxonomy will be possible, and, possibly, to provide morphological bases for all subdivisions. The traditional Acavoidea above mentioned looks to be a morphologically well-supported taxon by above mentioned characters. However, the present Rhytidoidea still needs a better phenotypic support, as the fusion of Acavoidea with Rhytidoidea [11] is only based upon two papers with incongruent results, one of them did not have the Acavoidea as a focus [32], and another is an unpublished dissertation [33].

The Table 3 summarizes the main anatomic differences among the species studied herein.

**Phylogenetic analysis.**   Once more it is important to emphasize that the present phylogenetic analysis is not concerned with resolving the family Strophocheilidae, nor any other higher taxa. The main intention, using already published data, is to demonstrate that a new genus was necessary to be introduced, and the species does not fit in any of the known genera.

The computational processing of the Matrix (Appendix 2), which was based on the given list of 56 characters/126 states (Appendix 1) (see Material and Methods and [13]), resulted in a single cladogram (Fig 27) with the following indices: length (L) = 283; consistency index (CI) = 68; retention index (RI) = 89.

In the cladogram (Fig 27), the position of *Catracca uhlei* (after node 3) clearly shows that it does not belong to any known genera. It is recognized that some strophocheilid genera are not represented in the cladogram, but all of them (i.e., *Gonyostomus* Beck, 1837; *Speironepion* Bequaert, 1948; and the Andean *Austroborus* Parodiz, 1949 and *Chiliborus* Pilsbry, 1926) are beforehand reasonably different. The isolated position of *Catracca* in the base of Strophocheilidae (node 3), with two autapomorphies (character 7: *Megalobulimus oblongus*-like protoconch; character 52: transverse septum in the penis) is compatible with the generic separation.

Additionally, some provisional conclusions are possible. The genus *Anthinus* resulted monophyletic (node 7), supported by six synapomorphies: the mosaic shell color (character 4); special arrangement of head muscles (ch. 11); the spatula-like radular mesocone (ch. 29), with rounded tip (ch. 30); the loss of the talon (ch. 42), and of the spermoviduct accessory gland (ch. 44). *Anthinus multicolor*, the type species, resulted as first branch, with two autapomorphies, and separated from the other congeners by three autapomorphies. The other four *Anthinus* species introduced herein resulted in successive branches, with *A. vailanti* at base, and *A. synchondrus*–*A. morenus* together as terminals.

Other interesting provisional results are the monophyly of the South American Strophocheilidae (node 2–14 synapomorphies), separated from the African Dorcasiidae (node 15–3 synapomorphies). The dichotomy of the family into Megalobuliminae (node 11–15 synapomorphies) and Strophocheilinae (node 3–5 synapomorphies) is also corroborated. Also noticeable is the division of the megalobulimines, in which what Leme [5] informally called as "*M. ovatus* group" (node 14–1 synapomorphy) and "*M. oblongus* group (node 12–2 synapomorphies). These possibly can be an indicative of genetic separation, an issue that is in analysis.

The present provisional cladogram achieved its goals, mainly concerned to *Catracca* allocation, as well as shows that *Anthinus* is monophyletic and has a perceptible internal arrangement that can arouse future subgeneric divisions. The increment of this cladogram with more taxa is an in-progress project.

**Ecologic-distribution information.**   The discovery of several new taxa in central regions of Brazil has indicated how weak the knowledge level on the Brazilian malacofauna is. With human activities, possibly species have been lost even before they were at least known. Several Brazilian land snails have shown to be endemic from small areas (personal observations), which is an important factor for conservation efforts and politics. The taxa described in this paper are examples that lots of Brazilian species and even genera still need to be discovered or better-known. Knowing the fauna is the first step to raise protective rules to them and their environments. Thus, their descriptions are relatively urgent.

The type localities of all *Anthinus* and *Catracca* are shown in the map of Fig 28. In that it is possible to see that there is a concentration in the northwest region of Minas Gerais (MG) (and adjacent region of Goiás–GO), and another in Rio de Janeiro (RJ). The single type locality far from those is that of *A. albolabiatus*, in east of Rio Grande do Sul (RS). Reports to *Anthinus* in literature and collections are not common, but samples and citations of *Anthinus* have been collected from south of Minas Gerais to North Argentina and Uruguay, sometimes even

**Table 3. Summary of main anatomic differences among the species studied in this paper (all measures are approximations of average measures, see Table 1).**

| species | Mantle edge folds | Reno-pericardial area | Pre-rectal pallial muscle | fusion of odontophore cartilages | Origin of m4 on cartilages | Jaw plate | Radula mesocone | Radular rachidian and neighboring | Seminal receptacle |
|---|---|---|---|---|---|---|---|---|---|
| *Anthinus multicolor* | rounded | small | absent | 90% | 50% | broad, thick | hook-like | similar | widely curved |
| *Anthinus synchondrus* | projected | large | present | 100% | 50% | narrow | Spatula-like | different | tightly curved |
| *Anthinus vailanti* | with pointed fold | medium | present | 90% | 80% | broad, central fold | Spatula-like | different | widely curved |
| *Anthinus morenus* | rounded | small | present | 50% | 40% | narrow | Spatula-like | similar | tightly curved |
| *Anthinus savanicus* | projected | large | present | 85% | 50% | broad | Spatula-like | similar | tightly curved |
| *Catracca uhlei* | projected | large | absent | 50% | 100% | Ticker at middle | hook-like | different | Balloon-like |
| species | Insertion of hermaphrodite duct | Accessory genital gland | Inner masculine folds | Free oviduct | Length penis x spermoviduct | Insertion of penis muscle | epiphallus papillae type | epiphallus papillae extension | penial inner folds |
| *Anthinus multicolor* | At base, curved | small | pair of similar-shaped | With distal hump | 50% | Mostly in penis tip | tall, uneven | along entire penis chamber | wide and low, with papillae |
| *Anthinus synchondrus* | At base, perpendicular | large | pair of similar-shaped | usual | 60% | Pair on epiphallus | Low, uniform | Only inside | Larger folds composed of transverse subfolds |
| *Anthinus vailanti* | At middle, perpendicular | large | Single, coiled | usual | 60% | Pair on epiphallus | Medium, not uniform | Along penial chamber | Pair simple, part fused |
| *Anthinus morenus* | At base, perpendicular | large | Pair, thick | Usual | 50% | Mostly on epiphallus | tall | Only inside | Single, large |
| *Anthinus savanicus* | At base, curved | small | Pair, one T-shaped | Usual | 80% | Thick branch to epiphallus | tall | Only inside | Pair, large, secondary folds |
| *Catracca uhlei* | At base, curved | absent | Pair, thick | With genital fold | 75% | Penis only | absent | — | Transverse septum |

identified at a specific level. As the areas in such the *Anthinus* samples were better studied–Rio de Janeiro and NW Minas Gerais (Fig 28) show that several species with restricted distribution occur in relatively small areas, the question arises: are the *Anthinus* from the other regions belonging to undescribed species? As in the moment there is no possibility of checking this, no answer is possible.

## Appendix 1 –list of characters that base the phylogenetic analysis

1. Shell walls: 0 = thin; 1 = thick.

2. Shell form: 0 = bulimoid; 1 = discoid.

3. Shell size: 0 = 2–3 cm; 1 = 4–5 cm; 2 = over 6 cm.

4. Shell color: 0 = monochromatic; 1 = mosaic of spots.

5. Periostracum: 0 = permanent; 1 = deciduous.

6. Aperture: 0 = smaller than half of shell length; 1 = longer than half of shell length.

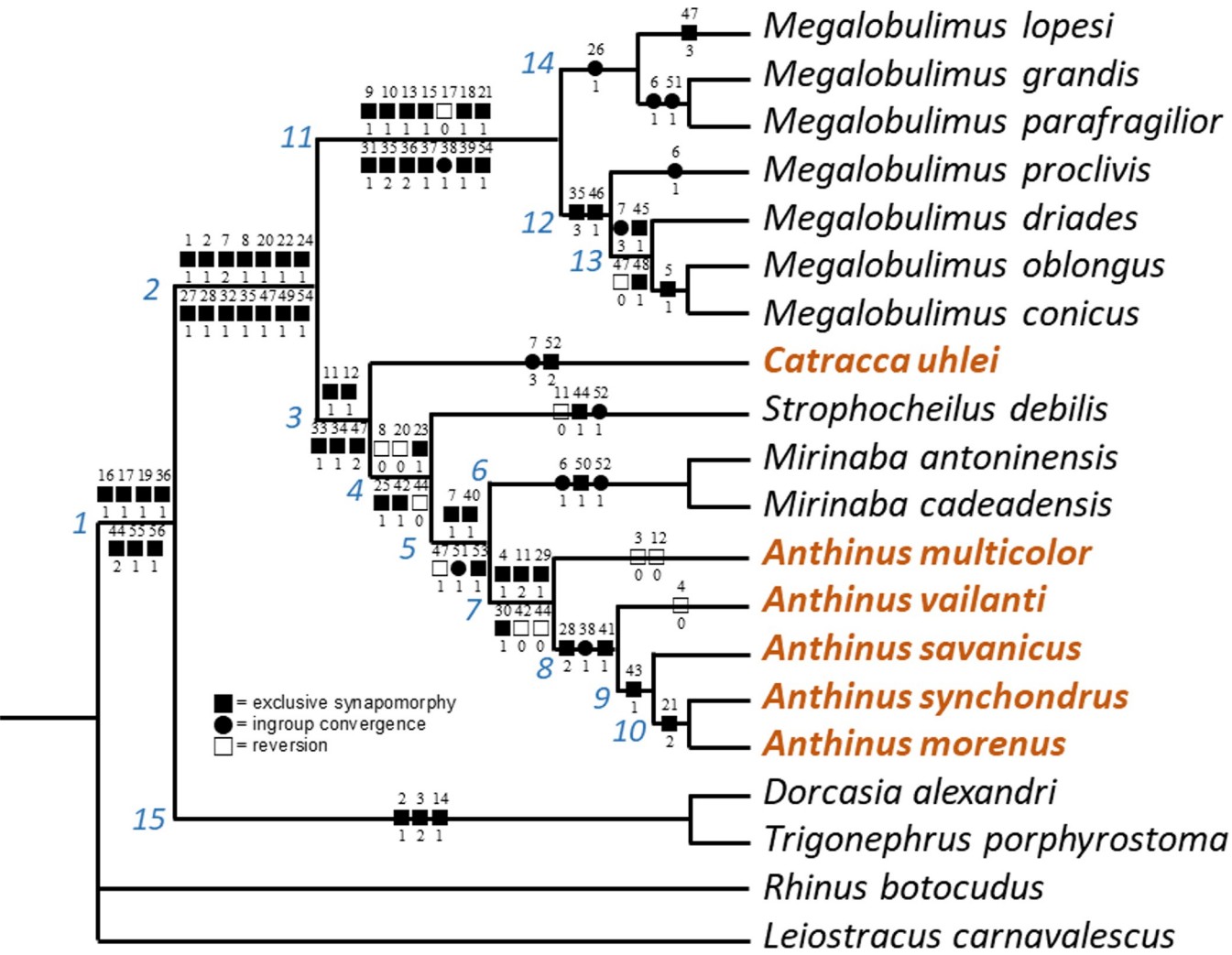

**Fig 27. Single cladogram obtained analyzing the matrix (Appendix 2) of the species assembly reported in Material and Methods (written black), plus the species studied herein (written brown).** Both branches below are far outgroups, a bulimulid (*Leiostracus*) and a simpulopsid (*Rhinus*). Branch *15* as close outgroups, two African dorcasiids. Each branch with synapomorphies shown. Each symbol as in the box; above number the character, below number the state. Branch numbers for discussion in blue italics. Method: parsimony; L = 283; CI = 68; RI = 89.

7. Protoconch sculpture: 0 = smooth or reticulate; 1 = reticulate with axial predominance; 2 = irregular axial threads; 3 regular narrow axial cords.

8. Protoconch form: 0 = pointed; 1 = blunt.

9. Head peri-oral flange: 0 = absent; 1 = present.

10. Haemocoel layers of head rectractor muscles: 0 = two; 1 = single.

11. Inner layer of head retractors: 0 = only present in ommatophores; 1 = ommatophores and peribuccals separated; 2 = ommatophore and peribuccal fused.

12. Mantle edge inner pneumostome fold: 0 = simple; 1- angulous.

13. Differentiated vessel close to pneumostome: 0 = absent; 1 = present.

14. Afferent pallial pulmonary vessels: 0 = single; 1 = a pair.

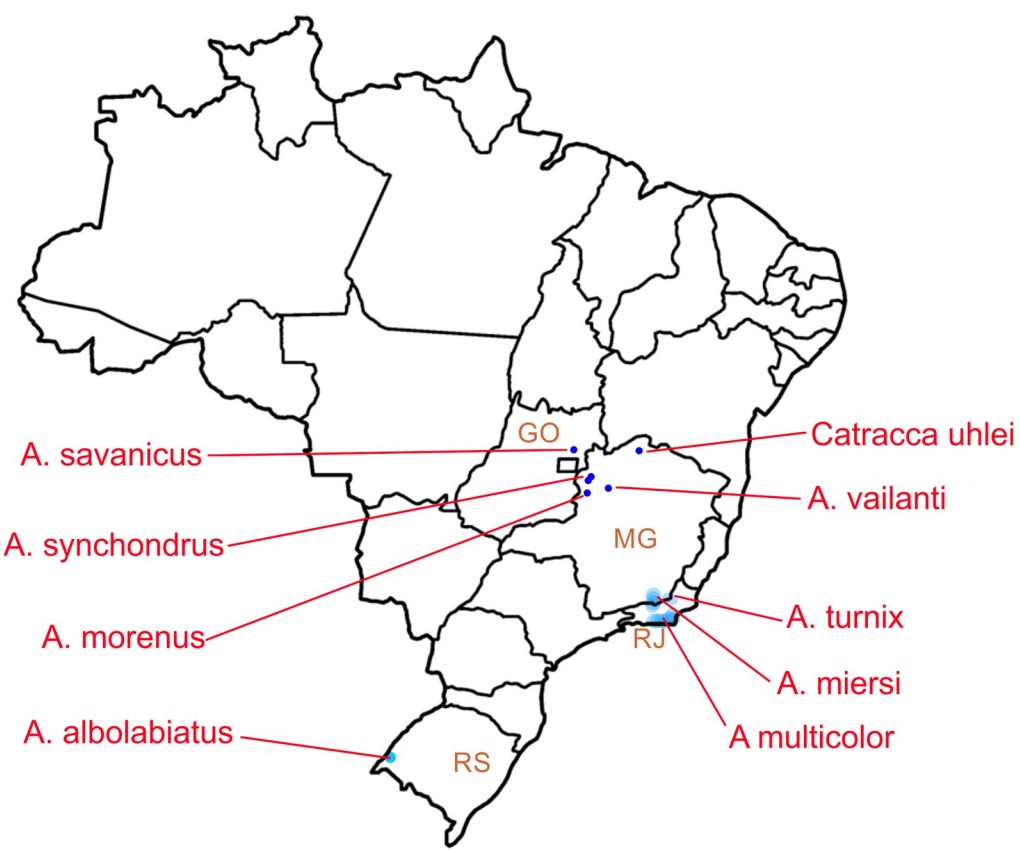

**Fig 28. Self-produced Brazil map with states outlined and type localities of each *Anthinus* (*A.*) and *Catracca* shown.** New localities in dark blue. Previous localities in light blue. States abbreviations: GO: Goiás; MG: Minas Gerais; RJ: Rio de Janeiro; RS: Rio Grande do Sul.

15. Pulmonary septum: 0 = absent; 1 = present.

16. Nephropore: 0 = protected in primary ureter; 1 = exposed in pallial cavity.

17. Nephropore: 0 = simple; 1 = protected by anterior renal fold (? in outgroup).

18. Kidney: 0 = smooth; 1 = covered by anastomosed net of vessels.

19. Urinary pallial way: 0 = via duct; 1 = via groove.

20. Ureter groove: 0 = smooth; 1 = transversely folded.

21. Jaw plate: 0 = smooth; 1 = transversely folded; 2: narrow.

22. Horizontal muscle (m6): 0 = present; 1 = absent.

23. Insertion of pair m5: 0 = on pair m4; 1 = partially on cartilages.

24. Pair of muscles inside radular sac (m7): 0 = present; 1 = absent.

25. Inner edge of cartilages (anterior to fusion): 0 = thick; 1 = thin.

26. Radular teeth: 0 = rachidial similar to lateral; 1 = different.

27. Radular teeth: 0 = bi- or multicuspid; 1 = unicuspid.

28. Radular mesocone: 0 = smaller than base; 1 = as long as base; 2 = longer than base.

29. Radular mesocone: 0 = stubby; 1 = flattened, spatula-like.

30. Radular mesocone tip: 0 = pointed; 1 = rounded.

31. Tissue on radula at end of radular sac (to): 0 = simple; 1 = reinforced.

32. Salivary duct between salivary gland and buccal mass: 0 = long; 1 = short.

33. Esophageal duct to anterior lobe of digestive gland: 0 = far from stomach; 1 = close to stomach.

34. Esophageal duct to anterior lobe of digestive gland: 0 = single; 1 = closely bifid

35. Stomach: 0 = non-muscular; 1 = weakly muscular; 2 = highly muscular; 3 = high muscular connected to typhlosole (add).

36. Stomach inner folds in its smaller curve: 0 = absent; 1 = present; 2 = extending to remaining stomach (add).

37. Pre-rectal valve: 0 = absent; 1 = present.

38. Pre-rectal pallial muscle: 0 = absent; 1 = present.

39. Anus: 0 = turned outside; 1 = opening inside pallial cavity.

40. Seminal receptacle: 0 = balloon-like; 1 = U-shaped.

41. U-shaped seminal receptacle: 0 = widely curved; 1- tightly curved.

42. Talon (swollen end of hermaphroditic duct) in carrefour; 0 = absent; 1 = present.

43. Duct of albumen gland: 0 = connected in spermoviduct; 1 = connected directly to albumen chamber.

44. Spermoviduct accessory gland: 0 = absent; 1 = small; 2 = large.

45. Prostate: 0 = narrow (~1/3 of diameter); 1 = wide (~1/2).

46. Vaginal appendix: 0 = absent; 1 = present.

47. Epiphallus: 0 = conic, long; 1 = short, rounded; 2 = pseudo-epiphallus; 3 = conic, short.

48. Flagellum: 0 = absent; 1 = single and small.

49. Penis shield: 0 = present; 1 = absent.

50. Penis retractor muscle: 0 = simple; 1 = multiple.

51. Penis retractor muscle insertion: 0 = in penis tip only; 1 = branches in epiphallus.

52. Penis with transverse fold ("penial papilla"): 0 = absent; 1 = present, simple; 2 = with sphincter.

53. Pair of parallel inner folds connecting epiphallus and penis: 0 = absent; 1 = present.

54. Penis inner surface: 0 = smooth; 1 = papillate; 2 = with large longitudinal fold in a side (pilar).

55. Spawn: 0 = minute and numerous capsules; 1 = few, large capsules.

56. Capsules cover: 0 = thin, weakly calcified; 1 = thick, well-calcified (egg like).

## Appendix 2 –matrix of characters that base the phylogenetic analysis

| Species\ character | 1 2 3 4 5 5<br>1234567890 1234567890 1234567890 1234567890 1234567890 123456 |
|---|---|
| *Mirinaba antoninensis* | 1010011000 1100011010 0111101100 0111110001? 100001011 111111 |
| *Mirinaba cadeadensis* | 1010011000 1100011010 0111101100 0111110001? 100001011 111111 |
| *Megalobulimus lopesi* | 1020002111 0010110111 1101011100 1100221110? 102003010 000211 |
| *Megalobulimus grandis* | 1020012111 0010110111 1101011100 1100221110? 102001010 100211 |
| *Megalobulimus parafragilior* | 1020012111 0010110111 1101011100 1100221110? 102001010 100211 |
| *Megalobulimus proclivis* | 1020012111 0010110111 1101001100 1100321110? 002011010 000211 |
| *Megalobulimus oblongus* | 1020103111 0010110111 1101001100 1100321110? 002110110 000211 |
| *Megalobulimus conicus* | 1020103111 0010110111 1101001100 1100321110? 002110110 000211 |
| *Megalobulimus driades* | 1020003111 0010110111 1101001100 1100321110? 002110110 000211 |
| *Strophocheilus debilis* | 1010002000 0100011010 0111101100 0111110000? 101002010 010111 |
| *Catracca uhlei* | 1010003100 1100011011 0101001100 0111110000? 000002010 020111 |
| *Anthinus multicolor* | 1001001000 2000011010 0111101111 0111110001 0002001010 101111 |
| *Anthinus synchondrus* | 1011001000 2100011010 2111101211 0111110101 1012001010 101111 |
| *Anthinus vailanti* | 1010001000 2100011010 0111101211 0111110101 1002001010 101111 |
| *Anthinus morenus* | 1011001000 2100011010 2111101211 0111110101 1012001010 101111 |
| *Anthinus savanicus* | 1011001000 2100011010 0111101211 0111110101 1012001010 101111 |
| *Dorcasia akexandri* | 0100000000 0001011010 000?000000 0000010000? 002000000 000011 |
| *Trigonephrus porphyrostoma* | 0100?00000 0001011010 0????00000 0000010000? 0?2000000 000011 |
| *Leiostracus carnavalescus* | 0000000000 0000000000 0000000000 0000000000? 000000000 000000 |
| *Rhinus botocudus* | 0000000000 0000000000 0000000000 0000000000? 000000000 000000 |

## Acknowledgments

A special thanks to José Coltro Jr, Mauricio Uhle, and their team of collectors for the collect and donation of most studied samples of the present paper. To Lee Kremer for the thoughtful correction of the text. To Gary M. Barker, Jonathan Ablett, a third anonymous referee, and the Editors for the comments, criticisms and substantial improvements to the paper. To Lara Guimarães, MZSP, for the assistance in SEM examinations. The collects were performed under the IBAMA-Sisbio license 10560–2.

## Author Contributions

**Conceptualization:** Luiz Ricardo L. Simone.

**Data curation:** Luiz Ricardo L. Simone.

**Formal analysis:** Luiz Ricardo L. Simone.

**Investigation:** Luiz Ricardo L. Simone.

**Methodology:** Luiz Ricardo L. Simone.

**Project administration:** Luiz Ricardo L. Simone.

**Resources:** Luiz Ricardo L. Simone.

**Software:** Luiz Ricardo L. Simone.

**Supervision:** Luiz Ricardo L. Simone.

**Validation:** Luiz Ricardo L. Simone.

**Visualization:** Luiz Ricardo L. Simone.

**Writing – original draft:** Luiz Ricardo L. Simone.

**Writing – review & editing:** Luiz Ricardo L. Simone.

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
