## [Decision Letter · Decision Letter 0]

7 Mar 2022

PONE-D-22-00958Additions to the genus Anthinus occurring in Minas Gerais and Goiás regions, Brazil, with description of five new species, one of them in the new related genus Catracca (Gastropoda, Eupulmonata, Strophocheilidae).PLOS ONE

Dear Dr. Simone,

Thank you for submitting your manuscript to PLOS ONE. After careful consideration, we feel that it has merit but does not fully meet PLOS ONE’s publication criteria as it currently stands. Therefore, we invite you to submit a revised version of the manuscript that addresses the points raised during the review process.

We look forward to receiving your revised manuscript.

Kind regards,

Michael Schubert

Academic Editor

PLOS ONE

Journal Requirements

2. We noted in your submission details that a portion of your manuscript may have been presented or published elsewhere. 

(previous submission (PONE-D-21-27031) edited by Michael Schubert) 

3.Thank you for stating the following financial disclosure: 

(No) 

(NO authors have competing interests)

4.Please take this opportunity to be sure you have met all of our guidelines for new species. When publishing papers that describe a new botanical taxon, PLOS aims to comply with the requirements of the International Code of Nomenclature for algae, fungi, and plants (ICN). In association with the International Plant Names Index (IPNI), the following guidelines for publication in an online-only journal have been agreed such that any scientific botanical name published by us is considered effectively published under the rules of the Code. Please note that these guidelines differ from those for zoological nomenclature, and apply only to seed plants, ferns, and lycophytes.

Effective January 2012, "the description or diagnosis required for valid publication of the name of a new taxon" can be in either Latin or English. This does not affect the requirements for scientific names, which are still to be Latin.

Also effective January 2012, the electronic PDF represents a published work according to the ICN for algae, fungi, and plants. Therefore the new names contained in the electronic publication of a PLOS ONE article are effectively published under that Code from the electronic edition alone, so there is no longer any need to provide printed copies.

For proper registration of the new taxon, we require two specific statements to be included in your manuscript.

a) In the Methods section, include a sub-section called "Nomenclature" using the following wording:

The electronic version of this article in Portable Document Format (PDF) in a work with an ISSN or ISBN will represent a published work according to the International Code of Nomenclature for algae, fungi, and plants, and hence the new names contained in the electronic publication of a PLOS ONE article are effectively published under that Code from the electronic edition alone, so there is no longer any need to provide printed copies. 

In addition, new names contained in this work have been submitted to IPNI, from where they will be made available to the Global Names Index. The IPNI LSIDs can be resolved and the associated information viewed through any standard web browser by appending the LSID contained in this publication to the prefix http://ipni.org/. The online version of this work is archived and available from the following digital repositories: [INSERT NAMES OF DIGITAL REPOSITORIES WHERE ACCEPTED MANUSCRIPT WILL BE SUBMITTED (PubMed Central, LOCKSS etc)].

All PLOS ONE articles are deposited in PubMed Central and LOCKSS. If your institute, or those of your co-authors, has its own repository, we recommend that you also deposit the published online article there and include the name in your article.

b) In the Results section, the globally unique identifier (GUID), currently in the form of a Life Science Identifier (LSID), should be listed under the new species name, for example:

Solanum aspersum S.Knapp, sp. nov. [urn:lsid:ipni.org:names:77103633-1] Type: Colombia. Putumayo: vertiente oriental de la Cordillera, entre Sachamates y San Francisco de Sibundoy, 1600-1750 m, 30 Dec 1940, J. Cuatrecasas 11471 (holotype, COL; isotypes, F [F-1335119], US [US-1799731]).

PLOS ONE staff will contact IPNI to obtain the GUID (LSID) after your manuscript is accepted for publication, and this information will then be added to the manuscript at that time. You may indicate where the number or numbers will be added using XXXXX or an equivalent placeholder.

A complete explanation of our guidelines for publishing new species can be found on our website: http://www.plosone.org/static/guidelines#botanical

5. We note that Figure 27 in your submission contain map images which may be copyrighted. All PLOS content is published under the Creative Commons Attribution License (CC BY 4.0), which means that the manuscript, images, and Supporting Information files will be freely available online, and any third party is permitted to access, download, copy, distribute, and use these materials in any way, even commercially, with proper attribution. For these reasons, we cannot publish previously copyrighted maps or satellite images created using proprietary data, such as Google software (Google Maps, Street View, and Earth). For more information, see our copyright guidelines: http://journals.plos.org/plosone/s/licenses-and-copyright.

a. You may seek permission from the original copyright holder of Figure 27 to publish the content specifically under the CC BY 4.0 license.  

Reviewers' comments:

Reviewer's Responses to Questions

**Comments to the Author**

1. Is the manuscript technically sound, and do the data support the conclusions?

Reviewer #1: Partly

Reviewer #2: Yes

Reviewer #3: Yes

2. Has the statistical analysis been performed appropriately and rigorously? 

Reviewer #1: N/A

Reviewer #2: I Don't Know

Reviewer #3: Yes

3. Have the authors made all data underlying the findings in their manuscript fully available?

Reviewer #1: Yes

Reviewer #2: Yes

Reviewer #3: Yes

4. Is the manuscript presented in an intelligible fashion and written in standard English?

Reviewer #1: Yes

Reviewer #2: Yes

Reviewer #3: Yes

5. Review Comments to the Author

Reviewer #1: I have read the manuscript and the reviews with the Author’s comments. The manuscript arouse a lively discussion between the Author and the Reviewers, the tension between them was clearly visible. Therefore, I hope my opinion will be helpful. After I read the manuscript, I had an impression it should be sent (after improvements mentioned below) rather to some purely taxonomic journal more than to PLOSone. I also think, the paper has rather poor chances to be cited. The Author has 165 papers indexed by SCOPUS, which is a huge number, of course. However, the h-index is 15. It is not much for such a numerous bibliography (sorry to rise this issue, but it was mentioned in the comments that the Author published 300 papers therefore I decided to check the indexing).

Graphs and photos are nicely done. I think the Author put a lot of effort to prepare the documentary. However, in my opinion, the “scale” problem was not solved properly. I do agree with the Reviewer that the scale is needed and helpful for a potential reader, and does not disturb the plates’ aesthetics (which is a secondary issue, anyway). However, drawing just a line in on the graph and providing in the figure caption that e.g. “Scales= 2 mm” is vague. It would be better to provide a number next to the line in the graph, with additional information about the unit in the caption e.g. “Scale bars in mm”. This is of course just a small point, which is easy to be solved. I also think, the Author should provide specification of the optical equipment used for all measurements and photographic documentation.

About the molecular analyses – I also agree with the Reviewer that such the study should be done and presented. The Author mentioned in the Introduction that molecular analyses are scheduled for the future. Such the statement indicate that the study are not finished. Molecular analyses may bring some interesting results changing the currently presented results. I would rather recommend to perform the analyses and presented them with conchological and anatomical approach in a single manuscript. I also feel that providing an information that “Shell about 40 mm” is too general. The Author should prepare a summarizing table (or two tables – one for conchological traits, the second – for anatomical traits) containing the number of shells and specimens analyzed, with the measurements range for each of the characteristics, mean value, standard deviation, as well as measurements for a holotype. Such the table would made easier to find the data about the number of shell measured and specimens dissected, as the information is, indeed, lost in the text. The Author did not provide a number of dissected snails for all the species. In the main text I found the information on 9 specimens of A. multicolor and 2 specimens of C. uhlei, while figures containing anatomical characteristics are given for A. multicolor, A. synchondrus, A. vailanti, A. morenus, A. savanticus, and C. uhlei (In the caption of the Fig. 11 the Author mentioned that he used 2 specimens, but no data is available within the main document).

Overall recommendation: Although the Author put lot of effort to prepare the manuscript, I cannot recommend the paper to be published in PLOSone. In my opinion, the material presented is more adequate for some purely taxonomic journal. Apart of this suggestion, I think the data are too scanty - molecular analyses pending. The data are presented vaguely. I also feel that some summarization of all measurements, containing data on all material analyzed is extremely needed.

Reviewer #2: Dear author and editor - I would suggest this paper is accepted for review with some very minor revisions.

The paper is a well written description of the new species and an important addition to our knowledge of Molluscs from this group/region. I have made a few suggestions to the text to make the article clearer to the readers and there are a few points I have asked for clarification where the meaning is not clear to me. Unlike previous reviewers I do not feel molecular work is necessary for new species descriptions (at least in molluscs) where there is a history and ongoing precedent of species descriptions based on shell only let alone anatomical characters. I thought the phylogenetic work was a nice addition to the paper.

The species descriptions are done to a high standard and are easily understandable to the malacological community. May I suggest a short (2-3 sentences) 'differential diagnosis' added to the end of each description to succinctly allow this species differences to be highlighted.

I feel that the discussion is very thorough but my feeling is it is overly long and complex and by adding some 'differential diagnosis' to the species and generic descriptions this could be condensed and simplified.

In the 'specific debate' there is a lot of shell information which whilst useful is difficult to follow. I wonder if a table showing data such as aperture size, W/L index and spire angle etc. could be used to shorten and simplify this?

Possibly more difficult but again could a table be used to highlight anatomical differences to shorten/simply or even add to the data presented.

I found the 'anatomical and broader taxonomic' section very informative and well structured.

I am not a phylogenetic specialist but the addition of this data was interesting and understandable to me.

The references are well structured and wide spread and the figures are well produced and clearly understandable. I do prefer scale bars in figures but I understand that when so many images are made available it would not be applicable in such a layout without it appearing messy.

If the author or editor have any further questions I am happy to contacted.

Reviewer #3: The revised ms is much improved, and has greatly benefited from inclusion of the anatomical data for the type species of Anthinus.

No page numbering on ms. Main comments refer to sections of text identified by numbers pencilled in the left margin of the annotated ms.

Abstract

1. “A bulimulid, a simpulopsid and two dorcasiids are outgroups.”

2. “Mirinaba and Anthinus also resulted monophyletic (3 and 6 synapomorphies respectively.”

Suggest changing to “Mirinaba and Anthinus were also supported as monophyletic (3 and 6 synapomorphies respectively).

Introduction

3. “The description of the type species is, the remaining descriptions and illustrations, to minimize redundancy, are mostly presented compared to it.”

Sentence ambiguous. Re-work.

4. “The anatomical base of all of them will certainly be useful for future taxonomic revisions and for phylogenetic inferences.”

Suggest modifying as

The anatomical information presented on these new taxa will certainly be useful for future taxonomic revisions and for phylogenetic inferences.

5. “Despite these aspects are part of a large revisional project in such this study is inserted, an initial phylogenetic treatment is provided in this paper, based on the species with anatomy more detailly known in the literature and in the herein studied species. The main intention is not to present a “phylogeny of the Strophocheilidae”, as many more species would be necessary, but to provide a better scenario for a discussion and the base for the new taxa.

Suggest the following changes

Despite these aspects are part of a large revisional project in such this study is inserted, An initial phylogenetic treatment is provided in this paper, based on species for which anatomy is reasonably well known in the literature, coupled with new information on the species herein erected. The main intention is not to present a “phylogeny of the Strophocheilidae”, as anatomical investigation many more species would be necessary, but as the basis for erection of the new taxa, and to provide a better scenario for a comparative discussion.

Materials and methods

6. “The anatomical drawings are an average of all examined specimens. Color photos were obtained by digital cameras, either hand-held or attached to the dissecting microscope. Shell measurements were obtained with a digital caliper, in a minimum of 10 adult shells. The specimens were dissected by standard techniques [13] under dissecting stereomicroscopes, with the specimen immersed under the fixative. All drawings were obtained with the aid of a camera lucida; initially pencilled, afterwards inked; usually drawings produced for each species include data derived from several specimens. The type and voucher….”

Suggest a slight restructure of paragraph

Color photos were obtained by digital cameras, either hand-held or attached to the dissecting microscope. Shell measurements were obtained with a digital caliper, for a minimum of 10 adult shells. The specimens were dissected by standard techniques [13] under dissecting stereomicroscopes, with the specimen immersed under the fixative. All drawings were obtained with the aid of a camera lucida; initially pencilled, afterwards inked; usually drawings produced for each species include data derived from several specimens. Thus, the anatomical drawings are an average (composite) of all examined specimens. The type and voucher ….

There is a potential issue arising with presentation composite illustrations rather than illustrating data gathered from individual specimens. Potentially information is lost on variation (if any) among the examined specimens. As a consequence, if there was variation between the examined type specimen [the name bearing specimen sensu Code of Zoological Nomenclature] and other examined specimens [non-name-bearing paratypes, topotypes, etc] then it may be ambiguous as to whether the composite encompasses anatomical information drawn from more than one species, such as can occur if one or more specimens were misidentified as belonging to the species in question. Is ambiguity can only be reduced if there is illustration of anatomical features for all examined specimens, or if the author explicitly states there was insignificant or no variation among these specimens.

Where only non-name-bearing specimens are examined ( e.g. paratypes, topotypes, etc) then there will always be uncertainty as to whether this information also pertains to the species as defined by the name-bearing holotype.

7. “… The collector team works for a company called Femorale [www.femorale.com; http://www.femorale.com/femorale/index.asp]. The places of collection are not protected areas, which do not require special permits. The collections, anyway, are supported by the general/permanent license IBAMA-Sisbio 10560-2, which permits extraction of wildlife samples for scientific purposes. ….”

Suggest the following:

The material studied and described in this work was collected by a team working for Femorale, a private company [www.femorale.com; http://www.femorale.com/femorale/index.asp]. The places of collection are not within protected areas, and as such collection activity did not require special permits. Nonetheless, the collections were made under general/permanent license IBAMA-Sisbio 10560-2, which permits extraction of wildlife samples for scientific purposes.

8. “Thus, details on vegetation, climate, soil, rainfall, etc., were not available, but, when relevant, these data were extracted from the literature or official websites…”

Suggest

Thus, details on vegetation, climate, soil, rainfall, etc., were not available, but, when possible and relevant, these data were extracted from the literature, digital online resources, or official websites.

9. “For comparison of the presently studied species with those already known, the large MZSP collection was consulted, as a previous curator, Dr. José L.M. Leme, was a strophocheilid specialist.”

Suggest

For comparison of the presently studied species with those already known, the large MZSP collection was consulted, including the material assembled by Dr. José L.M. Leme, a previous curator and strophocheilid specialist.

10. “Although additional approaches can be added to any taxonomical paper, such as multivariate statistical analyses of shell measurements, molecular sequencing, etc., the presently presented anatomical investigation was so informative and convincing that it already bases very well the taxonomic conclusions. Of course, other approaches, molecular in particular, will be explored in future papers, considering an already well-based taxa provided by the present paper. It is important to emphasize that most of the land snails taxonomy is based only on their shells alone; the present paper goes further and presents convincing anatomical descriptions.”

This text has been added following adverse comments from previous reviewers, and is the author’s attempt to justify the methodology and pre-empt any future commentary. This text added by the author is unnecessary, redundant, as the published work should stand on its own merits.

11. “The chosen model to do the taxonomic remarks on each described species is reuniting all of them in a single Discussion, which ends this paper. This is interpreted as a more intelligible, fluent, and economic manner to explore the distinctions and similarities of the addressed taxa. Rather than individual remarks after each description.”

This text too was added by the author as a response to comments from previous reviewers. The criticism was not directed as the author’s model to aggregate the comparative remarks in the Discussion, but in the way that aggregation was done – too verbose, clumsy presentation of comparative morphometric information, lack of clarity about diagnostic features useful in species discrimination and identification, ….

I think this text in the main is redundant and should be replaced by a single sentence stating the fact that the taxonomic remarks are aggregated into the Discussion.

12. “Phylogenetic analysis. In a family such as Strophocheilidae, …..

Elements of this text belongs in Results and Discussion. All that is needed here is a description of the methodology used.

“The phylogenetic methodology is the same as reported by Simone [13, 23], that basically consists of the matrix mounted in Nexus, analyzed by programs TNT and PAUP.” Is insufficient. Information needs to be added on phylogenetic analyses settings/search criteria etc. It is important that, given the data matrix, the results are reproducible [in as much as phylogenetic tree as estimates of a ‘true phylogeny’ are reproducible] based on the description given in Materials and Methods.

The Simone [13, 23] publications were consulted to determine what phylogenetic methodologies were used in those earlier publications. The description of the methods in Simone [13] are brief to the extreme, do not meet modern standards of reporting, and certainly are not informative about what analyses were actually undertaken for the present study, other than imply a maximum parsimony approach.

The link provided to Simone [23, Bosquejos de filogenia] only takes the reader to a small selection of pages that provide no relevant information. Evidently the full publication is not available for direct access on online, but can be requested.

13. Simone LRL. 2011. Phylogeny of the Caenogastropoda (Mollusca), based on comparative morphology. Arq Zool. 2011; 42: 161–323.

23. Simone LRL. Bosquejos de filogenia. Clube de Autores. Curitiba, 60 pp. https://www.google.com.br/books/edition/Bosquejos_De_Filogenia/b5TtDwAAQBAJ?hl=pt-BR&gbpv=0

13. “Nomenclatural acts. The electronic edition of this article conforms …..”

This paragraph has been added by the author, apparent in response to some commentary by earlier reviewers about the merits or validity of erecting taxa in online/digital-only journals. Personally I think this text is unnecessary, but do share the concerns about taxa being erected in the digital-only format. In any event, the Commission has been strongly challenged about this aspect of the Code.

14. “Material examined. Types (reported above).”

That is

“Types. Holotype MZSP 152074. Paratypes: MZSP 152252, 2 shells, MZSP 152075, 1 shell from type locality. BRAZIL. Minas Gerais; Unaí (W.V. Matos col, iv/2020), Gueda, 16°16’04”S 46°39’43”W, 647-650 m altitude, MZSP 152236, 1 shell.

Type locality. BRAZIL. Minas Gerais; Unaí, Pedra da Fartura, 16°31’37”S 46°49’18”W, 628- 750 m altitude [Weslley Vailant de Matos col, iv.2020].”

So which specimen(s) were used for the anatomical investigation?

Discussion

15. “The taxon descriptions presented above include comparative remarks, which shows the detailed differences amongst the taxa herein studied. This is particularly evidenced in given diagnoses. The present section is, thus, more focused on the most interesting features and those more relevant to taxonomy and present/future phylogenetic approaches to resolving relationships among South American strophocheilids.”

Contrary to the author’s statement, the taxon descriptions do not include comparative remarks. Apparently this is a legacy of an earlier version of the ms.

16. “…The anatomical features were fundamental to determine both as separate genera.”

‘both as separate genera’ is grammatically lazy. Better would be ‘… to determine these are distinct genera’.

17. “… Both, A. multicolor and C. uhlei, share short, blunt mesocones, particularly those of the rachidian teeth …”

‘both’ redundant. Delete.

18. “… The present discussion focuses in the characters, every numerical parameter is based on a N from 10 to 20 specimens and is an integer value. Related to the shell elongation, A. multicolor is the slenderer (width/length W/L index = 46%) (Figs. 1A–C, 25C), with a group orbiting W/L ~50%, such as A. synchondrus (Fig. 25E – 54%), A. albolabiatus (Fig. 25A – 49%), A. miersi (Fig. 25B – 51%) and A turnix (Fig. 25D – 55%); while another group has shells broader, encompassing species with W/L = 60%, such as A. vailanti, A. morenus (Fig. 25G) and A. savanicus (Fig. 25H). In respect to the spire angulation, A. synchondrus has the sharpest pointed spire, ~35°, followed by A. multicolor ~40°, and A. albolabiatus ~42°; A, turnix has the broader spire, with ~58°, followed by A. miersi ~55°; while the remaining species (A. morenus, A. vailanti and A. savanicus) have ~50° …..”

In this paragraph, and to a least extent in subsequent paragraphs, the author has persisted with the same presentation style criticised by reviewers of the earlier version of the ms. Thus this section of the discussion continues to suffer in being too verbose, clumsy presentation of comparative morphometric information, lack of clarity about diagnostic features useful in species discrimination and identification, etc. Suggestions on some alternative formats for presentation were suggested earlier, but have not been taken up. This is the author’s call, but at some detriment to clarity.

6. PLOS authors have the option to publish the peer review history of their article (what does this mean?). If published, this will include your full peer review and any attached files.

Reviewer #1: No

Reviewer #2: **Yes: **Jonathan Ablett

Reviewer #3: **Yes: **Gary M. Barker

---

## [Author Response · Author response to Decision Letter 0]

1 Jun 2022

I thank you and the three referees for the important comments that improved the present paper. This is the fourth round of submission, and this new reformulated paper is very different from the first version, result of the comments of all 6 reviewers. I think this new version is finally worthy for publication. However, if something is still missing, or some of the few disagreement points do not meet the Editorial requirement or judgement, I am available for changing. There is no need to refuse the publication because of them.

Additionally, I have no special funding for this study in particular. It was developed only supported by my own laboratory. This fact we do not use to report, instead if required. Additionally, I have no competing interest in any point.

Answers to the last email (April 1, 2022, by Rose Puetes) are in red in the cover-letter. But, note that most of them were already answered in "answers to reviewers" of the previous submission.

---

## [Decision Letter · Decision Letter 1]

25 Jul 2022

PONE-D-22-00958R1Additions to the genus Anthinus occurring in Minas Gerais and Goiás regions, Brazil, with description of five new species, one of them in the new related genus Catracca (Gastropoda, Eupulmonata, Strophocheilidae).PLOS ONE

Dear Dr. Simone,

Thank you for submitting your manuscript to PLOS ONE. After careful consideration, we feel that it has merit but does not fully meet PLOS ONE’s publication criteria as it currently stands. Therefore, we invite you to submit a revised version of the manuscript that addresses the points raised during the review process.

We look forward to receiving your revised manuscript.

Kind regards,

Michael Schubert

Academic Editor

PLOS ONE

Journal Requirements:

Reviewers' comments:

Reviewer's Responses to Questions

**Comments to the Author**

1. If the authors have adequately addressed your comments raised in a previous round of review and you feel that this manuscript is now acceptable for publication, you may indicate that here to bypass the “Comments to the Author” section, enter your conflict of interest statement in the “Confidential to Editor” section, and submit your "Accept" recommendation.

Reviewer #1: All comments have been addressed

Reviewer #2: All comments have been addressed

Reviewer #3: All comments have been addressed

2. Is the manuscript technically sound, and do the data support the conclusions?

Reviewer #1: Yes

Reviewer #2: Yes

Reviewer #3: Yes

3. Has the statistical analysis been performed appropriately and rigorously? 

Reviewer #1: N/A

Reviewer #2: I Don't Know

Reviewer #3: N/A

4. Have the authors made all data underlying the findings in their manuscript fully available?

Reviewer #1: Yes

Reviewer #2: Yes

Reviewer #3: Yes

5. Is the manuscript presented in an intelligible fashion and written in standard English?

Reviewer #1: Yes

Reviewer #2: Yes

Reviewer #3: Yes

6. Review Comments to the Author

Reviewer #1: Dear Author, thank you for detail answers. If the Editorial Board will allow using the bars as you decided, than OK.

Molecular analyses issue: Yes, I do undersnand that the other Review had other opinion on molecular analyses. This is why Editors ask couple of reviewers (not only one) for their opinion. I sustain my opinion, I think that molecular analyses would be worthy since may be decisive in species distinguishing. However, I do understand, that you do not have such the opportunity do perform such the studies, now. Therefore, if other reviewers are in accordance, I will not insist to do so.

However, I do insist to prepare an additional table with number of individuals studied for each of the species. The data are lost in the text. Please, in the tables you prepared (on shell and anatomical characteristics) add an additional column containing the data on how many (precisely) individuals you used for the measurements! The expression “several specimens” is not acceptable! I would prefer to have some data on the variation (e.g. SD) within the individuals you did collect and used for your study.

Reviewer #2: I think the manuscript is much improved in this revision. It is clearer and more concise.

I think the inclusion of table 1 and 2 are a great addition and make the differential diagnoses between species much easier to understand. However I think in places the discussion is overly long and could be condensed especially where points are repeated.

I hope the author and editor do not mind but I have made some suggestions to the grammar/sentence structure to improve the English in places (which is in general is of a high standard). There are a few areas where I am not sure what the author quite means and I have noted these for clarification.

I have added 'I don't know' to the question 'Has the statistical analysis been performed appropriately and rigorously?' since I am not an expert in this area. However I found the results easy to understand as a non-specialist.

Please find a word document attached with specific comments and queries

Overall I think this is a useful piece of research and of interest and importance to the community studying this group, and to the knowledge of the Brazilian malacofauna.

Reviewer #3: Several minor editorial corrections and changes to text suggested in the returned annotated manuscript.

7. PLOS authors have the option to publish the peer review history of their article (what does this mean?). If published, this will include your full peer review and any attached files.

Reviewer #1: No

Reviewer #2: **Yes: **Jonathan Ablett

Reviewer #3: **Yes: **Gary M. Barker

---

## [Author Response · Author response to Decision Letter 1]

28 Jul 2022

I thank to the editorial council and to the three referees for this time relatively positive analysis on my paper on Anthinus and Catracca from Brazil.

This new round of comments and criticisms were very pertinent and useful for further improving the paper. They are all fully accepted, and the new version that is now re-submitted has all of them integrally corrected.

I answer below only the more relevant questions IN RED, as all remaining ones were simply corrected, including those that were written directly on the manuscript.

Reviewer #1: However, I do insist to prepare an additional table with number of individuals studied for each of the species. The data are lost in the text. Please, in the tables you prepared (on shell and anatomical characteristics) add an additional column containing the data on how many (precisely) individuals you used for the measurements! The expression “several specimens” is not acceptable! I would prefer to have some data on the variation (e.g. SD) within the individuals you did collect and used for your study.

ANSWER: A Table was inserted in the “Material and methods” item as the referee demanded. It is the Table 1 now. This required a renumber of the remaining tables, being now Tables 2 and 3 in the new manuscript (MS). I also corrected the indicated remaining points on the new MS.

Reviewer #2: I think the manuscript is much improved in this revision. It is clearer and more concise.

I think the inclusion of table 1 and 2 are a great addition and make the differential diagnoses between species much easier to understand. However I think in places the discussion is overly long and could be condensed especially where points are repeated.

ANSWER: some parts of the text were removed as the referee demanded in order to condense it. However, that discussion is not a mere compilation of what is written in the table. It was also used to put the species’ distinctions into grades, ranks, which in a future will be useful in comparative and phylogenetic scenarios. Thus, the cut of the text was not too much. I hope the text can be maintained this way.

I hope the author and editor do not mind but I have made some suggestions to the grammar/sentence structure to improve the English in places (which is in general is of a high standard). There are a few areas where I am not sure what the author quite means and I have noted these for clarification.

ANSWER: Everything corrected.

Reviewer #3: Several minor editorial corrections and changes to text suggested in the returned annotated manuscript.

ANSWER: Everything corrected

---

## [Editor Report · Decision Letter 2]

3 Aug 2022

Additions to the genus Anthinus occurring in Minas Gerais and Goiás regions, Brazil, with description of five new species, one of them in the new related genus Catracca (Gastropoda, Eupulmonata, Strophocheilidae).

PONE-D-22-00958R2

Dear Dr. Simone,

We’re pleased to inform you that your manuscript has been judged scientifically suitable for publication and will be formally accepted for publication once it meets all outstanding technical requirements.

Kind regards,

Michael Schubert

Academic Editor

PLOS ONE

---

## [Editor Report · Acceptance letter]

12 Aug 2022

PONE-D-22-00958R2 

Additions to the genus *Anthinus* occurring in Minas Gerais and Goiás regions, Brazil, with description of five new species, one of them in the new related genus *Catracca* (Gastropoda, Eupulmonata, Strophocheilidae). 

Dear Dr. Simone:

I'm pleased to inform you that your manuscript has been deemed suitable for publication in PLOS ONE. Congratulations! Your manuscript is now with our production department. 

Kind regards, 

on behalf of

Dr. Michael Schubert 

Academic Editor

PLOS ONE